# Enhanced Generative Model Evaluation with Clipped Density and Coverage

**Nicolas Salvy**
Inria, CentraleSupélec, Université Paris-Saclay
nicolas.salvy@inria.fr

**Hugues Talbot**
CentraleSupélec, Inria, Université Paris-Saclay
hugues.talbot@centralesupelec.fr

**Bertrand Thirion**
Inria, CEA, Université Paris-Saclay
bertrand.thirion@inria.fr

## Abstract

Although generative models have made remarkable progress in recent years, their use in critical applications has been hindered by an inability to reliably evaluate the quality of their generated samples. Quality refers to at least two complementary concepts: fidelity and coverage. Current quality metrics often lack reliable, interpretable values due to an absence of calibration or insufficient robustness to outliers. To address these shortcomings, we introduce two novel metrics: *Clipped Density* and *Clipped Coverage*. By clipping individual sample contributions, as well as the radii of nearest neighbor balls for fidelity, our metrics prevent out-of-distribution samples from biasing the aggregated values. Through analytical and empirical calibration, these metrics demonstrate linear score degradation as the proportion of bad samples increases. Thus, they can be straightforwardly interpreted as equivalent proportions of good samples. Extensive experiments on synthetic and real-world datasets demonstrate that *Clipped Density* and *Clipped Coverage* outperform existing methods in terms of robustness, sensitivity, and interpretability when evaluating generative models.

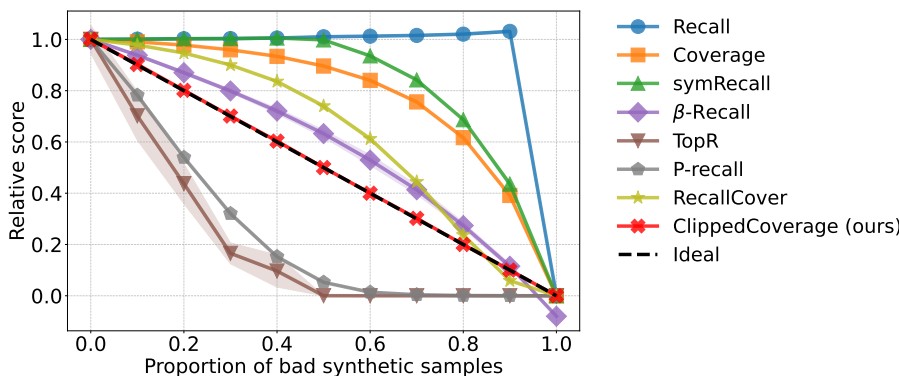

Figure 1: **Measuring the coverage of a mixture of good and bad samples (CIFAR-10)**: Various coverage metrics are evaluated relative to their maximum value as the proportion of bad synthetic samples increases. Only *Clipped Coverage* displays the desired linear degradation.

## 1 Introduction

In recent years, remarkable progress has been achieved in generative models, which are being actively explored in various fields, such as healthcare (Pinaya et al., 2022; Fernandez et al., 2024; Tudosiu et al., 2024; Zhu et al., 2024; Bluethgen et al., 2024). However, deploying them in high-stakes applications depends on reliably evaluating the quality of synthetic data to ensure its trustworthiness.

This evaluation is inherently challenging, especially for high-dimensional data. The true underlying distributions of this data are often unknown and complex. They also do not conform to known parametric families. These factors make computing the support or density infeasible in practice. Current model evaluation often relies on metrics such as Fréchet Inception Distance (FID) (Heusel et al., 2017) and FD-DINOv2 (Stein et al., 2023) for images. These metrics provide a single, compound score representing overall sample quality. Thus, it is impossible to determine whether poor performance stems from a lack of realism or variety (Sajjadi et al., 2018).

To address this issue, the quality of synthetic data can be broken down into at least two core concepts, fidelity and coverage, and measured separately with a pair of metrics. Fidelity metrics assess how similar each synthetic sample is to the input data (Naeem et al., 2020). Conversely, coverage metrics measure the extent to which synthetic samples represent the distribution of real data, taking into consideration how the rarity or commonness of real data is reflected in the synthetic samples. However, a recent position paper by Räisä et al. (2025) argues that all existing fidelity and diversity metrics are flawed, highlighting an urgent need for new metrics that address these shortcomings.

To determine whether generative models can be truly dependable, particularly in sensitive applications, the metrics used to evaluate them must be trustworthy. This means that they must be robust, sensitive to genuine deficiencies, and provide interpretable scores. A key challenge is robustness to outliers. Real-world datasets often contain out-of-distribution samples such as corruptions, anomalies, or simply samples very different from the rest (see Figure 7 for examples in CIFAR-10). Similarly, generative models can produce "bad" samples that are far from the real data distribution. These outliers can disproportionately influence evaluation scores, masking true performance issues.

Beyond robustness, interpretability is crucial. Metrics should offer more than just relative comparisons (i.e., knowing that one model is better than another). As emphasized by Räisä et al. (2025), for a metric to be truly useful in practice, its absolute value must be meaningful, since even the best-performing model in a comparison might still be of poor quality. Critically, Räisä et al. (2025) notes that, currently, no fidelity or coverage metric offers this property. Ideally, for straightforward interpretability, a score of $x$ would indicate performance equivalent to having a proportion of $x$ good samples and $(1 - x)$ bad ones. However, as shown in Figure 1, current coverage metrics fail to achieve this. We show in Figure 4 that current fidelity metrics are similarly untrustworthy.

In this paper, we introduce *Clipped Density* and *Clipped Coverage*, two novel metrics designed to overcome these limitations. Our contributions are:

- **Trustworthy Evaluation**: Our metrics achieve robustness to outliers by clipping individual sample contributions to the aggregate score and, for *Clipped Density*, by limiting the radii of nearest-neighbor spheres used to measure the density. This prevents out-of-distribution samples from skewing the evaluation while preserving sensitivity to genuine issues.

- **Interpretable Absolute Scores**: Through empirical calibration for *Clipped Density* and theoretical analysis for *Clipped Coverage*, we ensure that scores degrade linearly with the proportion of bad samples, providing absolute interpretability.

The paper is structured as follows: Section 2 provides background on existing metrics. Section 3 reviews related work. Section 4 introduces our *Clipped Density* and *Clipped Coverage* metrics. Section 5 details our experiments and results, and Section 6 discusses implications and limitations.

## 2 BACKGROUND

We consider a setting in which we are given $N$ i.i.d. samples $\{x_i^r\}_{i=1}^N$ from an unknown data (reference, or real) distribution $p_r$ and $M$ i.i.d. samples $\{x_j^s\}_{j=1}^M$ generated from an unknown synthetic data distribution $p_s$. In this section, we review relevant metrics for evaluating generative models that aim to disentangle two aspects of synthetic data: fidelity (how realistic each synthetic sample is) and coverage (how well the synthetic samples populate the real density distribution). These metrics serve as the basis for our proposed improvements.

Assuming given supports $S^r$ for the real distribution and $S^s$ for the synthetic distribution, *Precision* measures the proportion of synthetic samples that fall within $S^r$, while *Recall* measures the

proportion of i.i.d. real samples that fall in $S^s$ (Sajjadi et al., 2018; Simon et al., 2019).

$$\text{Precision} = \mathbb{P}_{p_s}[S^r] = \mathbb{P}_{p_s}[S^r \cap S^s] \qquad \text{Recall} = \mathbb{P}_{p_r}[S^s] = \mathbb{P}_{p_r}[S^r \cap S^s] \qquad (1)$$

However, the underlying densities $d^r$ and $d^s$ and their supports $S^r$ and $S^s$ are unknown, making such computation infeasible. In practice, *improved Precision* and *Recall* (Kynkäänniemi et al., 2019) approximate these supports by the union of balls centered at each observed sample, with a radius equal to the distance $\text{NND}_k$ from its center to its $k$-th Nearest Neighbor (Burman & Nolan, 1992). We denote by $\text{NND}_k^r$ and $\text{NND}_k^s$ the distance to the $k$-th nearest real and synthetic sample, respectively.

$$\text{iPrecision} = \mathbb{P}_{p_s}[\hat{S}^r \cap S^s] = \frac{1}{M} \sum_{j=1}^{M} \mathbf{1}_{x_j^s \in \cup_{i=1}^{N} B(x_i^r, \text{NND}_k^r(x_i^r))} \qquad (2)$$

$$\text{iRecall} = \mathbb{P}_{p_r}[S^r \cap \hat{S}^s] = \frac{1}{N} \sum_{i=1}^{N} \mathbf{1}_{x_i^r \in \cup_{j=1}^{M} B(x_j^s, \text{NND}_k^s(x_j^s))} \qquad (3)$$

Yet, *iPrecision* and *iRecall* are strongly biased by out-of-distribution samples. Real outliers, being far from their nearest neighbors, create large balls and bias the estimation of $S^r$. Similarly, bad synthetic samples compromise the approximation of the synthetic support (Naeem et al., 2020).

*Density* and *Coverage* (Naeem et al., 2020) aim to alleviate this issue by going beyond a binary in/out decision. *Density* counts, for each synthetic sample, how many real $k$-NN balls it falls within, normalized by $k$. Thus, outliers with large balls contribute only $\frac{1}{k}$ to the fidelity of a given synthetic sample, instead of 1 for *iPrecision* (Naeem et al., 2020). As approximating density is impractical in high-dimensional spaces, the *Density* metric instead uses distances through $k$-NN balls. It represents the average relative density $d^s/d^r$, providing a relative measure of how many synthetic points fall within real balls, normalized by their mass $k$. To avoid estimation bias when approximating the synthetic distribution, which may contain many bad samples, additional considerations are needed. *Coverage* flips the perspective (Naeem et al., 2020): instead of computing the proportion of real samples within at least one synthetic ball, *Coverage* calculates the proportion of real samples that are *covered*, by having at least one synthetic sample within their ball: Coverage $= \mathbb{P}_{p_r}[\hat{S}^r \cap S^s]$.

$$\text{Density} = \frac{1}{kM} \sum_{j=1}^{M} \sum_{i=1}^{N} \mathbf{1}_{x_j^s \in B(x_i^r, \text{NND}_k^r(x_i^r))} \qquad \text{Coverage} = \frac{1}{N} \sum_{i=1}^{N} \mathbf{1}_{\exists j, x_j^s \in B(x_i^r, \text{NND}_k^r(x_i^r))} \qquad (4)$$

Although *Density* is less affected by outliers in the target distribution, it remains influenced by them (Park & Kim, 2023). Additionally, *Density* is not bounded by 1 (Naeem et al., 2020), making interpretation challenging when empirical estimates exceed this value (Cheema & Urner, 2023; Kim et al., 2023). Similar to *iPrecision* and *iRecall*, *Coverage* is limited to analyzing supports and misses density mismatches: a few synthetic samples can cover high-density regions by lying within many real balls, and adding more synthetic samples there might not increase the score (Park & Kim, 2023).

## 3 RELATED WORK

Recent works have aimed to improve these metrics. *Precision Cover* and *Recall Cover*, analogous to *Precision* and *Recall*, were introduced by Cheema & Urner (2023). They count a ball as covered only if it contains at least $k' > 1$ samples. A probabilistic approach was introduced by Park & Kim (2023): *P-precision* and *P-recall*. These measure the probability that a synthetic (resp., real) sample lies within a random sub-support of the real (resp., synthetic) distribution.

To address the problem of large-radius balls, Khayatkhoei & AbdAlmageed (2023) have employed a dual-perspective approach with *symPrecision* and *symRecall*: *symRecall* is defined as the minimum of *Recall* and *Coverage*, while *symPrecision* is the minimum of *Precision* and the *complementary Precision* metric computed from the reversed perspective. Some alternative approaches do not rely on nearest-neighbor approximations. $\alpha$-*Precision* and $\beta$-*Recall* (Alaa et al., 2022) employ a one-class approach to estimate supports containing a fraction $\alpha$ (resp. $\beta$) of a dataset. They measure the proportion of the other dataset found within supports of varying levels. *Topological Precision and Recall* (Kim et al., 2023) estimate support via topologically conditioned density kernels.

Recently, Räisä et al. (2025) developed a benchmark of sanity checks for generative model metrics using synthetic data, finding that all existing fidelity and coverage metrics are flawed. Our own tests

based on real-world data (Figures 1, 4 and 5) confirm this conclusion. That work further emphasizes that no current metric is suitable for absolute evaluations, calling for more research in this area. Our work addresses these concerns directly by introducing new metrics that overcome these limitations. We show in Appendix H that our proposed metrics also perform well on their synthetic benchmark.

In the following section, we introduce metrics designed to resolve these issues by being *i)* robust to outliers, and by *ii)* providing clearly interpretable values.

## 4    METHOD

Our proposed metrics are designed to satisfy several key properties. Firstly, they should be *robust to outliers* (Desideratum 1). Secondly, the metrics should exhibit *linear score degradation*: if a proportion $x$ of samples are bad, the score should decrease by $x$ (Desideratum 2a). The metrics should be *normalized* between 0 and 1 (Desideratum 2b), allowing their values to be interpreted on their own.

These properties enable a straightforward interpretation: a score of $x$ indicates that the synthetic dataset achieves the same fidelity or coverage as a dataset composed of a proportion $x$ of real samples and $(1 - x)$ bad samples. This does not imply that the synthetic dataset is a mixture of good and bad samples, but rather that its fidelity or coverage is equivalent to that of such a mixture. A score difference of $y$ between two datasets corresponds to $y$ more bad samples in the equivalent scoring scenario.

### 4.1    CLIPPED DENSITY

#### 4.1.1    A ROBUST FIDELITY METRIC

To measure fidelity, the *Density* metric counts the number of real balls each synthetic sample falls within. This approach can exceed 1 and is vulnerable to outliers, as it relies on adaptive ball radii.

Figure 2 illustrates a failure case of the *Density* metric on a synthetic dataset containing a single bad sample (bottom left), with $k = 2$. In this configuration, the two centered points each receive a fidelity score of $\frac{3}{2}$, while the bad sample obtains a fidelity of 0. So, the overall dataset fidelity is computed as $\frac{1}{3}\left(2 \times \frac{3}{2} + 0\right) = 1$: an ideal score. While this is an example in dimension 2, the problem worsens in higher dimensions as the number of balls a sample can belong to increases (Radovanovic et al., 2010), allowing one over-occurring sample to mask an increasing number of bad synthetic samples.

To prevent over-occurring samples from masking defects, we modify the aggregation approach. The intuitive idea is to limit the contribution of each sample to the metric: the fidelity of any synthetic sample should not exceed 1. Applying this modification to Figure 2, the fidelity score becomes $\frac{1}{3}\left(2 \times \min(\frac{3}{2}, 1) + 0\right) = \frac{2}{3}$, which effectively detects the presence of the bad sample.

On the other hand, real outliers can have extremely large distances to their nearest neighbors, resulting in balls with significantly larger radii. In dimension $d$, the volume of a ball of radius $r$ is proportional to $r^d$. Not only do outliers create much larger balls, but any point in a low-density region does so as well, which dramatically skews the metrics. These balls have a fixed *mass* $k$, but their *volume* varies. Balls of large *volume* can contribute disproportionately more than the others. To ensure balanced contributions that limit outlier influence, we clip the radius of each ball to the median of the distances to the $k$-th nearest neighbor.

$$R_k(x_i^r) = \min\left(\text{NND}_k^r(x_i^r), \text{median}(\{\text{NND}_k^r(x_l)\}_{l=1}^N)\right) \tag{5}$$

This results in the following metric that satisfies Desideratum 1, robustness to outliers:

$$\text{ClippedDensity}_{\text{unnorm}} = \frac{1}{M}\sum_{j=1}^M \min\left(\frac{1}{k}\underbrace{\sum_{i=1}^N \mathbf{1}_{x_j^s \in B(x_i^r, R_k(x_i^r))}}_{\#\{\text{clipped real balls } x_j^s \text{ is within}\}}, 1\right) \tag{6}$$

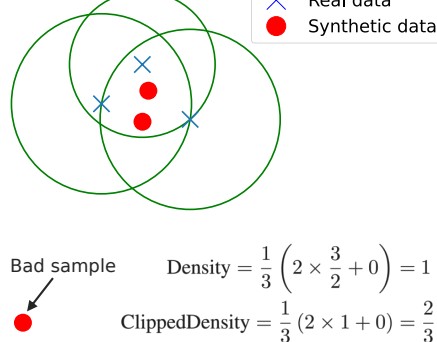

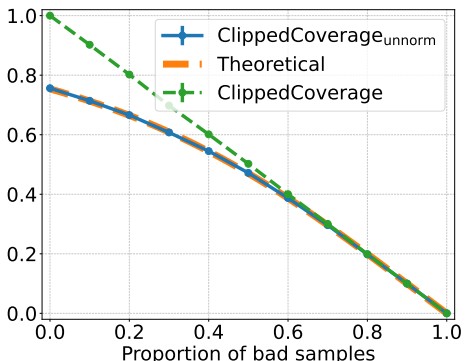

Figure 2: *Clipped Density* **corrects** *Density*'s **failure**: In a simple setup with a single bad synthetic sample, *Density* yields a value of 1. By clipping the fidelity of individual samples to 1, we obtain an adjusted score of $\frac{2}{3}$.

Figure 3: **Correcting** *Clipped Coverage* **linear decay**: The *unnormalized Clipped Coverage* (blue) does not decrease linearly with the proportion of bad samples. We theoretically compute its expected behavior (orange) and correct it (green).

### 4.1.2 NORMALIZING FOR INTERPRETABILITY

Since ClippedDensity$_{\text{unnorm}}$ is an average over synthetic samples, a proportion $x$ of bad samples directly reduces the score by $x$, satisfying Desideratum 2a. To achieve normalization between 0 and 1 (Desideratum 2b), we require an ideal value of 1. We achieve this ideal value empirically by evaluating the fidelity score of the real data using a leave-one-out strategy. For each sample $x_i^r$, we count the number of clipped real balls $j$ (with $j \neq i$) that contain it. This computation is efficient, as we have already obtained the indices of real samples inside each ball during the radius computation.

We normalize ClippedDensity$_{\text{unnorm}}$ by the value computed for real data, ClippedDensity$_{\text{real}}$, and clip the result to 1, since scores exceeding 1 would lack meaningful interpretation. The final normalized metric is:

$$\text{ClippedDensity} = \min\left(\frac{\text{ClippedDensity}_{\text{unnorm}}}{\text{ClippedDensity}_{\text{real}}}, 1\right) \tag{7}$$

### 4.2 CLIPPED COVERAGE

### 4.2.1 A ROBUST COVERAGE METRIC

To measure coverage, the *Coverage* metric considers the proportion of real samples whose balls contain at least one synthetic sample. To reflect the real data *distribution* and not just its support, we adopt an approach similar to *Clipped Density*. Instead of only checking for the presence of synthetic samples inside a real ball, we count how many there are. Then, to bound the contribution of each real sample, individual coverage scores are capped at 1. This results in the following formulation:

$$\text{ClippedCoverage}_{\text{unnorm}} = \frac{1}{N}\sum_{i=1}^{N}\min\left(\frac{1}{k}\underbrace{\sum_{j=1}^{M}\mathbf{1}_{x_j^s \in B(x_i^r, \text{NND}_k^r(x_i^r))}}_{\#\{\text{synthetic samples within } x_i^r\text{'s ball}\}}, 1\right) \tag{8}$$

For each real sample, we compare the *mass* of synthetic samples in its ball to the *mass* of real samples. To balance real points' contributions, the *mass* of each ball is fixed: no radius clipping is applied. ClippedCoverage$_{\text{unnorm}}$ satisfies Desideratum 1 of robustness to outliers.

### 4.2.2 CALIBRATING FOR INTERPRETABILITY

The blue curve in Figure 3 shows ClippedCoverage$_{\text{unnorm}}$ when the real and synthetic distributions are identical (far left) and as bad samples are progressively introduced into the synthetic distribution. To satisfy Desideratum 2, the score should follow $1 - x$ for $x$, the proportion of bad samples. To correct the metric's behavior, we start by deriving its expected value as a function of $x$.

**Lemma 1.** *If the real and synthetic distributions are identical, i.e., $\{x_i^r\}_{i=1}^N$ and $\{x_j^s\}_{j=1}^M$ are $N+M$ i.i.d. samples from the same distribution, then the expected value of ClippedCoverage$_{unnorm}$ is:*

$$\mathbf{E}\left[ClippedCoverage_{unnorm}\right] = \sum_{j=1}^{M} \min\left(\frac{j}{k}, 1\right) \binom{M}{j} \frac{\beta(k+j, M-j+N-k)}{\beta(k, N-k)} \tag{9}$$

*where $\beta$ is the beta function.*

The proof of Lemma 1 is provided in Appendix A. To parameterize the curve, we now consider the case with a proportion $x$ of bad synthetic samples. Since such samples always lie outside any real ball, the expected score becomes equivalent to the ideal case in Lemma 1 but with $M_x = \lfloor M(1-x) \rfloor$ synthetic samples instead of $M$. We denote this as $f_{\text{expected}}(x)$, shown in orange in Figure 3.

To satisfy Desideratum 2, the expected value of the metric should be $1 - x$ when the proportion of bad samples is $x$. This calibration ensures both a linear degradation of the score (Desideratum 2a) and normalization to a consistent [0, 1] range regardless of the dataset or choice of $k$ (Desideratum 2b). We seek a function $g$ such that $g \circ f_{\text{expected}}(x) = 1 - x$. Since $M_x$ can only take integers values $m$ between 0 and $M$, it suffices to find $g$ for $f_{\text{expected}}(M_x = m)$ where $m \in \{0, \dots, M\}$.

We can efficiently compute $f_{\text{expected}}(M_x = m)$ for all $m$ (see Appendix B.1). Given these values, the function $g$ is computed numerically. Since $f_{\text{expected}}$ decreases with $x$, we reverse it to form a sorted list of $f_{\text{expected}}$ values. For a given ClippedCoverage$_{\text{unnorm}}$ score $s$, we find the index $i(s) \in \{0, \dots, M\}$ such that inserting the value $s$ at index $i(s)$ keeps the list sorted. Then, $g(s) = 1 - \frac{i(s)}{M}$. The final normalized metric, which recovers the desired behavior shown in green in Figure 3, is:

$$ClippedCoverage = g \circ ClippedCoverage_{unnorm} \tag{10}$$

## 5 EXPERIMENTS

### 5.1 EXPERIMENTAL SETUP

We compared our metrics against multiple baselines, using the original code for *α-Precision*, *β-Recall*, and *TopP&R*, and reimplementing *Precision*, *Recall*, *Density*, *Coverage*, *symPrecision*, *symRecall*, *P-precision*, *P-recall*, and *Precision Recall Cover* for performance reasons (see Appendix B.2 for comparisons). For consistency with previous works, we set $k = 5$ where applicable; $k' = 3$ and $C = 3$, as recommended for *Precision Recall Cover*, and used default parameters everywhere else.

Experiments were conducted on a single NVIDIA H100 GPU with 80GB of RAM. When images are evaluated, metrics are computed on image data in the embedding space of the "Large" DINOv2 model (DINOv2-ViT-L/14 (Oquab et al., 2023)), as recommended by Stein et al. (2023).

### 5.2 METRIC EVALUATION TESTS

To ensure proper behavior, we evaluated metrics in various scenarios by controlling dataset composition and hence the expected scores. Most tests were performed on CIFAR-10 (Krizhevsky et al., 2009), with 2500 samples from each of the 10 classes for the real set and the other 2500 for the synthetic set. We report mean ± std over 10 splits. Results are summarized in Figures 4e and 5c.

**CIFAR-10 Simultaneous mode dropping:** To verify that fidelity metrics do not capture coverage, we performed a simultaneous mode dropping test on CIFAR-10 (Figure 4b, from Naeem et al. (2020)). In this test, the synthetic set progressively replaced samples from all but one class with samples from the remaining class, while the real set remained unchanged. To maintain a constant number of samples per set, only 2500 samples were used in this test, with results reported as mean ± std over 100 splits. Since the synthetic set is always a subset of CIFAR-10, fidelity scores should remain stable and close to their maximum. However, *Precision Cover* (yellow) and *symPrecision* (green) appear to capture coverage, as they deviate from their maximum value.

**CIFAR-10 matched real and synthetic out-of-distribution sample proportion:** To evaluate robustness to out-of-distribution samples, we conducted a test on CIFAR-10 (Figures 4c and 5a, inspired by Figure 5 of Naeem et al. (2020)), where we progressively replaced both real and synthetic

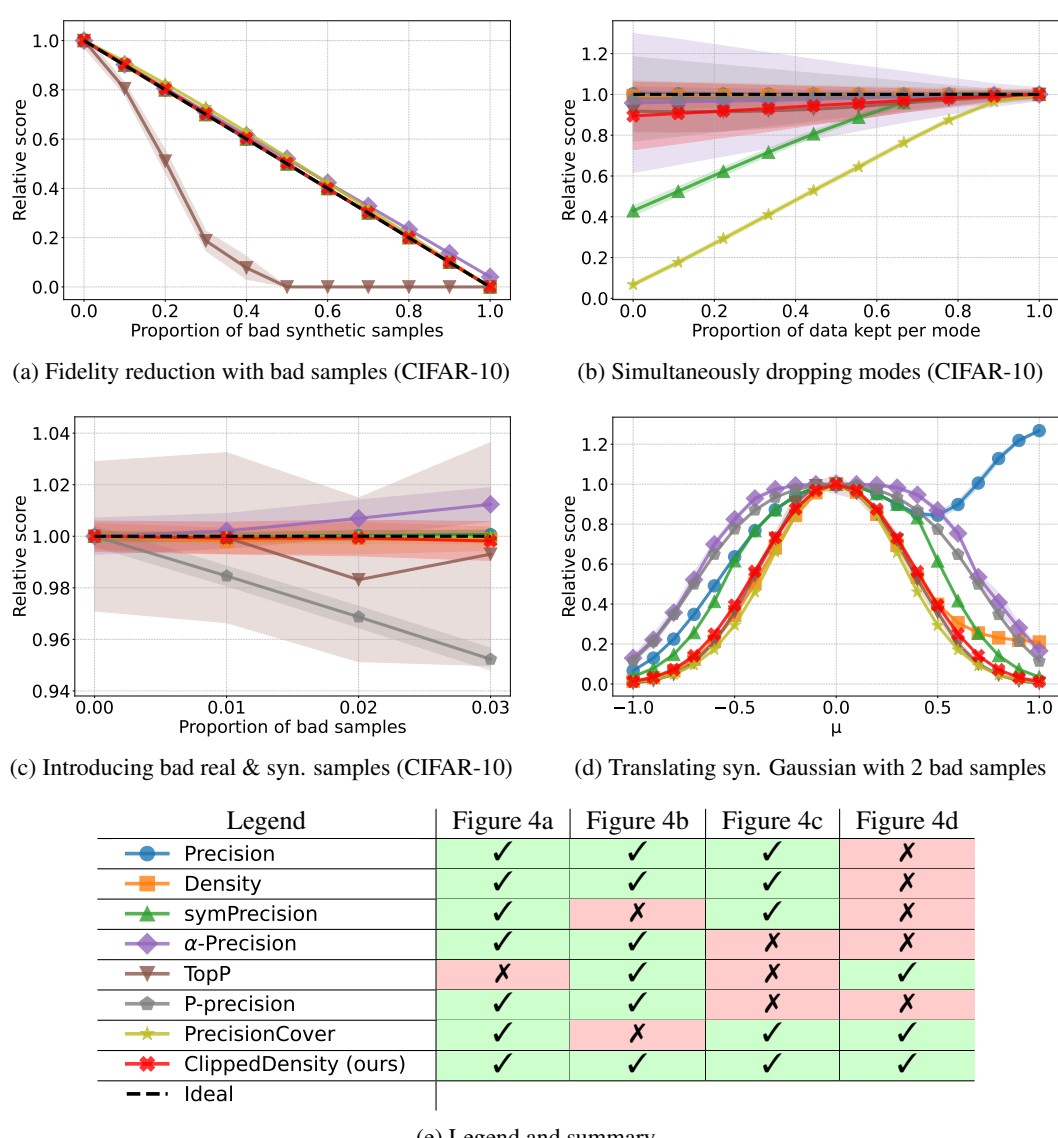

(a) Fidelity reduction with bad samples (CIFAR-10)

(b) Simultaneously dropping modes (CIFAR-10)

(c) Introducing bad real & syn. samples (CIFAR-10)

(d) Translating syn. Gaussian with 2 bad samples

| Legend | Figure 4a | Figure 4b | Figure 4c | Figure 4d |
|---|---|---|---|---|
| Precision | ✓ | ✓ | ✓ | ✗ |
| Density | ✓ | ✓ | ✓ | ✗ |
| symPrecision | ✓ | ✗ | ✓ | ✗ |
| $\alpha$-Precision | ✓ | ✓ | ✗ | ✗ |
| TopP | ✗ | ✓ | ✗ | ✓ |
| P-precision | ✓ | ✓ | ✗ | ✗ |
| PrecisionCover | ✓ | ✗ | ✓ | ✓ |
| ClippedDensity (ours) | ✓ | ✓ | ✓ | ✓ |
| Ideal | | | | |

(e) Legend and summary

Figure 4: **Testing fidelity metrics** (legend/summary in (e)). Scenarios: (a) increasing bad synthetic sample proportion; (b) simultaneous mode dropping: progressively replacing all but one class with the last class; (c) matched real & synthetic out-of-distribution samples at equal rates; (d) synthetic distribution translation with one real outlier and one bad synthetic sample. Only *Clipped Density* consistently behaves as expected: linearity (a), stability (b, c), and symmetry with sensitivity (d).

CIFAR-10 samples with out-of-distribution samples (noise images) at the same rate. Since the synthetic set is constructed identically to the real set in all cases, scores should remain stable and close to their maximum value. However, *α-Precision* (purple), *TopP* and *TopR* (brown), and *P-precision* and *P-Recall* (gray) show instability and deviate from the expected value.

**Synthetic data translation:** Since real-world datasets like CIFAR-10 contain outliers, we use a test based on synthetic data (Figures 4d and 5b, adapted from Naeem et al. (2020)) to evaluate robustness to the first out-of-distribution sample. We use 25000 standard Gaussian samples (dim 32, 5 splits). The synthetic set is translated by $\mu \in [-\mathbf{1}, \mathbf{1}]$ in all dimensions and includes a bad sample at $-\mathbf{3}$. The real set includes an outlier at $\mathbf{3}$. Ideally, scores should form symmetric bell-shaped curves, dropping rapidly as $\mu$ moves away from $0$ to detect distribution shifts. For fidelity (Figure 4d), *Precision* (blue) and *Density* (orange) are non-symmetric due to the real outlier's large ball, which in turn affects *symPrecision* (green). *α-Precision* (purple) and *P-precision* (gray) show low sensitivity (flat

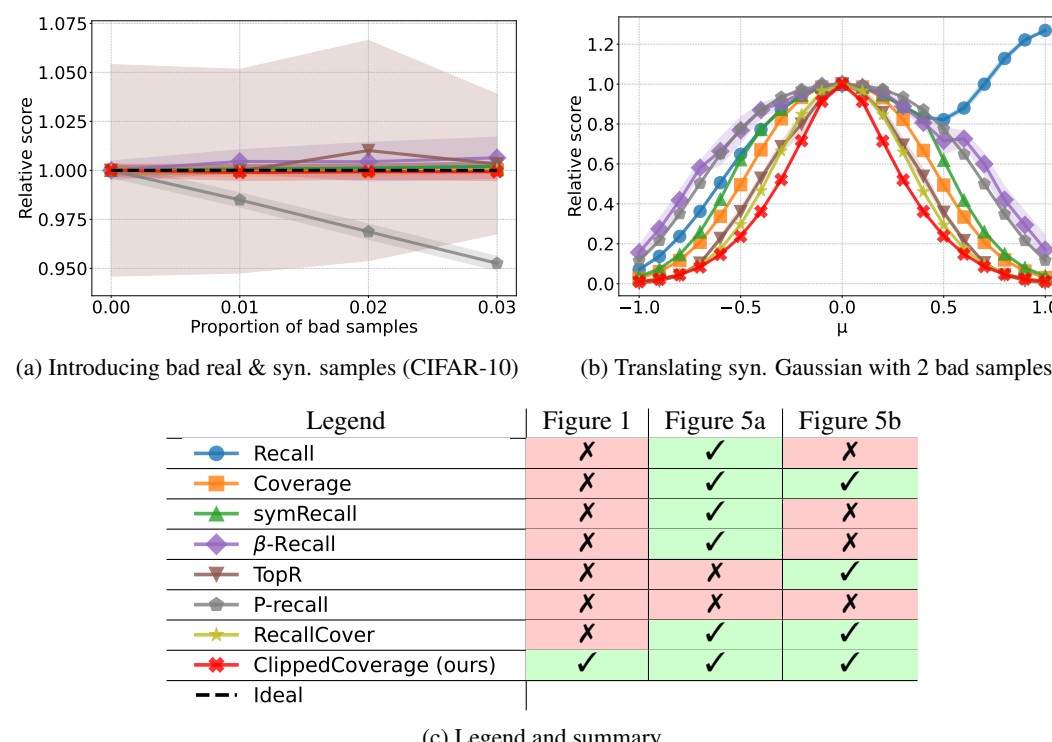

(a) Introducing bad real & syn. samples (CIFAR-10)      (b) Translating syn. Gaussian with 2 bad samples

| Legend | Figure 1 | Figure 5a | Figure 5b |
|---|---|---|---|
| ● Recall | ✗ | ✓ | ✗ |
| ■ Coverage | ✗ | ✓ | ✓ |
| ▲ symRecall | ✗ | ✓ | ✗ |
| ◆ $\beta$-Recall | ✗ | ✓ | ✗ |
| ▼ TopR | ✗ | ✗ | ✓ |
| ⬟ P-recall | ✗ | ✗ | ✗ |
| ★ RecallCover | ✗ | ✓ | ✓ |
| ✖ ClippedCoverage (ours) | ✓ | ✓ | ✓ |
| – – Ideal | | | |

(c) Legend and summary

Figure 5: **Testing coverage metrics** (legend/result summary in (c)). Metrics are evaluated under different scenarios: (a) matched real & synthetic out-of-distribution samples; (b) distribution translation with a real outlier at **3** and a bad synthetic sample at **-3**. *Clipped Coverage* exhibits all desired properties: linearity (Figure 1), stability (a), symmetry and sensitivity (b), unlike other metrics.

curves around 0). For coverage (Figure 5b), *Recall* (blue) is non-symmetric due to the bad synthetic sample, which also affects *symRecall* (green). This happens because the bad sample's ball grows as the rest of the synthetic data moves away, covering the real set when $\mu$ is near **1**. *$\beta$-recall* (purple) and *TopR* (brown) are also non-symmetric, indicating instability or insufficient robustness.

**CIFAR-10 Progressive bad sample introduction:** To test sensitivity and interpretability, we progressively introduced bad samples (noise images) into a synthetic set of CIFAR-10 images (Figures 1 and 4a). Scores should decrease linearly with the proportion of bad samples (see Section 4). For coverage metrics, only *Clipped Coverage* (red) exhibited this linear degradation. Most fidelity metrics, being averages of individual sample fidelities, decreased linearly. However, *TopP* (brown) deviated significantly, while *Precision Cover* (yellow) and *$\alpha$-Precision* (purple) showed slight deviations.

**Summary:** Across all results (see also Figures 4e and 5c), only *Clipped Density* and *Clipped Coverage* show the desired behavior in all tests. Other fidelity metrics either inappropriately capture coverage (e.g., *symPrecision, Precision Cover*), lack sensitivity (e.g., *$\alpha$-Precision, P-precision*), or lack robustness to outliers (e.g., *Precision, Density, TopP*). For coverage metrics, only *Clipped Coverage* shows a linear decrease in score with increasing bad sample fraction, while some metrics are also insufficiently robust to outliers (e.g., *Recall, symRecall, $\beta$-Recall, TopR, P-recall*).

## 5.3 EVALUATION ON REAL DATASETS

We evaluated various generative models on CIFAR-10 (Krizhevsky et al., 2009), ImageNet (Deng et al., 2009), LSUN Bedroom (Yu et al., 2015), and FFHQ (Kazemi & Sullivan, 2014), using categories and 50000 samples from the data publicly shared by Stein et al. (2023) (see Figure 6 and Appendix I). When possible, for conditional models, we kept an equal number of samples from each class (see Appendix C or Appendix A of Stein et al. (2023) for more details). A comparison with

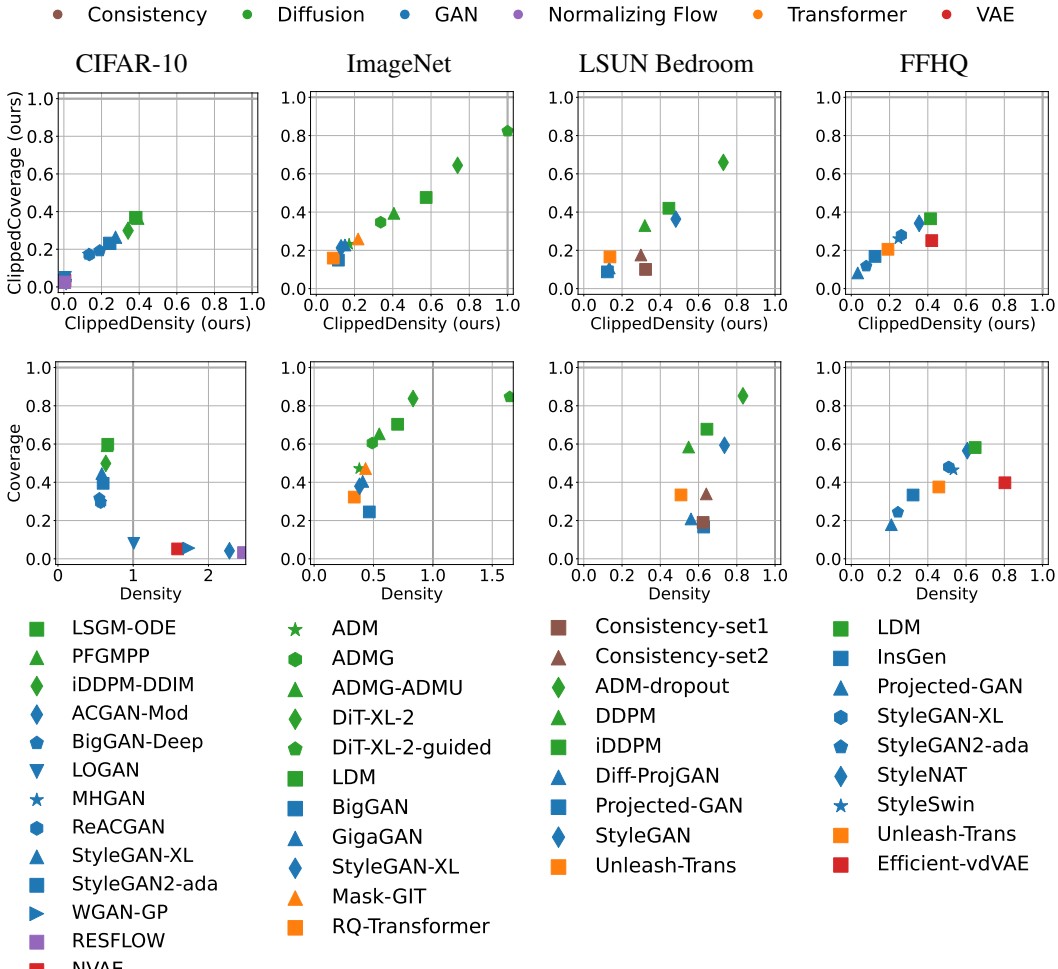

Figure 6: **Fidelity vs Coverage on various datasets**

human evaluations in Appendix G confirms that *Clipped Density* and *Clipped Coverage* align well with human judgment.

As shown in Figure 6, on CIFAR-10 and ImageNet, the *Density* values exceed 1, while the *Clipped Density* values remain stable. In Appendix D, we analyze RESFLOW-generated CIFAR-10 data, demonstrating that its inflated *Density* score of $2.47$ is driven by real out-of-distribution samples. Additionally, Appendix E provides a step-by-step evaluation of *Clipped Density* on the generated datasets, quantifying the impact of each modification to the original *Density* metric.

Across all datasets, our results consistently show diffusion models outperforming GANs. The range of values observed for our metrics appears to reflect the training dataset size: CIFAR-10 ($50k$ samples, max score $\approx 0.4$), FFHQ ($70k$, max $\approx 0.4$), LSUN Bedroom ($1.5M$, max $\approx 0.7$), and ImageNet ($14M$, max $1.0$). Interpreting the absolute scores, a value of $0.4$ for CIFAR-10 and FFHQ suggests that, on these datasets, top generative models achieve results equivalent to only $40\%$ of good samples and $60\%$ bad samples. This highlights substantial room for improvement, an insight only possible because the absolute values of *Clipped Density* and *Clipped Coverage* are interpretable.

This absolute interpretability also allows us to assess models without requiring a reference model for comparison, as demonstrated on Chest X-Rays in Appendix F.

Furthermore, using GAN-generated data with varying truncation parameters, we show in Appendix J that *Clipped Density* and *Clipped Coverage* effectively capture the expected trade-off between fidelity and coverage. We found *Clipped Density* and *Clipped Coverage* to be robust to the choice

of $k$ (Appendix L) and empirically studied their stability as a function of the sample size, showing a standard deviation proportional to $1/\sqrt{N}$ (Appendix M). Additionally, we tested the metrics in diverse contexts, including imputed distortions (Appendix N), failure scenarios such as missing rare modes (Appendix O), and other modalities like music data (Appendix P). Finally, samples from the datasets can be visually inspected in Appendix S.

## 6 DISCUSSION AND CONCLUSION

*Clipped Density* and *Clipped Coverage* offer significant improvements in robustness, sensitivity, and interpretability. These improvements result from several key design choices. To enhance robustness, we cap individual sample contributions at $1$. For *Clipped Density*, we further mitigate outlier impact by clipping the *volumes* of real nearest-neighbor balls to prevent them from dominating the score. In contrast, for *Clipped Coverage*, we maintain balls of constant *mass* to ensure balanced contributions. Capping the contribution prevents high-performing samples from disproportionately hiding low-performing ones. This careful handling of sample contributions and ball properties enhances robustness while preserving sensitivity.

For interpretability, both metrics are calibrated to degrade linearly as the proportion of bad samples increases. *Clipped Density* achieves this through normalization by the real data's score. For *Clipped Coverage*, whose uncalibrated behavior is non-linear (as shown in Figure 3), we theoretically derived its expected behavior and applied a correction to ensure linearity. This combination of robust aggregation and principled calibration allows *Clipped Density* and *Clipped Coverage* to provide not only trustworthy relative comparisons between models but also meaningful absolute scores. This is crucial in practice for assessing whether a model meets a required quality threshold.

Despite these improvements, limitations remain. *Clipped Density* and *Clipped Coverage* build on an accepted benchmark of progressively passed tests, but it is not exhaustive, and there might be missing cases. Furthermore, we did not provide a theoretical analysis of the metrics in the infinite sample limit. Additionally, while we evaluate fidelity and coverage, other aspects matter, such as *memorization*: are generated samples reproductions of training data? Memorization metrics include GAN-test (Shmelkov et al., 2018), identifiability (Yoon et al., 2020), authenticity (Alaa et al., 2022), and the calibrated $l_2$ distance (Carlini et al., 2023). We used DINOv2 as recommended (Stein et al., 2023), but comparing embedding models for generative model evaluation remains an open question.

In conclusion, *Clipped Density* and *Clipped Coverage* provide a trustworthy framework for generative model evaluation, offering robust, sensitive, and interpretable assessments.

Code to reproduce the experiments and use *Clipped Density* and *Clipped Coverage* is available at `https://github.com/nicolassalvy/ClippedDensityCoverage`.

### ACKNOWLEDGMENTS

This work benefited from state aid managed by the Agence Nationale de la Recherche under the France 2030 programme, reference ANR-22-PESN-0012 and from the European Union's Horizon 2020 Research Infrastructures Grant EBRAIN-Health 101058516. This work was performed using HPC resources from GENCI-IDRIS (Grant 2024-AD011014887R1).

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

CONTENTS

# A  PROOF OF LEMMA 1

*Proof.*

$$
\mathbb{E}\left[\frac{1}{N}\sum_{i=1}^{N}\min\left(\frac{1}{k}\sum_{j=1}^{M}\mathbf{1}_{x_j^s\in B(x_i^r,\mathrm{NND}_k^r(x_i^r))},1\right)\right] = \mathbb{E}\left[\min\left(\frac{1}{k}\sum_{j=1}^{M}\mathbf{1}_{x_j^s\in B(x_1^r,\mathrm{NND}_k^r(x_1^r))},1\right)\right]
$$

$$
= \frac{1}{k}\mathbb{E}\left[\min\left(\underbrace{\sum_{j=1}^{M}\mathbf{1}_{x_j^s\in B(x_1^r,\mathrm{NND}_k^r(x_1^r))}}_{\#\{\text{synthetic samples in } x_1^r\text{'s ball}\}},k\right)\right]
$$

$$
= \frac{1}{k}\mathbb{E}\left[\underbrace{\sum_{k'=1}^{k}\mathbf{1}_{\mathrm{NN}_{k'}^s(x_1^r)\in B(x_1^r,\mathrm{NND}_k^r(x_1^r))}}_{\substack{\text{how many of the } k\text{-nearest synthetic samples}\\ \text{of } x_1^r \text{ are within its ball}}}\right]
$$

$$
= \frac{1}{k}\sum_{k'=1}^{k}\mathbb{P}\left(\mathrm{NN}_{k'}^s(x_1^r)\in B(x_1^r,\mathrm{NND}_k^r(x_1^r))\right)
$$

We apply the linearity of expectation in lines 1, 2, and 4, and use the i.i.d. hypothesis for the $x_i^r$ in line 1. For line 3, we use the fact that the number of synthetic samples within the ball of $x_1^r$, when limited to $k$, is equal to the number of synthetic samples among the $k$-nearest synthetic neighbors of $x_1^r$ that fall within that ball. Here, $\mathrm{NN}_k^r$ denotes the $k$-th nearest real data sample, while $\mathrm{NN}_{k'}^s$ denotes the $k'$-th nearest synthetic data sample.

Let $S^r = \{\|x_i^r - x_1^r\|\}_{i=2}^N$ be the set of real distances to $x_1^r$, and let $S_k^r$ be the $k$-th order statistic of $S^r$ (i.e., the $k$-th smallest element): $S_k^r = \|\mathrm{NN}_k^r(x_1^r) - x_1^r\|$. Similarly, let $S^s = \{\|x_j^s - x_1^r\|\}_{j=1}^M$ be the set of synthetic distances to $x_1^r$, and let $S_k^s$ be the $k$-th order statistic of $S^s$: $S_k^s = \|\mathrm{NN}_k^s(x_1^r) - x_1^r\|$. Let $C_k^1$ be the number of synthetic samples contained in the $k$-ball of $x_1^r$.

$$
\begin{aligned}
\mathbb{P}\left(\mathrm{NN}_{k'}^s(x_1^r)\in B(x_1^r,\mathrm{NND}_k^r(x_1^r))\right) &= \mathbb{P}\left(\|\mathrm{NN}_{k'}^s(x_1^r) - x_1^r\| \le \|\mathrm{NN}_k^r(x_1^r) - x_1^r\|\right)\\
&= \mathbb{P}\left(S_{k'}^s \le S_k^r\right)\\
&= \mathbb{P}\left(k' \le C_k^1\right)
\end{aligned}
$$

$S_k^r$ divides the population into two parts: $k$ elements $\le S_k^r$ and $N-1-k$ elements $> S_k^r$. Since the real and synthetic distributions are the same, for a fixed $S_k^r$, $C_k^1$ follows a binomial distribution with parameters: number of trials $M$ and probability of success equal to $F(S_k^r)$, where $F$ is the CDF of the random variable $\|X - x_1^r\|$, with $X \sim p_r$. Thus, $C_k^1|S_k^r \sim \mathrm{Binomial}(M, F(S_k^r))$.

Since $S_k^r$ is random, so is $F(S_k^r)$. For any continuous distribution $Y$, $F(Y)$ is uniform (Embrechts & Hofert, 2013). Because the CDF is monotonically increasing, $F(S_k^r)$, the CDF of the $k$-th order statistic of $S^r$, is the $k$-th smallest element of the set $F(S^r)$: $F(S_k^r) = F(S^r)_k$. The $k$-th order statistic of a uniform distribution follows a Beta distribution (Gentle, 2009), so we have: $F(S_k^r) \sim \mathrm{Beta}(k, (N-1)-k+1) \sim \mathrm{Beta}(k, N-k)$.

When the success probability of a binomial distribution is itself a random variable following a Beta distribution, the resulting distribution is a Beta-Binomial distribution: $C_k^1 \sim \mathrm{Beta\text{-}Binomial}(M, k, N-k)$.

Let $\beta$ be the beta function:

$$
\mathbb{P}(k' \le C_k^1) = \sum_{j=k'}^{M}\binom{M}{j}\frac{\beta(k+j, M-j+N-k)}{\beta(k, N-k)}
$$

$$\frac{1}{k}\sum_{k'=1}^{k}\mathbb{P}\left(\mathrm{NN}_{k'}^{s}(x_1^r)\in B(x_1^r,\mathrm{NND}_k^r(x_1^r))\right)=\frac{1}{k}\sum_{k'=1}^{k}\sum_{j=k'}^{M}\binom{M}{j}\frac{\beta(k+j,M-j+N-k)}{\beta(k,N-k)}$$

$$=\frac{1}{k}\sum_{j=1}^{M}\sum_{k'=1}^{\min(j,k)}\binom{M}{j}\frac{\beta(k+j,M-j+N-k)}{\beta(k,N-k)}$$

$$=\sum_{j=1}^{M}\min\left(\frac{j}{k},1\right)\binom{M}{j}\frac{\beta(k+j,M-j+N-k)}{\beta(k,N-k)}$$

$\square$

## B  IMPLEMENTATION DETAILS

### B.1  CLIPPED COVERAGE

To compute $f_{\text{expected}}$ efficiently for all $m$, we begin by precomputing log-gamma values for all integers $l$ between $1$ and $M+N+1$. Then, all required beta values and binomial coefficients can be computed as differences of three precomputed log-gamma values: $\log\text{-Beta}(a,b)=\log\text{-Gamma}(a)+\log\text{-Gamma}(b)-\log\text{-Gamma}(a+b)$ and $\log\text{-}\binom{n}{k}=\log\text{-Gamma}(n+1)-\log\text{-Gamma}(k+1)-\log\text{-Gamma}(n-k+1)$.

For the computation of Clipped Coverage, we take an approach similar to that used in the proof of Lemma 1: we count how many of the $k$-closest synthetic samples to a real sample are contained within its ball. This can be computed using only nearest neighbor searches, which is more efficient than searching for all synthetic samples within the ball of each real sample.

### B.2  COMPARISON WITH EXISTING METRIC IMPLEMENTATIONS

We compare our implementation with the implementations of:

- *Precision*, *Recall*, *Density*, and *Coverage* by Naeem et al. (2020) (`https://github.com/clovaai/generative-evaluation-prdc/blob/master/prdc/prdc.py`)
- *symPrecision* and *symRecall* by Khayatkhoei & AbdAlmageed (2023) (`https://github.com/mahyarkoy/emergent_asymmetry_pr/blob/main/manifmetric/manifmetric.py`)
- *P-precision* and *P-recall* by Park & Kim (2023) using GPU (`https://github.com/kdst-team/Probablistic_precision_recall/blob/master/metric/pp_pr.py`)
- Our own implementation of *Precision Recall Cover*, following the pseudocode in Appendix A.3 of the original paper (Cheema & Urner, 2023)

Our implementation simultaneously computes *Precision*, *Recall*, *Density*, and *Coverage*, with optional computation of *symPrecision*, *symRecall*, *P-precision*, and *P-recall*. Since some intermediary computations are shared between these metrics, we compute them once and reuse them. We compute *Precision Recall Cover* separately because the $k$ value used differs from the other metrics ($k'=3$ instead of $k=5$, see Section 5).

In the original implementations, the number of parallel threads is sometimes hardcoded to $8$, so for a fair comparison, we use only $8$ of the $24$ threads available on our hardware, which is an H100 GPU with 80GB of RAM (as stated in Section 5).

We run the implementations on sets of $10000$ standard Gaussians in dimension $32$ and on the DINOv2 embedding of FFHQ with the synthetic set generated by the Latent Diffusion Model (LDM), hereafter "DINOv2-FFHQ-LDM", for a total of $50000$ samples in both the real and synthetic sets. The results are shown in Table 1.

Table 1: **Implementation comparison**

| | 10000 standard Gaussians, $d = 32$ | | | | DINOv2-FFHQ-LDM | | | |
| | Values | | Time (s) | | Values | | Time (min) | |
| Metric name | Initial | Ours | Initial | Ours | Initial | Ours | Initial | Ours |
|---|---|---|---|---|---|---|---|---|
| Precision | 0.7706 | 0.7706 | | | 0.7352 | 0.7352 | | |
| Recall | 0.7764 | 0.7764 | 3.93 | | 0.3969 | 0.3969 | 2.05 | |
| Density | 0.9591 | 0.9591 | | | 0.6475 | 0.6475 | | |
| Coverage | 0.9645 | 0.9645 | | 8.43 | 0.5817 | 0.5817 | | 25.2 |
| symPrecision | 0.7706 | 0.7706 | 17.3 | | 0.3267 | 0.3267 | 157 | |
| symRecall | 0.7764 | 0.7764 | | | 0.3969 | 0.3969 | | |
| P-precision | 0.9251 | 0.9251 | 1.66 | | DNF | 0.7229 | DNF | |
| P-recall | 0.9249 | 0.9249 | | | DNF | 0.6896 | | |
| Precision Cover | 0.9553 | 0.9553 | 55.7 | 2.23 | 0.1785 | 0.1785 | 88.2 | 2.90 |
| Recall Cover | 0.9357 | 0.9357 | 61.0 | 2.25 | 0.4637 | 0.4637 | 67.9 | 2.99 |

When both the original and our implementations finish, we obtain the exact same values. The original implementation of *P-precision* and *P-recall* does not finish (DNF) on the FFHQ test due to memory issues.

The main difference between our implementation and the existing ones is the use of Scikit-learn's (Pedregosa et al., 2011) `NearestNeighbors` method for the nearest neighbor search. We also use it for ball tree radius queries, which find all samples within a given radius of a specific point.

For simplicity, we assume here that the number of real and synthetic samples is the same: $N = M$. Ball trees are built in $O(dN \log N)$ time and $O(Nd)$ space (Omohundro, 1989; Huang & Tung, 2023) for $N$ samples in dimension $d$, while queries take $O(d \log N)$ operations in low dimensions and up to $O(dN)$ time in high dimensions (Liu et al., 2006) (note that this query is performed $N$ times).

In contrast, initial implementations usually compute the distances between all pairs of samples and search through these distances instead. The construction complexity is then $O(dN^2)$ time and $O(N^2)$ space. Finding all points within a given radius of a specific point can then be done in $O(N)$ time because the distances are already computed.

Our method therefore has at most the same time complexity as the initial implementation, but with memory usage of $O(Nd)$ instead of $O(N^2)$.

For DINOv2-FFHQ-LDM, where $N = 50000$ and $d = 1024$, the original *P-precision* and *P-recall* implementation fails due to out-of-memory errors, as it attempts to allocate an additional 24GB of RAM on top of the 73GB already in use ($73 = 24 \times 3 + 1$). This implementation stores four pairwise distance matrices, whereas `prdc.py` (for *Precision*, *Recall*, *Density*, and *Coverage*) stores only one.

## B.3 COMPUTATIONAL REQUIREMENTS

The approximate computation times for the experiments were as follows:

- Extracting DINOv2 embeddings took 3.7 minutes per dataset. With 46 datasets in total ($14 + 12 + 10 + 10$), this amounted to 2.8 hours.

- For synthetic data tests (25000 samples, dimension $d = 32$), each evaluation of all metrics took 8 minutes. The total computation time for these tests, encompassing various scenarios and repetitions ($21 \times 5 + (11 + 11/2 + 4) \times 10$), was 41 hours.

- For real tests using DINOv2 embeddings of CIFAR-10 (25000 samples, dimension $d = 1024$), each evaluation of all metrics took 13.7 minutes. The total computation time for these tests ($(11 + 11/2 + 4) \times 10$) was 47 hours.

- The evaluation of generative models on real datasets took 40 minutes per generated set. With 42 generated sets evaluated ($13 + 11 + 9 + 9$), this totaled 28 hours.

Overall, reproducing all the experimental results presented in this paper would require approximately 120 hours of computation time. The complete research project, including preliminary experiments and explorations not detailed in the final paper, required an estimated 200-300 hours of computation time.

## C  DATASETS

### C.1  REAL DATA FOR METRIC EVALUATION TESTS

For tests conducted on real data, the samples are DINOv2 embeddings derived from the CIFAR-10 dataset (Krizhevsky et al., 2009). When a test requires out-of-distribution (or "bad") samples, these are DINOv2 embeddings of Gaussian noise images.

### C.2  GENERATED DATASETS

This section details the generated datasets evaluated in Figures 6, 12 and 13. All data were publicly shared by Stein et al. (2023) through the link provided in their GitHub repository: `https://github.com/layer6ai-labs/dgm-eval`. For more information on the generation process, see Appendix A of Stein et al. (2023).

We used 50000 samples for each evaluation. For conditional models, an equal number of samples per class was taken, except when only 50000 unbalanced images were available.

#### C.2.1  CIFAR-10

For CIFAR-10 (Krizhevsky et al., 2009), data were generated using the following models:

- LSGM-ODE (Vahdat et al., 2021)
- PFGM++ (PFGMPP) (Xu et al., 2023)
- iDDPM-DDIM (Nichol & Dhariwal, 2021)
- StudioGAN models (Kang et al., 2023a)
    - ACGAN-Mod (Odena et al., 2017)
    - BigGAN (Brock et al., 2019)
    - LOGAN (Wu et al., 2019)
    - MHGAN (Turner et al., 2019)
    - ReACGAN (Kang et al., 2021)
    - WGAN-GP (Gulrajani et al., 2017)
- StyleGAN-XL (Sauer et al., 2022)
- StyleGAN2-ada (Karras et al., 2020)
- RESFLOW (Chen et al., 2019)
- NVAE (Vahdat & Kautz, 2020)

#### C.2.2  IMAGENET

For ImageNet (Deng et al., 2009), images were rescaled to $256 \times 256$ using center cropping followed by bicubic interpolation downsampling before being shared by Stein et al. (2023). The synthetic data were generated using the following models:

- Models for which datasets of 50000 unbalanced samples were initially shared by Dhariwal & Nichol (2021):
    - ADM (Dhariwal & Nichol, 2021)
    - ADMG (Dhariwal & Nichol, 2021)

- – ADMG-ADMU (Dhariwal & Nichol, 2021)
- – BigGAN (Brock et al., 2019)
- DiT-XL-2 (Peebles & Xie, 2023)
- DiT-XL-2-guided (Peebles & Xie, 2023)
- LDM (Rombach et al., 2022)
- GigaGAN (Kang et al., 2023b)
- StyleGAN-XL (Sauer et al., 2022)
- Mask-GIT (Chang et al., 2022)
- RQ-Transformer (Lee et al., 2022)

### C.2.3 LSUN Bedroom

For LSUN Bedroom (Yu et al., 2015), data were generated using the following models:

- Consistency (Song et al., 2023). We used the two sets provided by Stein et al. (2023), referred to as Consistency-set1 and Consistency-set2.
- Four models for which datasets of 50000 unbalanced samples were originally shared by Dhariwal & Nichol (2021):
  - – ADM-dropout (Dhariwal & Nichol, 2021)
  - – DDPM (Ho et al., 2020)
  - – iDDPM (Nichol & Dhariwal, 2021)
  - – StyleGAN (Karras et al., 2019)
- Diff-ProjGAN: Diffusion-Projected GAN (Wang et al., 2023)
- Projected GAN (Sauer et al., 2021)
- Unleash-Trans: Unleashing Transformers (Bond-Taylor et al., 2022)

### C.2.4 FFHQ

For FFHQ (Kazemi & Sullivan, 2014), images were downsampled to $256 \times 256$ using Lanczos interpolation before being shared by Stein et al. (2023). The data were generated using the following models:

- LDM (Rombach et al., 2022)
- InsGen (Yang et al., 2021)
- Projected-GAN (Sauer et al., 2021)
- StyleGAN-XL (Sauer et al., 2022)
- StyleGAN2-ada (Karras et al., 2020)
- StyleNAT (Walton et al., 2022)
- StyleSwin (Zhang et al., 2022)
- Unleash-Trans: Unleashing Transformers (Bond-Taylor et al., 2022)
- Efficient-vdVAE (Hazami et al., 2022)

# D    EXAMPLE OF REAL OUT-OF-DISTRIBUTION SAMPLES INFLATING *Density*

The RESFLOW-generated CIFAR-10 dataset achieves a *Density* of 2.47, a value inflated by real out-of-distribution samples. To investigate this, we categorized real samples by counting the number of synthetic samples within their 5-ball (Table 2).

Table 2: **Synthetic points per real $k$-ball (RESFLOW CIFAR-10).**

| Synthetic points in a real ball | Real balls (no clipping) | Real balls (with clipping) |
|---|---|---|
| 0 | 48373 | 49981 |
| 1–5 | 680 | 15 |
| 6–100 | 513 | 3 |
| 101–1000 | 282 | 1 |
| 1001–10000 | 148 | 0 |
| 10001+ | 4 | 0 |

Without radii clipping, 48373 real balls contain no synthetic points, whereas just 4 contain over 10000. The 4 corresponding real images appear out-of-distribution (see Figure 7).

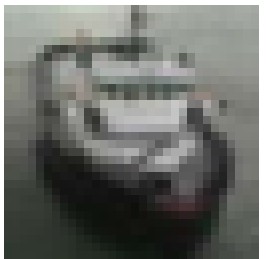 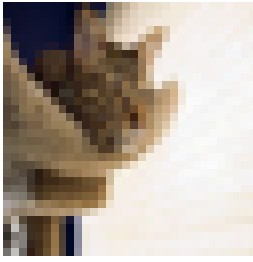 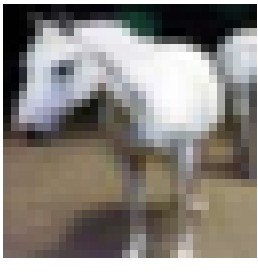 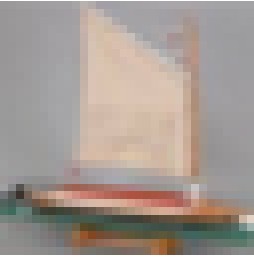

(a) Very gray ship image.    (b) Camouflaged cat.    (c) 2-legged horse.    (d) Toy ship on a stand.

Figure 7: **Real CIFAR-10 samples containing more than $10000$ synthetic points in their $5$-ball.** These images are atypical for their classes. (a) A ship viewed from above, maybe at night, resulting in a mostly gray image. (b) A cat on top of a similarly colored object. (c) An unusually shaped horse that appears to have only two legs. (d) A toy ship on a stand instead of in water.

A real point having many synthetic points in its ball is not inherently problematic, as generative models might generate many similar images. However, in this case, these 4 images are outliers that artificially increase the measured fidelity by being very far from other points. Here, clipping the real balls is enough to make the measured fidelity drop to 0.00.

# E FROM *Density* TO *Clipped Density*: OVER-OCCURRING SAMPLE PROPORTION AND RADII CLIPPING

This section measures the effects of the successive modifications that transform *Density* into *Clipped Density*. Below, we detail their impact on the evaluation of real generated datasets, along with the proportion of over-occurring samples (i.e., samples with a fidelity score greater than 1) at each step.

The columns, in order, present: the original *Density* score, the initial proportion of over-occurring samples (OOP), the *Density* score after radii clipping, the OOP after radii clipping, the scores after individual sample contributions are also clipped (resulting in ClippedDensity$_{unnorm}$), and finally, the fully normalized *ClippedDensity*.

Table 3: **From *Density* to *Clipped Density* on CIFAR-10.**

| Model | Density | Initial OOP (%) | Density (clipped radii) | OOP (clipped radii) (%) | ClippedDensity$_{unnorm}$ | ClippedDensity |
|---|---|---|---|---|---|---|
| LSGM-ODE | 0.66 | 18.4 | 0.22 | 6.0 | 0.14 | 0.38 |
| PFGMPP | 0.65 | 18.3 | 0.22 | 6.3 | 0.15 | 0.39 |
| iDDPM-DDIM | 0.64 | 17.4 | 0.19 | 5.6 | 0.13 | 0.34 |
| ACGAN-Mod | 2.28 | 43.0 | 0.00 | 0.0 | 0.00 | 0.01 |
| BigGAN | 0.55 | 14.4 | 0.10 | 2.6 | 0.07 | 0.19 |
| LOGAN | 1.01 | 22.4 | 0.00 | 0.0 | 0.00 | 0.01 |
| MHGAN | 0.57 | 14.9 | 0.06 | 1.6 | 0.05 | 0.13 |
| ReACGAN | 0.57 | 14.2 | 0.07 | 1.6 | 0.05 | 0.14 |
| StyleGAN-XL | 0.58 | 15.3 | 0.15 | 4.0 | 0.10 | 0.27 |
| StyleGAN2-ada | 0.60 | 16.4 | 0.13 | 3.4 | 0.09 | 0.24 |
| WGAN-GP | 1.74 | 37.2 | 0.00 | 0.0 | 0.00 | 0.01 |
| RESFLOW | 2.47 | 47.4 | 0.00 | 0.0 | 0.00 | 0.00 |
| NVAE | 1.59 | 31.9 | 0.00 | 0.0 | 0.00 | 0.00 |

Table 4: **From *Density* to *Clipped Density* on ImageNet.**

| Model | Density | Initial OOP (%) | Density (clipped radii) | OOP (clipped radii) (%) | ClippedDensity$_{unnorm}$ | ClippedDensity |
|---|---|---|---|---|---|---|
| ADM | 0.38 | 8.3 | 0.08 | 2.2 | 0.07 | 0.17 |
| ADMG | 0.49 | 12.8 | 0.17 | 4.7 | 0.13 | 0.34 |
| ADMG-ADMU | 0.55 | 15.5 | 0.21 | 5.8 | 0.16 | 0.41 |
| DiT-XL-2 | 0.83 | 28.2 | 0.43 | 14.6 | 0.29 | 0.74 |
| DiT-XL-2-guided | 1.65 | 63.8 | 1.12 | 42.3 | 0.63 | 1.00 |
| LDM | 0.70 | 21.5 | 0.32 | 10.0 | 0.22 | 0.57 |
| BigGAN | 0.46 | 10.5 | 0.05 | 0.9 | 0.05 | 0.12 |
| GigaGAN | 0.41 | 9.2 | 0.07 | 1.5 | 0.06 | 0.15 |
| StyleGAN-XL | 0.38 | 8.3 | 0.06 | 1.2 | 0.05 | 0.13 |
| Mask-GIT | 0.43 | 10.3 | 0.10 | 2.2 | 0.09 | 0.22 |
| RQ-Transformer | 0.34 | 6.8 | 0.04 | 0.8 | 0.03 | 0.09 |

Table 5: **From *Density* to *Clipped Density* on LSUN Bedroom.**

| Model | Density | Initial OOP (%) | Density (clipped radii) | OOP (clipped radii) (%) | ClippedDensity unnorm | ClippedDensity |
|---|---|---|---|---|---|---|
| Consistency-set1 | 0.62 | 18.7 | 0.14 | 3.4 | 0.11 | 0.32 |
| Consistency-set2 | 0.64 | 18.9 | 0.13 | 2.9 | 0.11 | 0.30 |
| ADM-dropout | 0.83 | 25.9 | 0.41 | 12.6 | 0.26 | 0.73 |
| DDPM | 0.55 | 14.5 | 0.16 | 4.6 | 0.11 | 0.32 |
| iDDPM | 0.64 | 18.2 | 0.23 | 6.8 | 0.16 | 0.44 |
| Diff-ProjGAN | 0.56 | 15.0 | 0.05 | 1.1 | 0.05 | 0.13 |
| Projected-GAN | 0.63 | 16.9 | 0.05 | 0.9 | 0.04 | 0.12 |
| StyleGAN | 0.74 | 21.8 | 0.26 | 7.4 | 0.17 | 0.48 |
| Unleash-Trans | 0.51 | 12.5 | 0.06 | 1.4 | 0.05 | 0.14 |

Table 6: **From *Density* to *Clipped Density* on FFHQ.**

| Model | Density | Initial OOP (%) | Density (clipped radii) | OOP (clipped radii) (%) | ClippedDensity unnorm | ClippedDensity |
|---|---|---|---|---|---|---|
| LDM | 0.65 | 18.4 | 0.24 | 6.6 | 0.16 | 0.41 |
| InsGen | 0.32 | 6.9 | 0.06 | 1.3 | 0.05 | 0.12 |
| Projected-GAN | 0.21 | 3.5 | 0.02 | 0.2 | 0.01 | 0.03 |
| StyleGAN-XL | 0.51 | 13.0 | 0.14 | 3.6 | 0.10 | 0.26 |
| StyleGAN2-ada | 0.24 | 4.2 | 0.04 | 0.7 | 0.03 | 0.08 |
| StyleNAT | 0.61 | 16.5 | 0.20 | 5.3 | 0.14 | 0.35 |
| StyleSwin | 0.53 | 13.8 | 0.13 | 3.1 | 0.10 | 0.25 |
| Unleash-Trans | 0.46 | 11.0 | 0.10 | 2.5 | 0.08 | 0.19 |
| Efficient-vdVAE | 0.80 | 23.2 | 0.28 | 7.9 | 0.17 | 0.42 |

## F  EVALUATION OF GENERATED CHEST X-RAYS

We evaluated a Progressively Growing GAN (Karras et al., 2018) from the Medigan library (Osuala et al., 2023), trained by Segal et al. (2021) on the ChestX-ray8 dataset (Wang et al., 2017) ($\approx 110000$ chest X-ray images). The model yielded a *Clipped Density* of $0.06$ and a *Clipped Coverage* of $0.03$.

The absolute interpretability of our metrics allows a direct assessment of these results without requiring comparison to other models. A *Clipped Density* of $0.06$ implies that the model's output has a fidelity equivalent to a dataset composed of only 6% good samples and 94% bad samples; similarly, a *Clipped Coverage* of $0.03$ is equivalent to a dataset with only 3% good samples.

This is a critical advantage over relative metrics, which can only rank models against each other without revealing whether even the best-performing one is adequate for a given task. In high-stakes domains like medical imaging, the ability to make an absolute judgment is essential. A low score provides a clear, unambiguous signal that the model is not yet fit for purpose, preventing the premature adoption of a technology that fails to meet the necessary standards for safety and reliability.

## G    COMPARISON TO HUMAN EVALUATIONS

To further validate *Clipped Density* and *Clipped Coverage*, we compared their scores against human error rates in distinguishing real from generated images (scores shared by Stein et al. (2023)), where higher error rates indicate better generative models.

We computed Pearson correlation coefficients between human error rates and both metrics across all datasets in Figure 8. All correlations were significant and exceeded $0.8$, except for FFHQ, where no significant correlation was found. This aligns with the findings of Stein et al. (2023) (Figure 4, DINOv2 column). This strong agreement with human judgment confirms that *Clipped Density* and *Clipped Coverage* correlate well with human perception of generation quality.

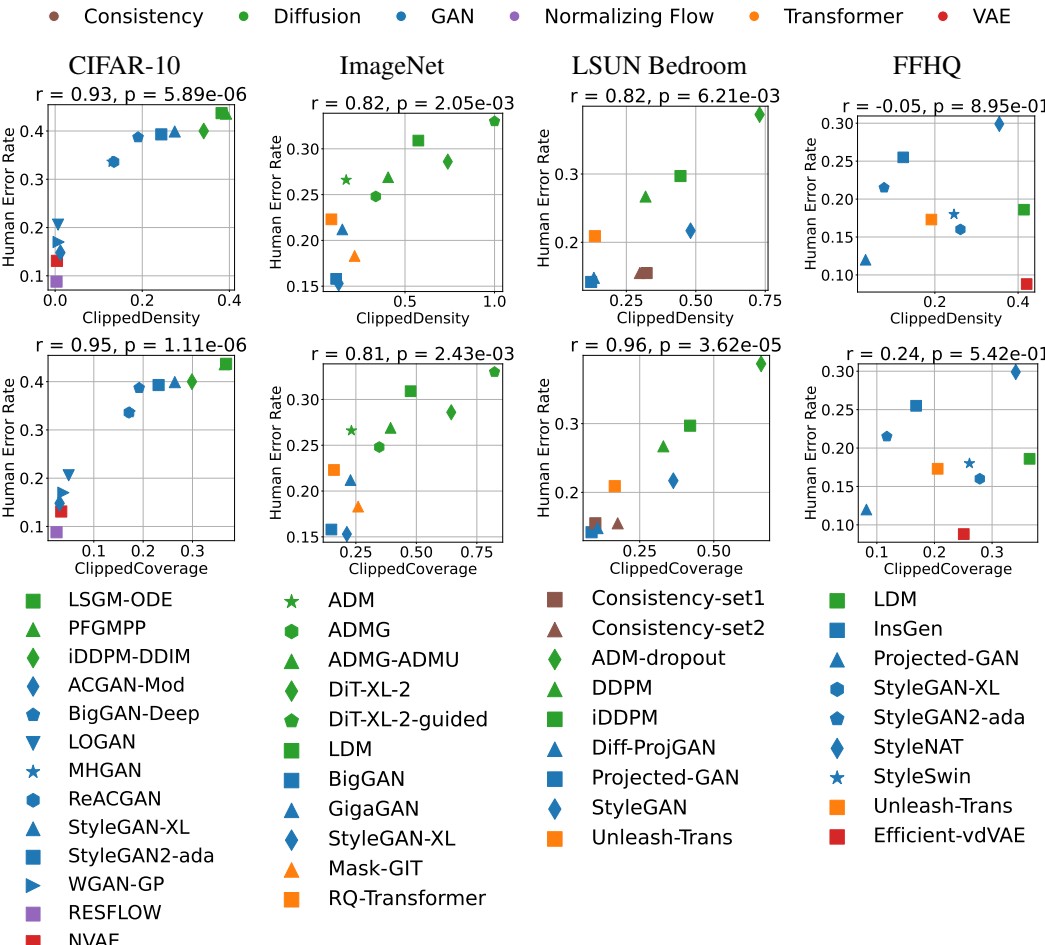

Figure 8: **Human Error Rate vs Clipped Density and Clipped Coverage**: We use human error rates from Stein et al. (2023), where subjects were asked to distinguish between real and generated images. A higher error rate indicates a better generative model. The first row shows human error rates vs. *Clipped Density*, and the second row vs. *Clipped Coverage*. Above each plot, we display the correlation coefficients (and p-values) between human error rates and each metric. All correlations are high except for the FFHQ dataset, where no significant correlation was found, consistent with Stein et al. (2023) (grayed in their Figure 4, DINOv2 column).

## H    COMPARISON TO A RECENT BENCHMARK

A recent position paper by Räisä et al. (2025) argues that "all current generative fidelity and diversity metrics are flawed" and that "no metric is suitable for absolute evaluations." Our work tackles these issues directly. In this section, we evaluate *Clipped Density* and *Clipped Coverage* on the benchmark from Räisä et al. (2025), using the authors' original code. All tests in this benchmark are performed on synthetic data, with a default sample size of 1000.

The results are summarized in Tables 7 and 8, which reproduce the tables from Räisä et al. (2025) with *Clipped Density* (Cl-Dens) and *Clipped Coverage* (Cl-Cov) added for comparison. Here, T denotes a passed test, while F indicates failure. The original benchmark distinguishes between high (H, support-based) and low (L, density-based) diversity metrics. Since we aim to measure coverage, which assesses densities, we interpret L as a success (T) and H as a failure (F).

We observed seven discrepancies with the original results, marked with a royal blue underline (T or F). These differences arise from the benchmark's seed generation: the seed is based on a hash of the metric name, test name, and repetition index, but Python's built-in `hash` function changes across different sessions unless `PYTHONHASHSEED` is set. As a result, seeds (and thus results) differ between runs.

Table 7: **Fidelity metrics comparison on a recent benchmark from Räisä et al. (2025)**

| Desiderata | Sanity Check | I-Prec | Density | IAP | C-Prec | symPrec | P-Prec | Cl-Dens |
|---|---|---|---|---|---|---|---|---|
| Purpose | Discrete Num. vs. Continuous Num. | F | F | F | F | F | F | F |
| | Gaussian Mean Difference | T | T | T | T | T | T | T |
| | Gaussian Mean Difference + Outlier | F | T | T | T | F | T | T |
| | Gaussian Mean Difference + Pareto | T | T | T | T | T | T | T |
| | Gaussian Std. Deviation Difference | T | T | F | F | F | T | T |
| | Hypercube, Varying Sample Size | F | F | F | F | F | F | F |
| | Hypercube, Varying Syn. Size | F | F | F | F | F | F | F |
| | Hypersphere Surface | F | F | T | F | T | F | F |
| | Mode Collapse | T | T | T | T | T | T | T |
| | Mode Dropping + Invention | T | T | F | F | F | T | T |
| | One Disjoint Dim. + Many Identical Dim. | F | F | F | F | F | F | F |
| | Sequential Mode Dropping | T | T | F | F | F | T | T |
| | Simultaneous Mode Dropping | T | F | F | F | F | T | T |
| | Sphere vs. Torus | T | T | T | F | T | T | T |
| Hyperparam. | Hypercube, Varying Syn. Size | T | T | T | F | F | T | T |
| Data | Hypercube, Varying Sample Size | F | F | F | F | F | F | F |
| Bounds | Discrete Num. vs. Continuous Num. | F | F | F | F | F | F | F |
| | Gaussian Mean Difference | F | T | F | T | F | F | T |
| | Gaussian Mean Difference + Outlier | F | F | F | T | F | F | F |
| | Gaussian Mean Difference + Pareto | F | T | T | T | T | F | T |
| | Gaussian Std. Deviation Difference | F | F | F | F | F | F | F |
| | Hypersphere Surface | F | F | F | F | T | F | F |
| | Mode Collapse | F | T | F | T | F | F | T |
| | Mode Dropping + Invention | T | T | F | F | F | F | T |
| | One Disjoint Dim. + Many Identical Dim. | F | F | T | F | F | F | F |
| | Scaling One Dimension | T | T | T | T | T | T | T |
| | Sequential Mode Dropping | F | T | F | F | F | F | T |
| | Simultaneous Mode Dropping | F | F | F | F | F | F | T |
| | Sphere vs. Torus | T | T | F | F | T | T | T |
| Invariance | Scaling One Dimension | F | T | T | T | T | T | T |

Table 8: **Coverage metrics comparison on a recent benchmark from Räisä et al. (2025)**

| Desiderata | Sanity Check | I-Rec | Coverage | IBR | C-Rec | symRec | P-Rec | Cl-Cov |
|---|---|---|---|---|---|---|---|---|
| Purpose | Discrete Num. vs. Continuous Num. | F | F | F | F | F | F | F |
| | Gaussian Mean Difference | T | T | T | T | T | T | T |
| | Gaussian Mean Difference + Outlier | F | T | T | T | T | T | T |
| | Gaussian Mean Difference + Pareto | T | T | T | T | T | T | T |
| | Gaussian Std. Deviation Difference | H | F | L | F | F | H | F |
| | Hypercube, Varying Sample Size | F | F | F | F | F | F | F |
| | Hypercube, Varying Syn. Size | F | F | F | F | F | F | F |
| | Hypersphere Surface | F | F | F | F | T | F | F |
| | Mode Collapse | F | F | F | F | F | F | L |
| | Mode Dropping + Invention | T | F | F | F | F | T | *F* |
| | One Disjoint Dim. + Many Identical Dim. | F | F | F | F | F | F | F |
| | Sequential Mode Dropping | T | T | F | F | T | T | T |
| | Simultaneous Mode Dropping | F | T | F | T | F | T | T |
| | Sphere vs. Torus | F | F | F | F | F | F | T |
| Hyperparam. | Hypercube, Varying Syn. Size | F | F | F | F | F | F | F |
| Data | Hypercube, Varying Sample Size | F | F | F | F | F | F | F |
| Bounds | Discrete Num. vs. Continuous Num. | F | F | F | F | F | F | F |
| | Gaussian Mean Difference | F | T | F | T | F | F | F |
| | Gaussian Mean Difference + Outlier | F | T | F | T | F | F | T |
| | Gaussian Mean Difference + Pareto | F | T | F | T | T | F | T |
| | Gaussian Std. Deviation Difference | F | F | F | F | F | F | F |
| | Hypersphere Surface | F | F | F | F | T | F | F |
| | Mode Collapse | F | T | F | T | F | F | F |
| | Mode Dropping + Invention | T | F | F | F | F | F | *F* |
| | One Disjoint Dim. + Many Identical Dim. | F | F | T | F | F | F | F |
| | Scaling One Dimension | F | T | T | T | T | T | T |
| | Sequential Mode Dropping | F | T | F | T | F | F | T |
| | Simultaneous Mode Dropping | F | T | F | T | F | F | T |
| | Sphere vs. Torus | T | F | F | T | F | T | T |
| Invariance | Scaling One Dimension | F | T | T | T | T | T | T |

Purple underlined entries (F) indicate cases where the benchmark's criterion for success is, in our view, overly strict (see Appendix H.1). Italic blue entries (*F*) indicate cases where the original implementation counts a failure, but we argue that for low diversity metrics, these should be considered successes (see Appendix H.2).

## H.1 HARSH CRITERIA

Purple underlined entries (F) correspond to cases where, in our view, the benchmark's threshold for passing may be too narrow.

In the "Gaussian Mean Difference" test (Figure 9), which is analogous to our synthetic Gaussian translation test but without bad samples, the bounds desideratum is not satisfied by *Clipped Coverage* in dimension 64 with no translation (0). In this case, the initial and synthetic distributions are the same. *Clipped Coverage* has a value of 0.947, while the criterion requires a value between 0.95 and 1.05.

Similarly, in the "Mode Collapse" test (Figure 10), where real data is a mixture of two Gaussians spaced by $\mu$ and the synthetic data is a single Gaussian, the value in dimension 64 with no translation (0, same initial and synthetic distributions) is 0.941, just below the same required threshold.

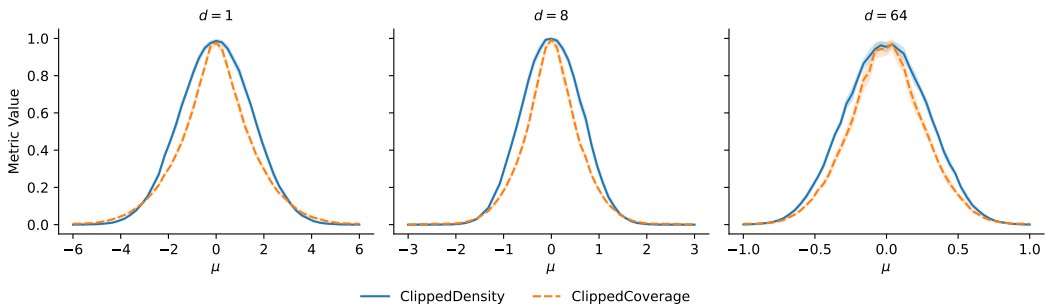

Figure 9: **Gaussian Mean Difference**: output from the original code for our metrics.

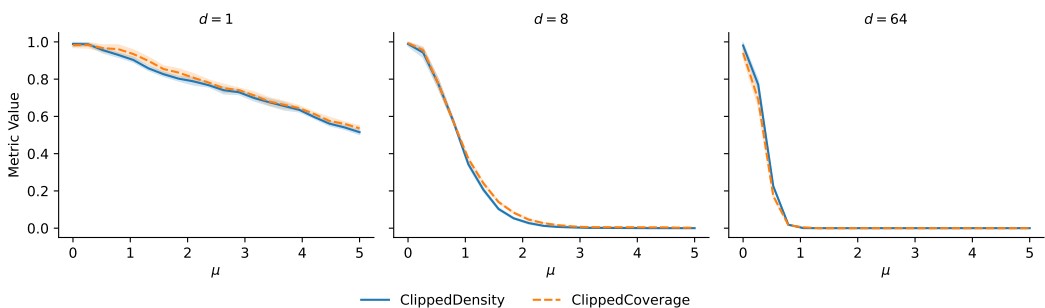

Figure 10: **Mode Collapse**: output from the original code for our metrics.

## H.2 INCORRECT CRITERION

Italic blue entries (*F*) indicate cases where we disagree with the benchmark's failure criterion.

In the "Mode Dropping + Invention" test (Figure 11), the real data is a mixture of 5 Gaussians, and the synthetic data progressively includes the 5 real modes and then 5 invented modes. Both real and synthetic sets always have 1000 samples.

*Clipped Coverage* is marked as failing because it decreases when invented modes are added. However, this decrease is expected for low-diversity metrics (density-based): when invented modes are included, real modes become under-covered as the total amount of synthetic data remains the same, so coverage metrics should decrease. Thus, this behavior should be considered T or L, both successes for coverage metrics.

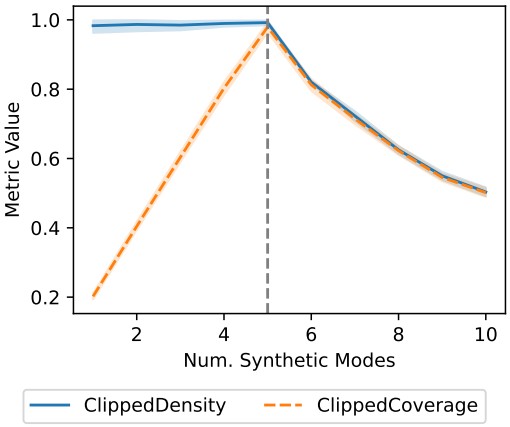

Figure 11: **Mode Dropping + Invention**: output from the original code for our metrics.

# I FIDELITY VS COVERAGE WITH OTHER METRICS

We reproduce Figure 6 with other metrics in Figures 12 and 13.

We observe that for some metric pairs, fidelity scores vary little, while coverage metrics vary significantly. This occurs for *Precision/Recall* (CIFAR-10, LSUN Bedroom), *TopP/TopR* (ImageNet, FFHQ), *P-precision/P-recall* (FFHQ), and *Precision Cover/Recall Cover* (FFHQ). *symPrecision* and *symRecall* show correlated scores. $\alpha$-*Precision* and $\beta$-*Recall* show distinct results for consistency models on LSUN Bedroom and yield scores generally confined to the lower-right quadrant.

However, the correct behavior of metrics on real data is unknown, so we cannot draw definitive conclusions about which metric pairs are better or worse from these plots alone.

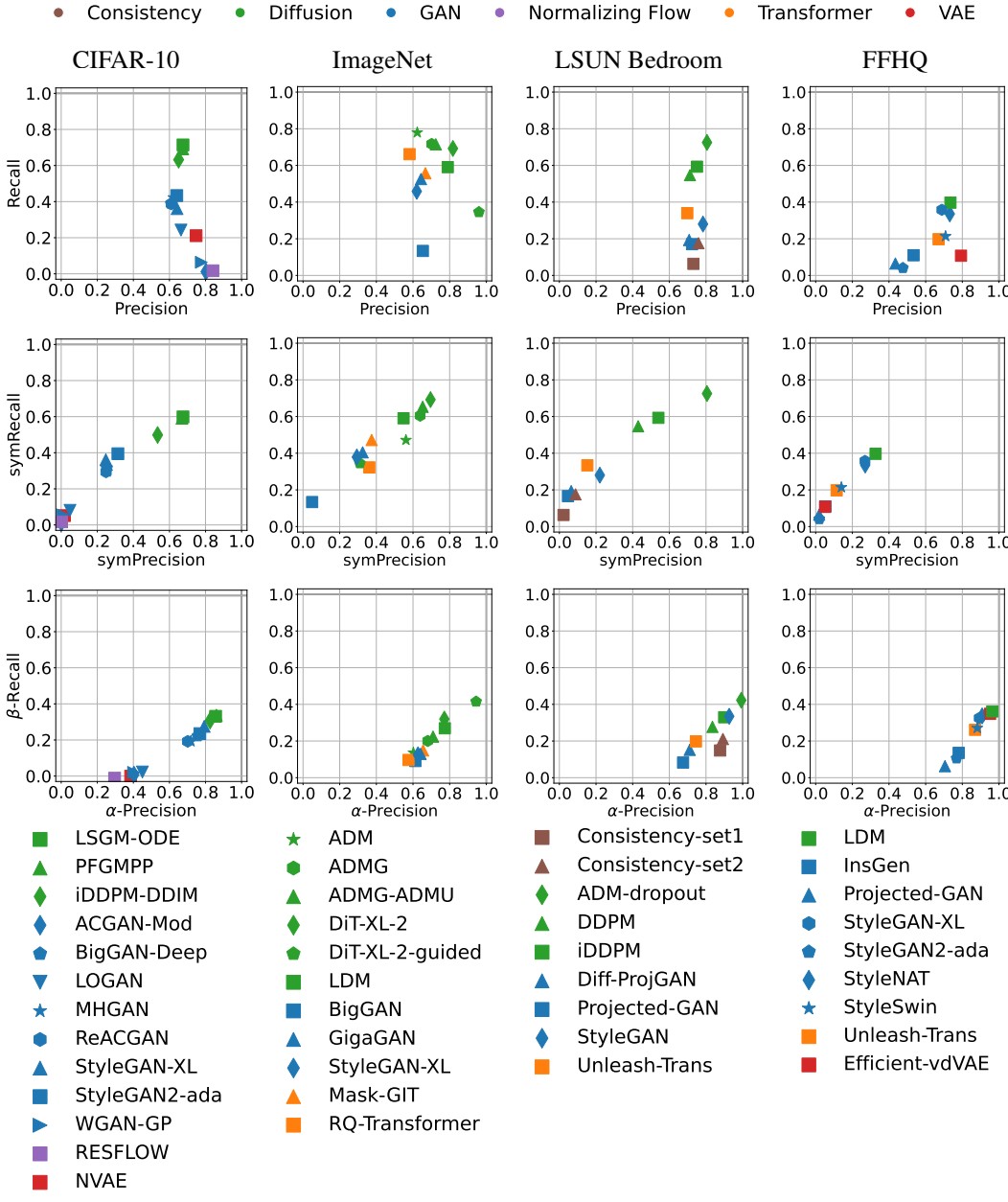

Figure 12: **Fidelity vs Coverage on various datasets, other metrics (1/2)**

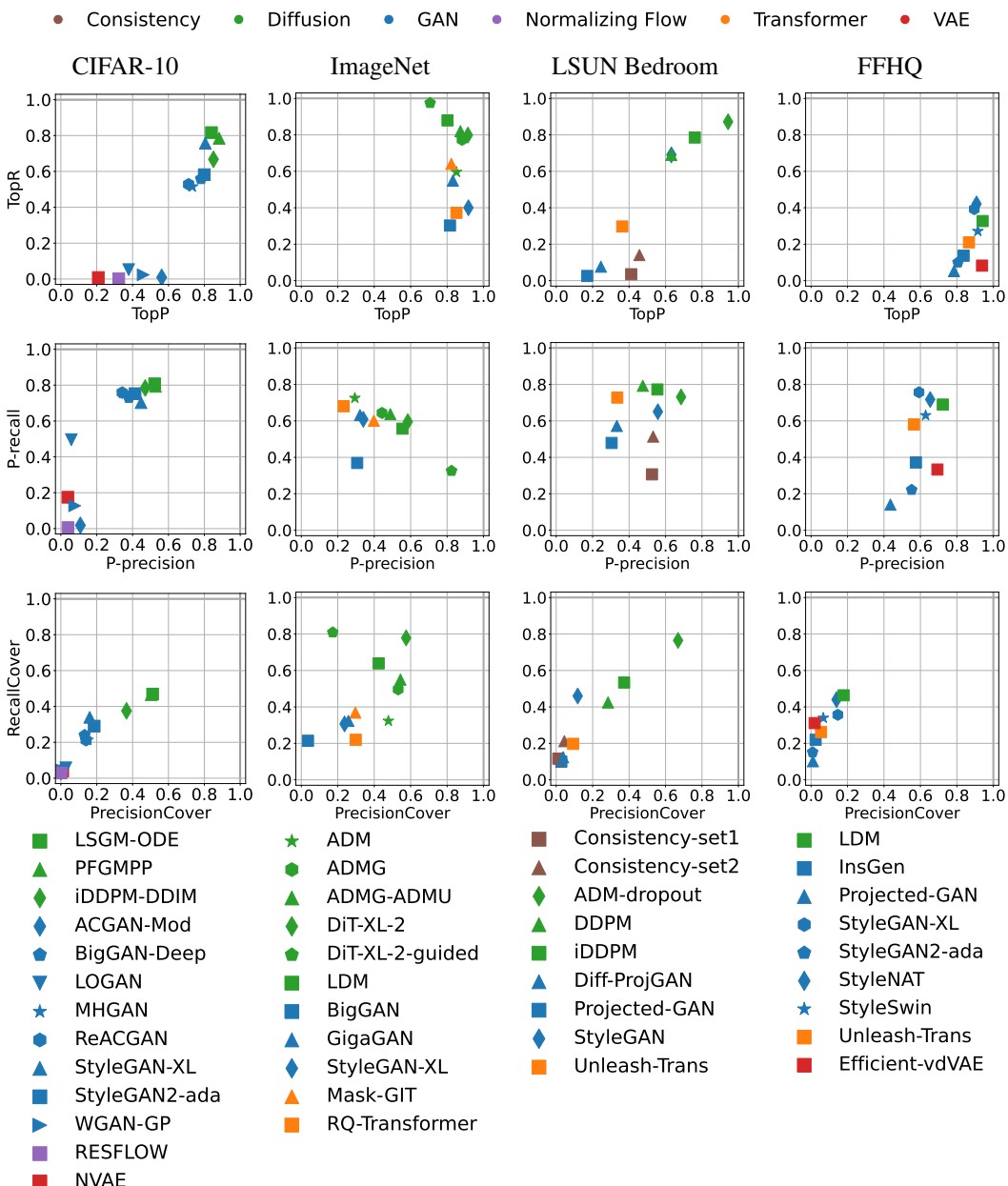

Figure 13: **Fidelity vs Coverage on various datasets, other metrics (2/2)**

## J  FIDELITY AND COVERAGE TRADE-OFF: TRUNCATION IN GANS

There is often a trade-off between fidelity and coverage, as improving one can come at the expense of the other for a given model (Li et al., 2024). In this section, we verify that *Clipped Density* and *Clipped Coverage* capture this trade-off by evaluating them on LSUN Bedroom StyleGAN data (Karras et al., 2019), shared by the original authors with varying truncation levels. For GANs, the truncation trick samples from a truncated latent space to improve fidelity at the expense of coverage. A smaller truncation value $\psi$ restricts sampling to a smaller, denser region. Note that $\psi = 1.0$ corresponds to no truncation, matching the StyleGAN model in Figure 6.

As shown in Figure 14, both scores initially increase as we move from no truncation ($\psi = 1.0$) to moderate truncation ($\psi = 0.7$). However, further truncation to $\psi = 0.5$ causes *Clipped Density* to rise higher while *Clipped Coverage* decreases, effectively illustrating the trade-off.

We investigated the initial rise of *Clipped Coverage* in Figure 15 by examining sample-wise score histograms. From $\psi = 1.0$ to $\psi = 0.7$, real samples in high-density regions that were barely covered become fully covered. Then, at $\psi = 0.5$, many real samples drop to $0$ coverage as parts of the distribution are ignored. Thus, the initial rise in the *Clipped Coverage* score is caused by a more intense coverage of dense real regions.

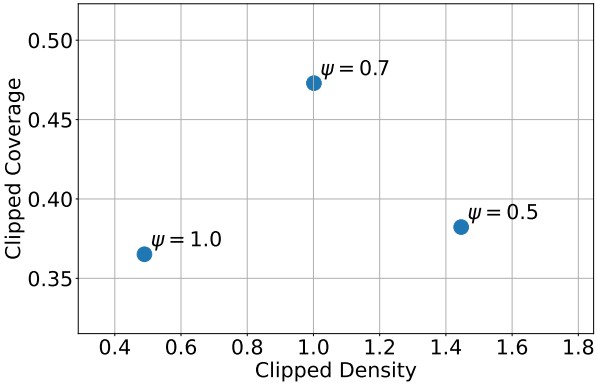

Figure 14: **Fidelity and Coverage trade-off**: To verify that *Clipped Density* and *Clipped Coverage* capture the fidelity-coverage trade-off, we evaluated them on LSUN Bedroom StyleGAN data (Karras et al., 2019) shared by the original authors with varying truncation levels. For GANs, the truncation trick samples from a truncated latent space to improve fidelity at the expense of coverage. A smaller truncation value $\psi$ restricts sampling to a smaller, denser region. $\psi = 1.0$ corresponds to no truncation, matching the StyleGAN model in Figure 6. From no truncation ($\psi = 1.0$) to moderate truncation ($\psi = 0.7$), both scores increase. However, further truncation to $\psi = 0.5$ causes *Clipped Density* to rise higher while *Clipped Coverage* decreases, illustrating the trade-off. Note that here, we display *Clipped Density* scores without clipping them to 1, see the next subsection for details.

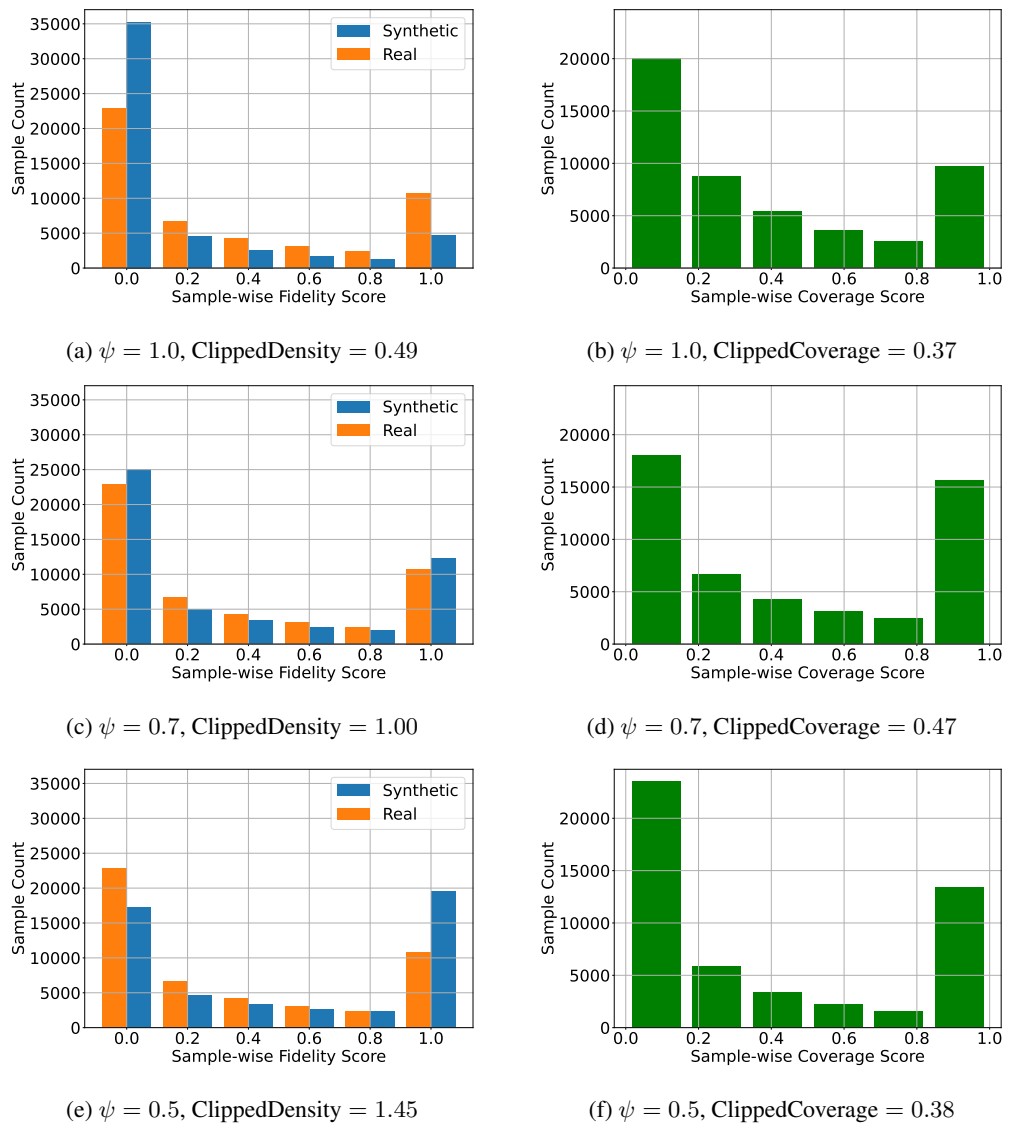

(a) $\psi = 1.0$, ClippedDensity $= 0.49$

(b) $\psi = 1.0$, ClippedCoverage $= 0.37$

(c) $\psi = 0.7$, ClippedDensity $= 1.00$

(d) $\psi = 0.7$, ClippedCoverage $= 0.47$

(e) $\psi = 0.5$, ClippedDensity $= 1.45$

(f) $\psi = 0.5$, ClippedCoverage $= 0.38$

Figure 15: **Sample-wise Fidelity and Coverage for different truncation levels**: Sample-wise *Clipped Density* (left) and *Clipped Coverage* (right) histograms for the StyleGAN data evaluated in Figure 14, with varying truncation $\psi$. As truncation is strengthened, fewer synthetic samples have 0-fidelity and more reach 1, reflecting sampling restricted to dense regions. From $\psi = 1.0$ to $\psi = 0.7$, *Clipped Coverage* improves as high-density regions become better covered. However, at $\psi = 0.5$, many real samples drop to 0 coverage as parts of the distribution are ignored.

# K *Clipped Density* SCORES EXCEEDING 1

In this section, we investigate the impact of clipping *Clipped Density* scores to 1.

First, we reproduced previous tests where *Clipped Density* was limited to 1, but without this clipping (Figure 16). We found that the results remained stable near 1 as expected: in these test settings, *Clipped Density* does not exceed 1 significantly.

In our evaluations of generative models, one instance can exceed 1: DiT-XL-2 with guidance on ImageNet. The sample-wise histograms (Figure 17) show that guidance increases the proportion of synthetic samples with perfect fidelity, surpassing that of real samples. Similar to StyleGAN truncation, which can also yield scores above 1 (see Appendix J), *Clipped Coverage* improves as covered points become fully covered while uncovered points remain stable. This reflects the similar effects of guidance and truncation: both alter sampling to favor high-density regions while ignoring low-density ones.

A *Clipped Density* score above 1 thus occurs when sampling systematically avoids low-density regions, effectively targeting a filtered real distribution. This is a feature of the unclipped metric: while using a filtered high-quality test set might be better practice in this scenario, the metric correctly reflects that the synthetic set achieves higher fidelity than the unfiltered real reference.

Scores above 1 indicate that the generative model does not target the full real distribution, but rather a subset of high-density regions. Since our goal is to evaluate how well the model matches the real set, scores exceeding 1 represent a deviation from this objective (often at the cost of coverage). Clipping to 1 enforces the interpretation that ideal fidelity consists of matching the real data exactly, rather than surpassing it by ignoring difficult regions.

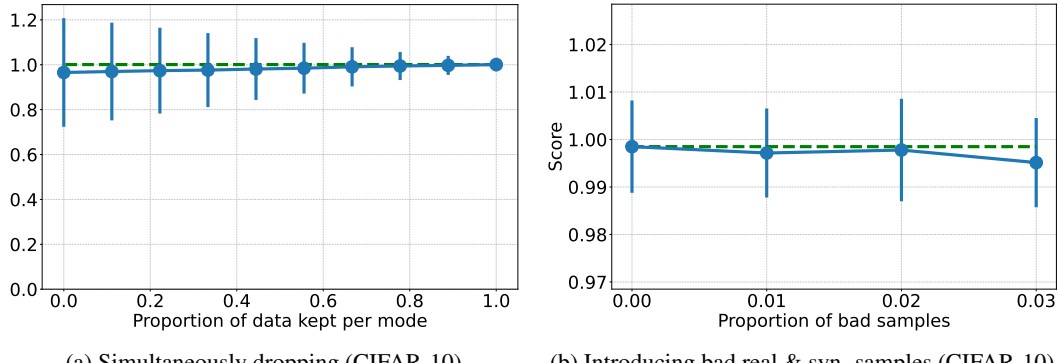

(a) Simultaneously dropping (CIFAR-10)   (b) Introducing bad real & syn. samples (CIFAR-10)

Figure 16: **Reproduced tests without limiting *Clipped Density* to 1**. We reproduced previous tests where *Clipped Density* was limited to 1. The results remain stable near 1 as expected. The standard deviation is high in (a) because only 2500 samples are used (see Figure 20 for an analysis of the standard deviation vs sample size). In test settings, *Clipped Density* does not exceed 1 significantly.

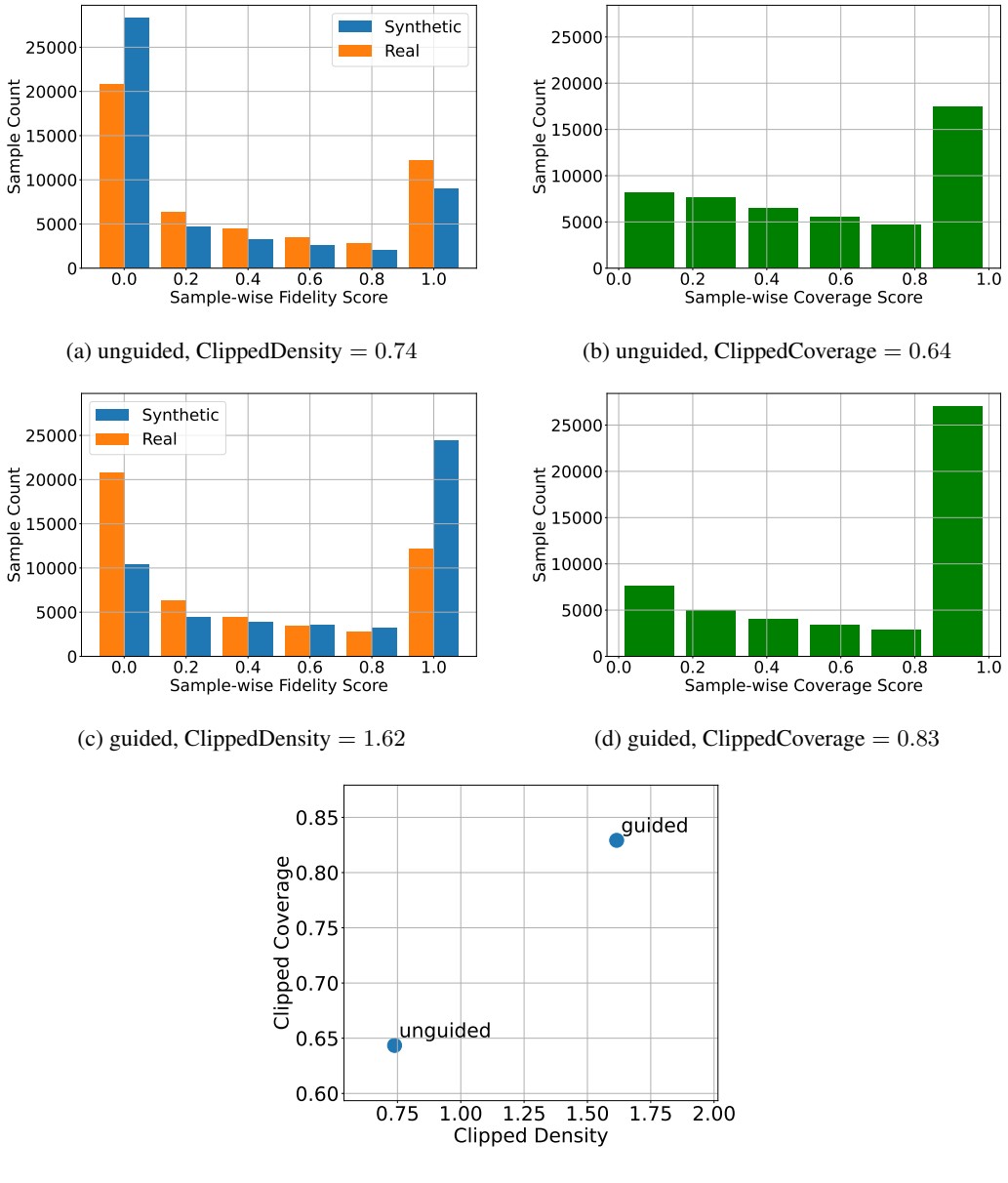

(a) unguided, ClippedDensity = 0.74

(b) unguided, ClippedCoverage = 0.64

(c) guided, ClippedDensity = 1.62

(d) guided, ClippedCoverage = 0.83

(e) Clipped Density vs Clipped Coverage

Figure 17: *Clipped Density* **exceeding 1 with Guidance on ImageNet**. With guidance on DiT-XL-2 on ImageNet, *Clipped Density* significantly exceeds 1. Sample-wise histograms show that guidance increases the proportion of synthetic samples with perfect fidelity, surpassing that of real samples. Similar to StyleGAN truncation, *Clipped Coverage* improves as covered points become fully covered while uncovered points remain stable, reflecting the similar effects of guidance and truncation: sampling favors high-density regions and ignores low-density ones. A *Clipped Density* above 1 occurs when sampling avoids low-density regions, effectively targeting a filtered real distribution. While using a filtered high-quality test set might be better practice, the metric correctly reflects that the synthetic set achieves higher quality than the unfiltered real reference.

# L    SENSITIVITY TO THE HYPERPARAMETER $k$

The number of nearest neighbors, $k$, is a hyperparameter for *Clipped Density*, *Clipped Coverage*, and several other metrics. We evaluate their sensitivity to $k$ on the DINOv2-FFHQ-LDM dataset (see the Figure below).

Scores increase with $k$, as this results in larger $k$-nearest neighbor balls and thus a higher likelihood of sample inclusion within these balls. *Clipped Density* and *Clipped Coverage* remain relatively stable, likely due to the normalization of the balls by their volume or mass.

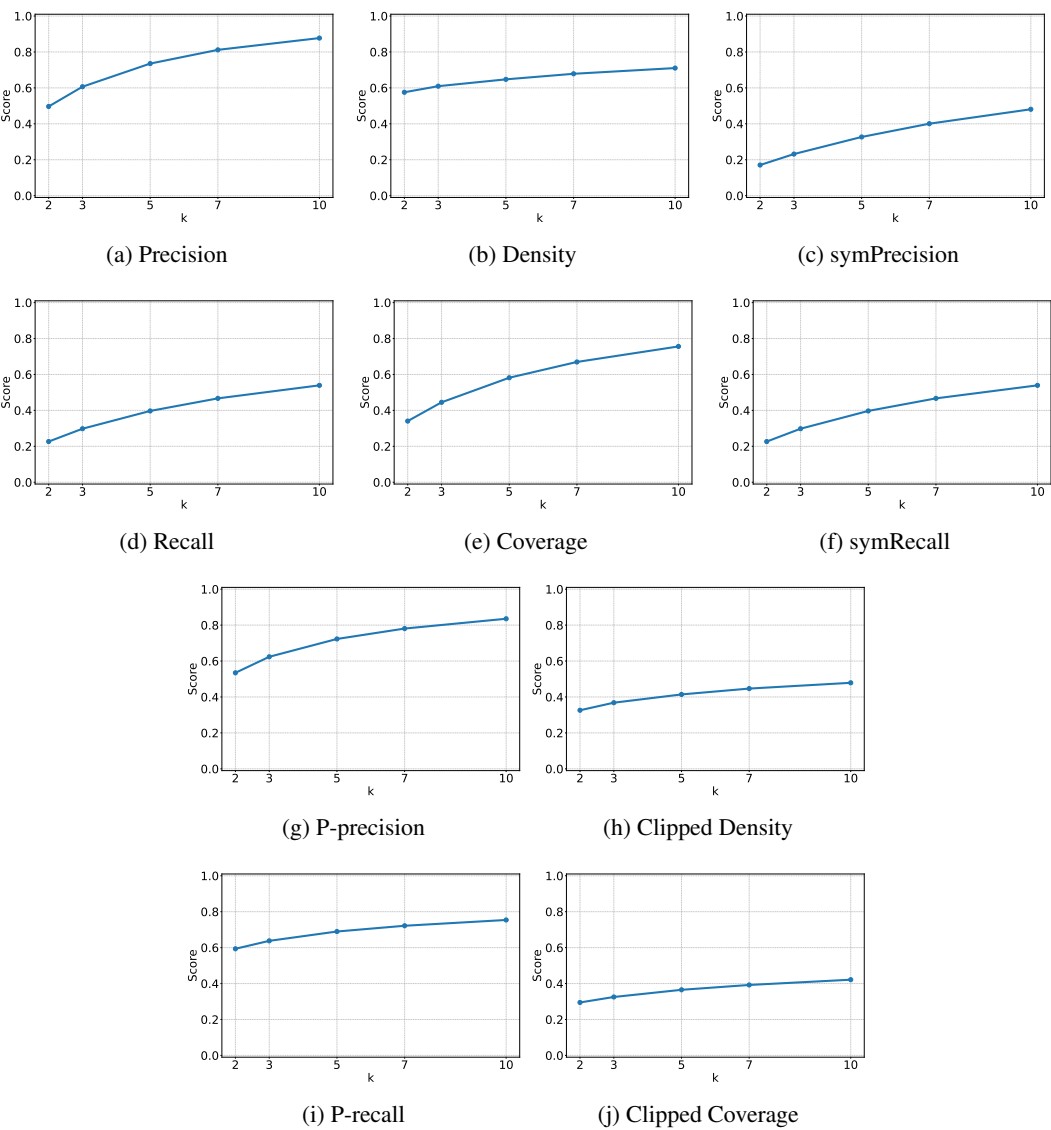

Figure 18: **Sensitivity to $k$ on DINOv2-FFHQ-LDM.**

## M  CALIBRATION STABILITY

To further validate the absolute calibration of *Clipped Density* and *Clipped Coverage*, we performed two additional experiments on the ImageNet dataset. First, we repeated the progressive bad sample introduction test 10 times using distinct ImageNet subsets (Figure 19), confirming that both metrics decrease linearly with the proportion of bad samples.

Second, we assessed the stability of the metric scores as a function of the sample size $N$ (Figure 20). We found that the empirical standard deviation of both metrics decreases proportionally to $1/\sqrt{N}$, reaching values below $0.01$ for $N = 50000$, thereby confirming their stability.

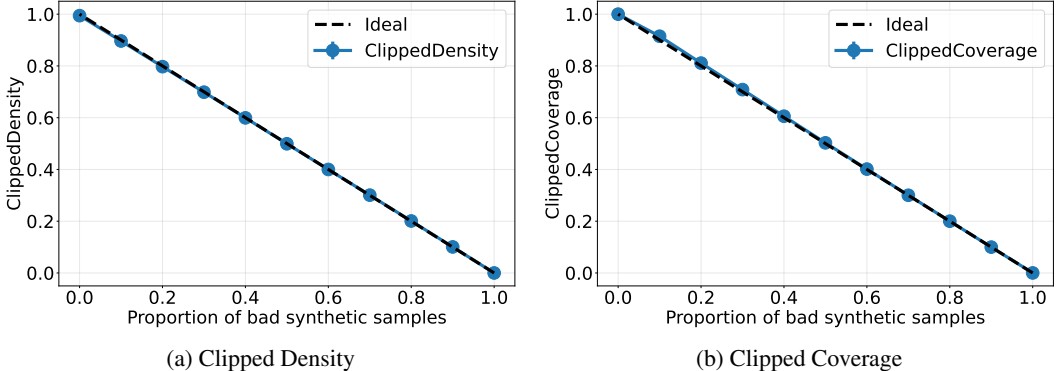

(a) Clipped Density          (b) Clipped Coverage

Figure 19: **Evaluating a mixture of good and bad samples (ImageNet)**: Synthetic sets mix DI-NOv2 embeddings of real ImageNet samples with noise (bad samples), using $N = 50000$ balanced samples per set. The experiment is repeated 10 times with different subsets of ImageNet. The standard deviation is shown but is smaller than the markers in all cases (at most $0.0056$ for *Clipped Density* and $0.0082$ for *Clipped Coverage*). Both metrics decrease linearly with the proportion of bad samples, confirming the absolute calibration on the high-dimensional ImageNet dataset.

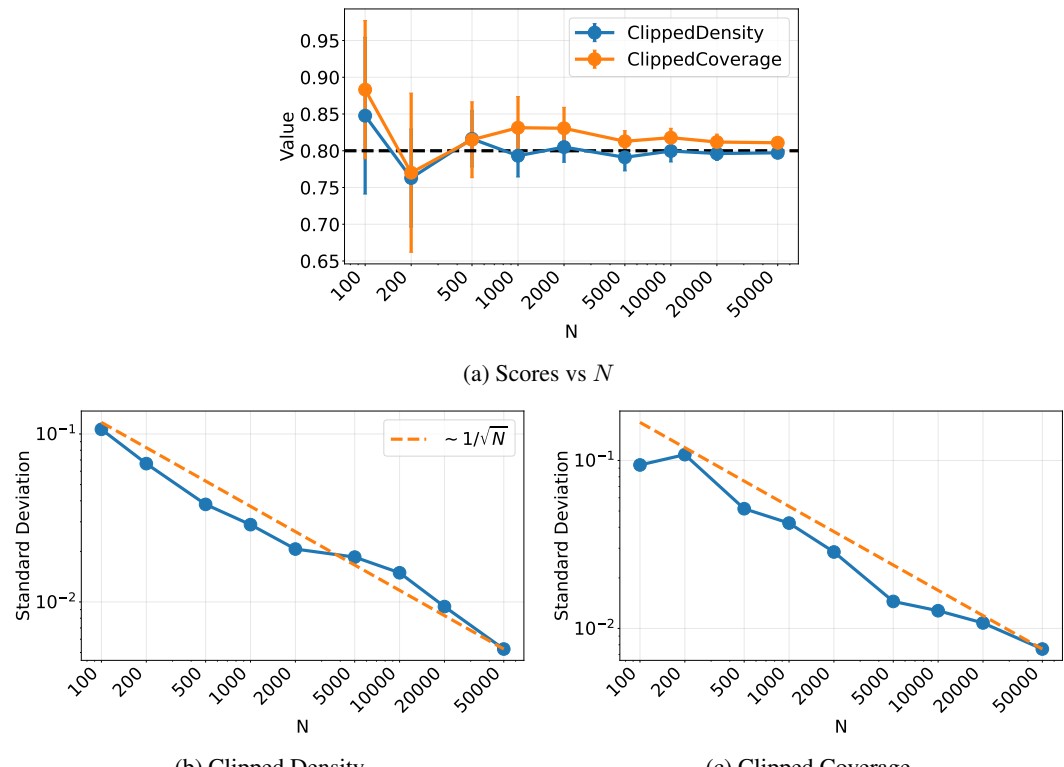

(a) Scores vs $N$

(b) Clipped Density

(c) Clipped Coverage

Figure 20: **Calibration Stability vs sample size** ($N = M$). Synthetic sets mix 80% real ImageNet samples with 20% noise, targetting a score of 0.8 to avoid boundary effects at 1. We use 50 samples per class, selecting random classes for smaller $N$ (10 repetitions). (a) Scores vs $N$. (b)-(c) Empirical standard deviation vs $N$. Both metrics' std decreases as $1/\sqrt{N}$, reaching values below 0.01 for $N = 50000$, even when adjusting by $1/0.8$ to account for higher target scores.

# N    FIDELITY UNDER IMPUTED DISTORTIONS

To further assess the behavior of *Clipped Density*, we evaluated the fidelity of PFGMPP-generated CIFAR-10 samples under various image distortions, replicating the setup from Figure 5 of Jiralerspong et al. (2023). Our results for *Clipped Density*, shown in Figure 21, align with those reported for the Feature Likelihood Divergence (FLD) metric in the original study, which were themselves better than those of FID.

We applied the transformations detailed in Appendix E.1 of Jiralerspong et al. (2023). For the Color Distort transformation, the default parameters produce an identity transform, so we used a non-default `hue = 0.3`. For the Center Crop transformation, two different values (28 and 30) are used throughout the paper. We proceeded with the value 28.

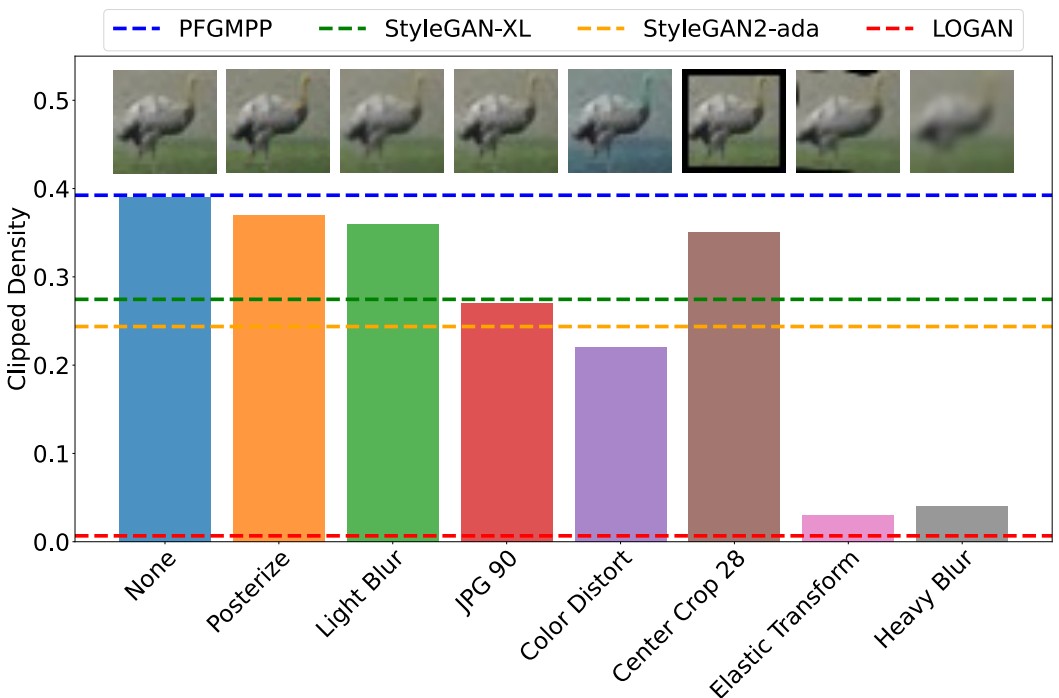

Figure 21: **Fidelity of PFGMPP samples under different image distortions, measured with** *Clipped Density***.** Horizontal lines represent the *Clipped Density* scores of other models for comparison.

## O    OTHER FAILURE MODES

In this section, we examine additional failure modes of generative models and analyze the response of *Clipped Density* and *Clipped Coverage*.

In the first scenario, we assess the detection of a small real mode absent from the synthetic data, following the protocol in Appendix F.2 of Lemos et al. (2025) (see Figure 22). The primary distribution consists of eight CIFAR-10 classes, while a rare mode (trucks) is progressively introduced solely into the real set. *Clipped Coverage* correctly decreases as this uncovered rare mode grows, showing sensitivity even when the mode represents only $0.00125\%$ of the real set. Conversely, *Clipped Density* remains stable, as the synthetic samples continue to reside within valid real modes.

In the second scenario, we evaluate the impact of incorrect intra-mode structure. We sample 10000 points from centered Gaussians with distinct orientations: /-shaped for the real data and \-shaped for the synthetic data (see Figure 23, top). These distributions correspond to the block-structured covariance matrices detailed in Figure 23 (d), where $\sigma$ controls the width of the shapes. Regardless of $\sigma$, both *Clipped Density* and *Clipped Coverage* correctly drop to 0 as the dimensionality increases, reflecting the fact that the intersection volume between the two distributions becomes negligible.

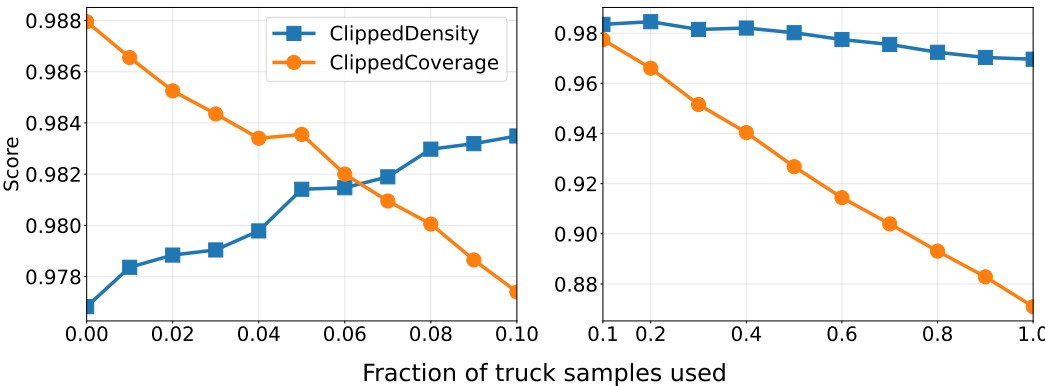

Figure 22: **Rare modes in real data**: Following Lemos et al. (2025) (Appendix F.2), we assess the detection of a small real mode absent from synthetic data. Eight CIFAR-10 classes form the main modes (all but trucks and ships, 20000 samples each for real/synthetic). A rare mode (trucks) is progressively introduced into the real set only (up to 2500 samples). We introduce 0% to 0.1% (Left) and 0.1% to 1% (Right) of truck samples, representing up to 0.0125% and 0.125% of the total real set, respectively. *Clipped Coverage* correctly decreases as the uncovered rare mode grows, while *Clipped Density* remains stable since synthetic samples still lie within valid real modes.

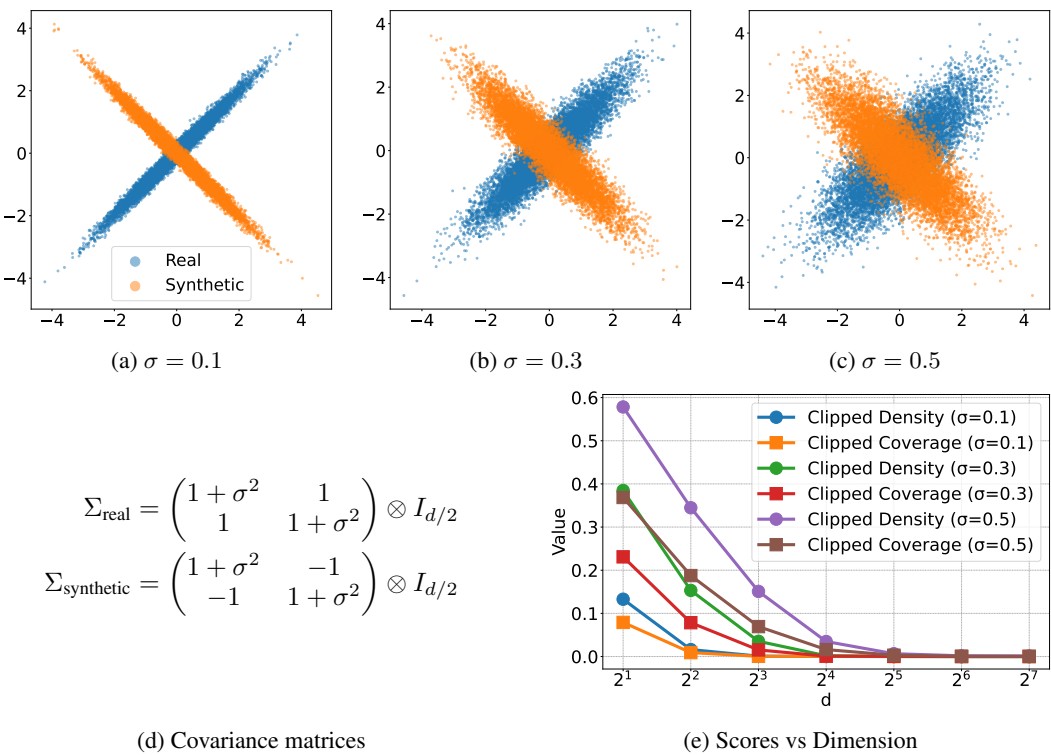

(a) $\sigma = 0.1$      (b) $\sigma = 0.3$      (c) $\sigma = 0.5$

$$\Sigma_{\text{real}} = \begin{pmatrix} 1+\sigma^2 & 1 \\ 1 & 1+\sigma^2 \end{pmatrix} \otimes I_{d/2}$$

$$\Sigma_{\text{synthetic}} = \begin{pmatrix} 1+\sigma^2 & -1 \\ -1 & 1+\sigma^2 \end{pmatrix} \otimes I_{d/2}$$

(d) Covariance matrices            (e) Scores vs Dimension

Figure 23: **Incorrect correlations**: Real and synthetic sets consist of $N = 10000$ samples from centered Gaussians with covariance matrices shown in (d). The first two components are shown for varying $\sigma$ in (a)-(c). (e) shows *Clipped Density* and *Clipped Coverage* versus dimension for different $\sigma$. Regardless of $\sigma$, both metrics drop to 0 as the dimension increases, because the intersection volume between the distributions becomes negligible.

# P OTHER MODALITIES

## P.1 TOY TIME SERIES EXAMPLE

Following Appendix F.3 of Lemos et al. (2025), we evaluated *Clipped Density* and *Clipped Coverage* using toy time series data. In this setup, the real signal consists solely of noise, whereas the synthetic data includes an added hallucinated signal of varying amplitude. As the amplitude of this hallucination increases, both metrics decrease, correctly identifying the discrepancy.

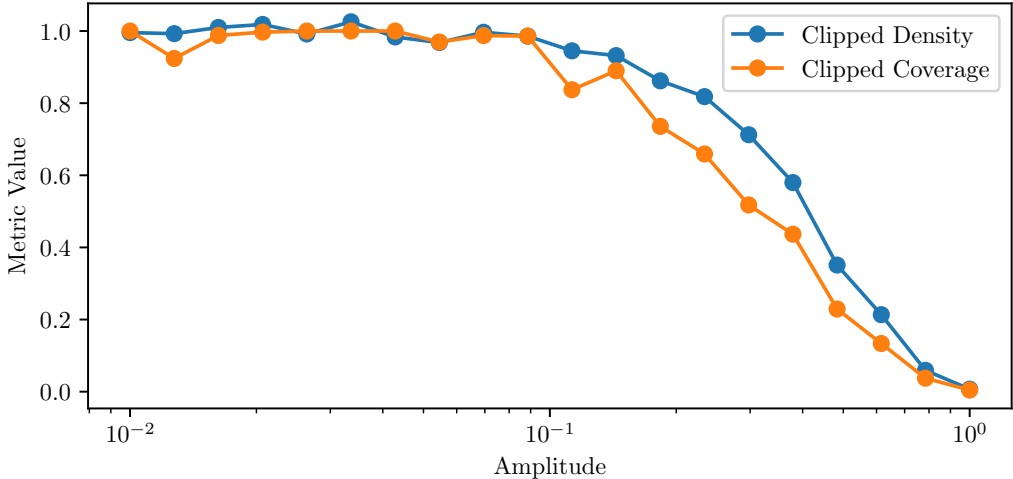

Figure 24: **Toy time series example**: Following Lemos et al. (2025) (Appendix F.3), we evaluate *Clipped Density* and *Clipped Coverage* on toy time series. The signal is defined as $A\cos(t) + \eta(t)$, where $\eta(t)$ is unit Gaussian noise, with 100 time points per sample and 5000 samples per set. The real set has zero amplitude ($A = 0$), while the synthetic set has varying amplitude. The goal is to detect this discrepancy. Both metrics decrease as the signal amplitude increases, dropping below the maximum slightly after an amplitude of 0.1. Some instability is observed due to the limited sample size of 5000.

## P.2 MUSIC DATASET EXAMPLE

To evaluate music data, we used the Free Music Archive large dataset, comprising 30s clips from 106574 tracks (Defferrard et al., 2017). Following Gui et al. (2024), who compared 10 embedding models for music, we employed the CLAP model (Elizalde et al., 2023). Since it extracts 1024-dimensional embeddings from 7s clips, we selected the middle 7s of each track.

An initial evaluation on balanced real and synthetic splits of 50000 samples each yielded a *Clipped Density* of 0.988 and a *Clipped Coverage* of 0.998. When splitting by genre, scores decreased to 0.943 for *Clipped Density* and 0.875 for *Clipped Coverage*. This reduction is moderate but expected, as music tracks span multiple genres, yet we split based on a single genre per track (a track's other associated genres might correspond to the other dataset).

Finally, we evaluated mixtures of good and bad samples, where bad samples consisted of noise MP3 files. The results show that both *Clipped Density* and *Clipped Coverage* decrease linearly with the proportion of bad samples, confirming their absolute calibration on this high-dimensional music dataset.

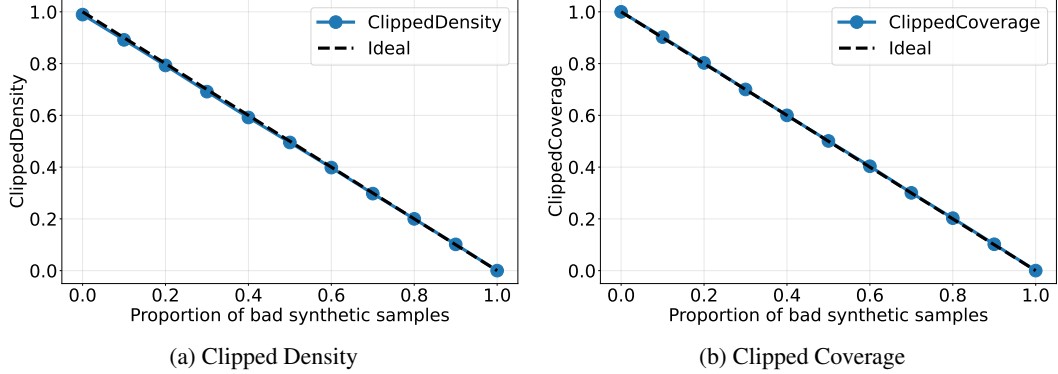

(a) Clipped Density      (b) Clipped Coverage

Figure 25: **Evaluating a mixture of good and bad samples (Free Music Archive)**: Synthetic sets mix CLAP embeddings of real music samples with embeddings of noise (bad samples), using $N = 50000$ balanced samples per set. Both *Clipped Density* and *Clipped Coverage* decrease linearly with the proportion of bad samples, confirming the absolute calibration on this high-dimensional music dataset. Note that compared to other plots of this test, the evaluation was performed only once per point and not repeated 10 times.

## Q   METRIC TESTS ON TOY DATASETS

This section details experiments on synthetic Gaussian data ($N = 25000, d = 32$, 10 repetitions), analogous to the CIFAR-10 tests in Section 5. Out-of-distribution samples are drawn from a Gaussian distribution with variance $\max(4, (10 + Z)^2)$, where $Z \sim \mathcal{N}(0, 1)$. The tests performed are:

- **Simultaneous mode dropping**: Progressively replacing data from all but one class with data from the remaining class (see Figure 26b, analogous to Figure 4b).
- **Matched real and synthetic out-of-distribution samples**: Progressively replacing data from both real and synthetic datasets with out-of-distribution samples (see Figures 26c and 27b, analogous to Figures 4c and 5a).
- **Introducing bad synthetic samples**: Progressively replacing synthetic data with out-of-distribution samples (see Figures 26a and 27a, analogous to Figures 1 and 4a).

### Q.1   FIDELITY METRICS

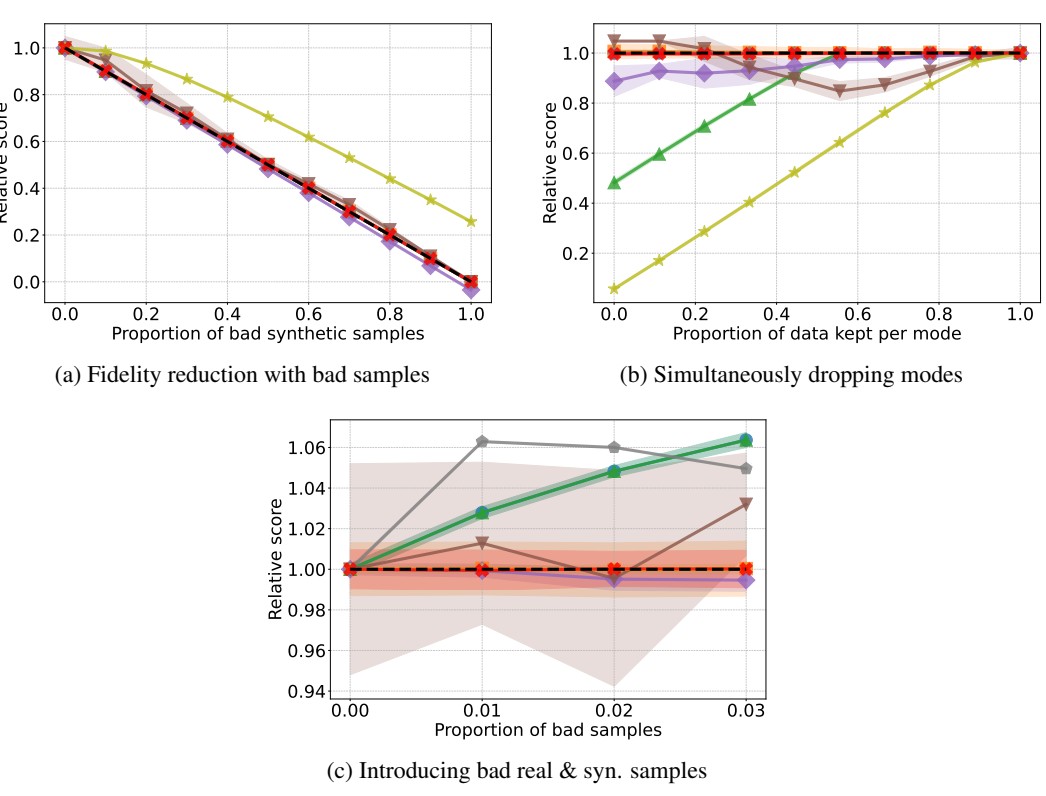

(a) Fidelity reduction with bad samples

(b) Simultaneously dropping modes

(c) Introducing bad real & syn. samples

| Legend | | Figure 26a | Figure 26b | Figure 26c | Figure 4d |
|---|---|:---:|:---:|:---:|:---:|
| ● | Precision | ✓ | ✓ | ✗ | ✗ |
| ■ | Density | ✓ | ✓ | ✓ | ✗ |
| ▲ | symPrecision | ✓ | ✗ | ✗ | ✗ |
| ◆ | $\alpha$-Precision | ✓ | ✗ | ✗ | ✗ |
| ▼ | TopP | ✓ | ✗ | ✗ | ✓ |
| ⬟ | P-precision | ✓ | ✓ | ✗ | ✗ |
| ★ | PrecisionCover | ✗ | ✗ | ✓ | ✓ |
| ✖ | ClippedDensity (ours) | ✓ | ✓ | ✓ | ✓ |
| – – | Ideal | | | | |

(d) Legend and summary

Figure 26: **Testing fidelity metrics on toy data**: this figure is the equivalent of Figure 4 on toy data.

## Q.2 COVERAGE METRICS

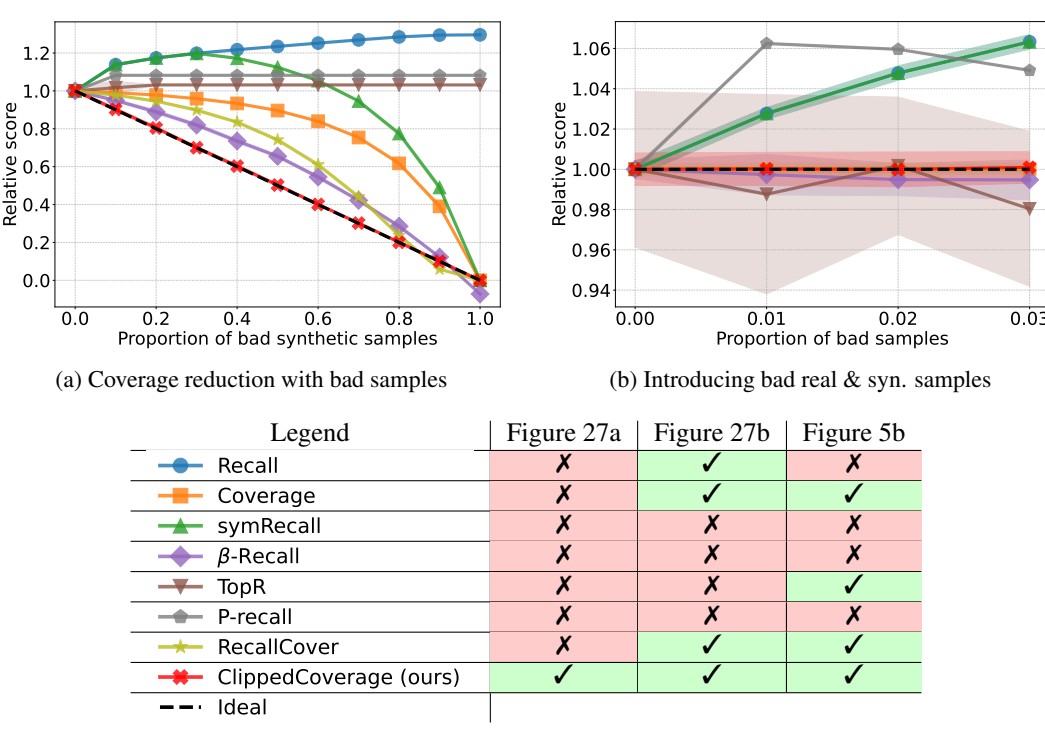

(a) Coverage reduction with bad samples

(b) Introducing bad real & syn. samples

| Legend | | Figure 27a | Figure 27b | Figure 5b |
|---|---|---|---|---|
| ● | Recall | ✗ | ✓ | ✗ |
| ■ | Coverage | ✗ | ✓ | ✓ |
| ▲ | symRecall | ✗ | ✗ | ✗ |
| ◆ | $\beta$-Recall | ✗ | ✗ | ✗ |
| ▼ | TopR | ✗ | ✗ | ✓ |
| ⬠ | P-recall | ✗ | ✗ | ✗ |
| ★ | RecallCover | ✗ | ✓ | ✓ |
| ✖ | ClippedCoverage (ours) | ✓ | ✓ | ✓ |
| - - | Ideal | | | |

(c) Legend and summary

Figure 27: **Testing coverage metrics on toy data**: this figure is the equivalent of Figure 5 on toy data.

# R UNNORMALIZED RESULTS

Figures 1, 4 and 5 present relative scores. This section shows the corresponding unnormalized scores, which include the maximum values and can be easier to read.

## R.1 MIXTURE OF GOOD AND BAD SAMPLES (CIFAR-10)

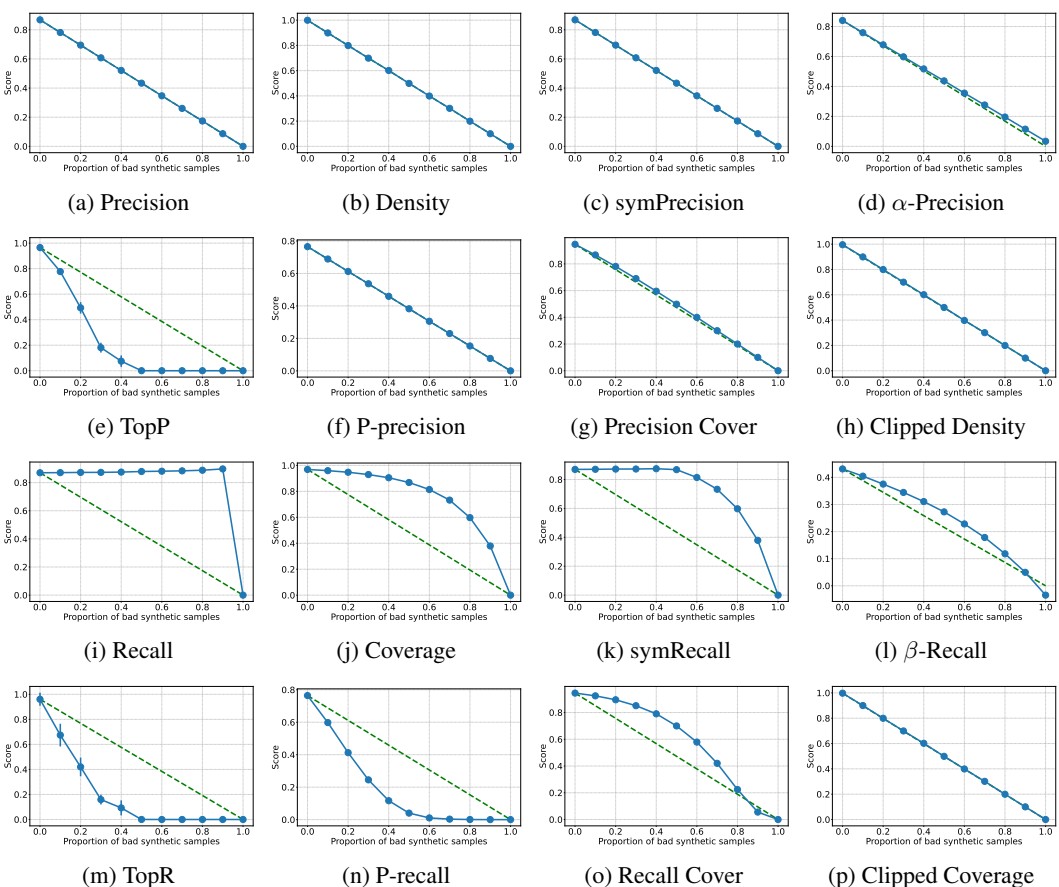

Figure 28: **Mixture of good and bad samples (CIFAR-10), unnormalized.**

## R.2 SIMULTANEOUS MODE DROPPING (CIFAR-10)

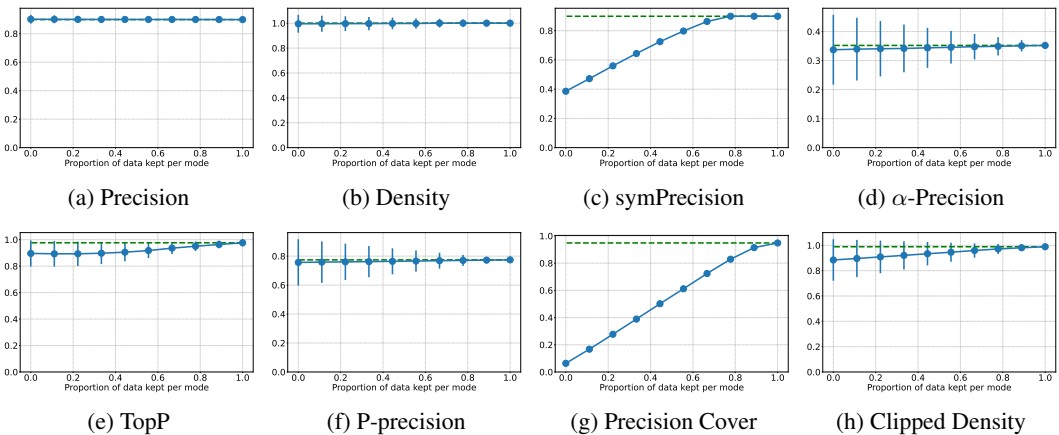

(a) Precision   (b) Density   (c) symPrecision   (d) $\alpha$-Precision

(e) TopP   (f) P-precision   (g) Precision Cover   (h) Clipped Density

Figure 29: **Simultaneously dropping modes (CIFAR-10), unnormalized.**

## R.3 MATCHED REAL & SYNTHETIC OUT-OF-DISTRIBUTION SAMPLES (CIFAR-10)

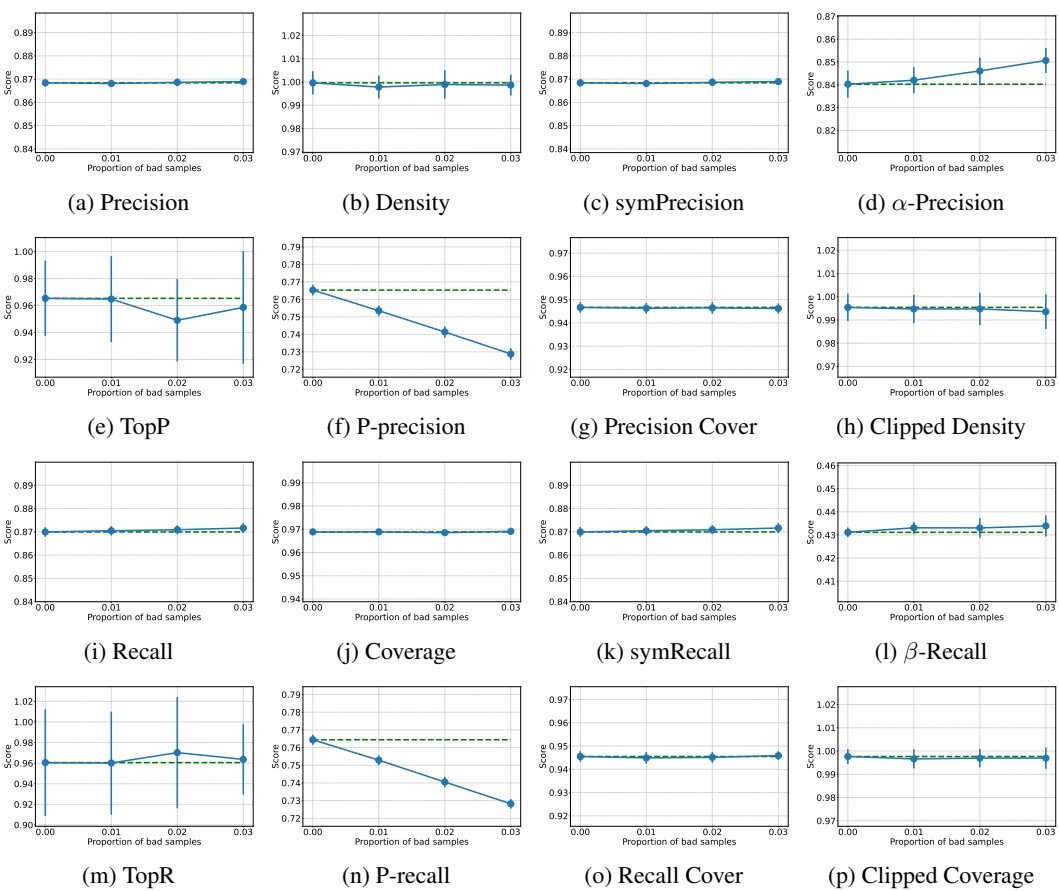

(a) Precision   (b) Density   (c) symPrecision   (d) $\alpha$-Precision

(e) TopP   (f) P-precision   (g) Precision Cover   (h) Clipped Density

(i) Recall   (j) Coverage   (k) symRecall   (l) $\beta$-Recall

(m) TopR   (n) P-recall   (o) Recall Cover   (p) Clipped Coverage

Figure 30: **Matched real & synthetic out-of-distribution samples (CIFAR-10), unnormalized.**

## R.4 SYNTHETIC DISTRIBUTION TRANSLATION

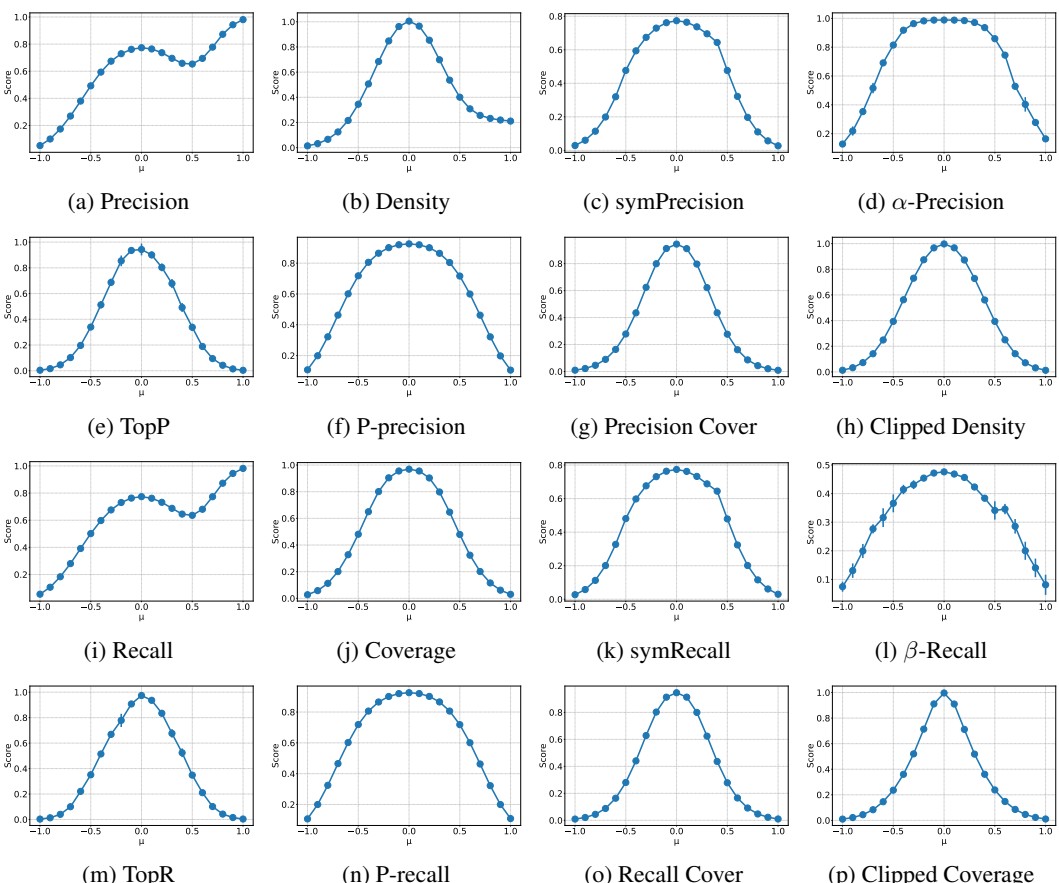

Figure 31: **Translating a synthetic Gaussian with 2 bad samples, unnormalized.**

# S VISUAL INSPECTION OF SAMPLES

In this section, we display real samples from each dataset (Figures 32, 34, 37 and 39), alongside data generated by the best-performing model for each case (Figures 33, 35, 38 and 40). Additionally, in Figure 36, we present manually annotated generated FFHQ samples to highlight visual artifacts and inconsistencies that may contribute to lower scores. These LDM samples primarily exhibit issues with background details, hands, and teeth, and occasionally with eyes and ears.

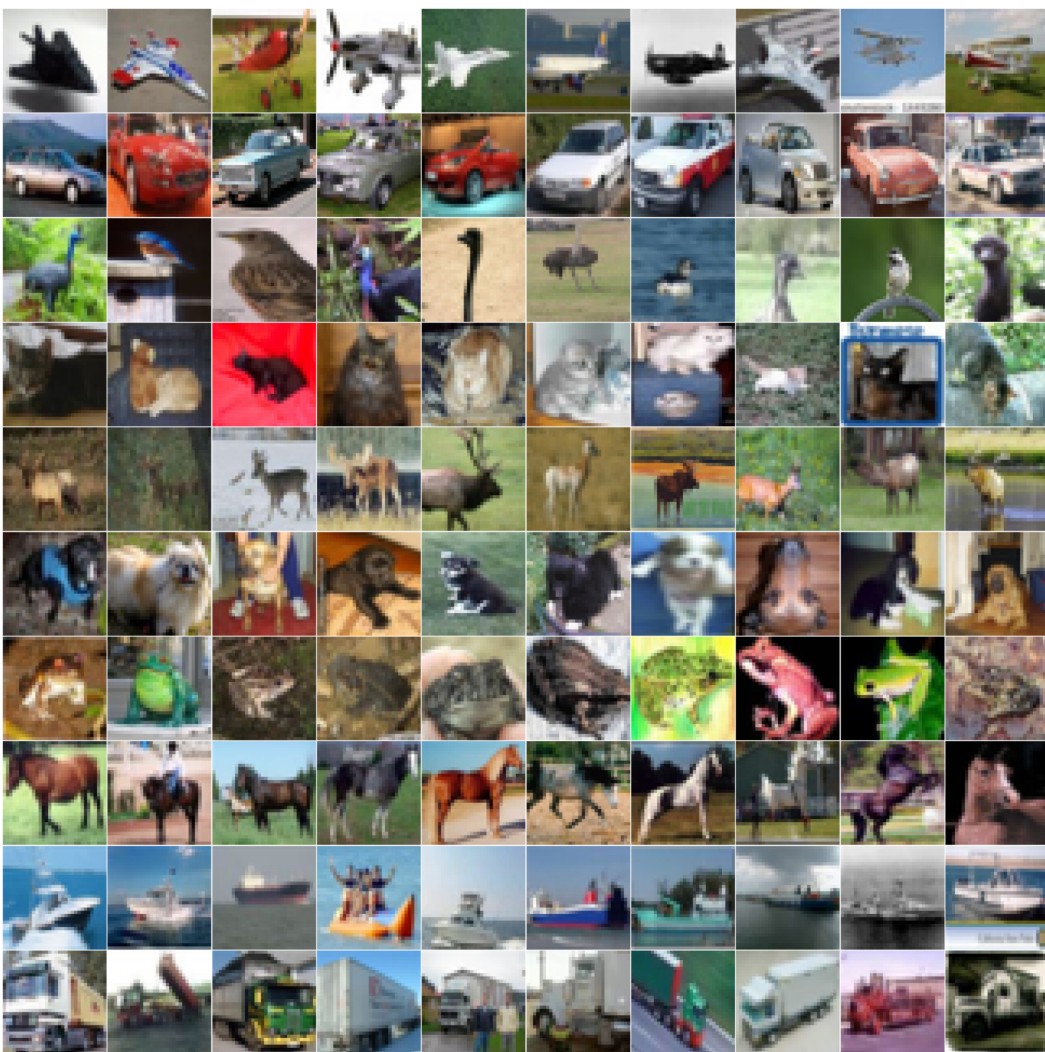

Figure 32: **Real samples from CIFAR-10**

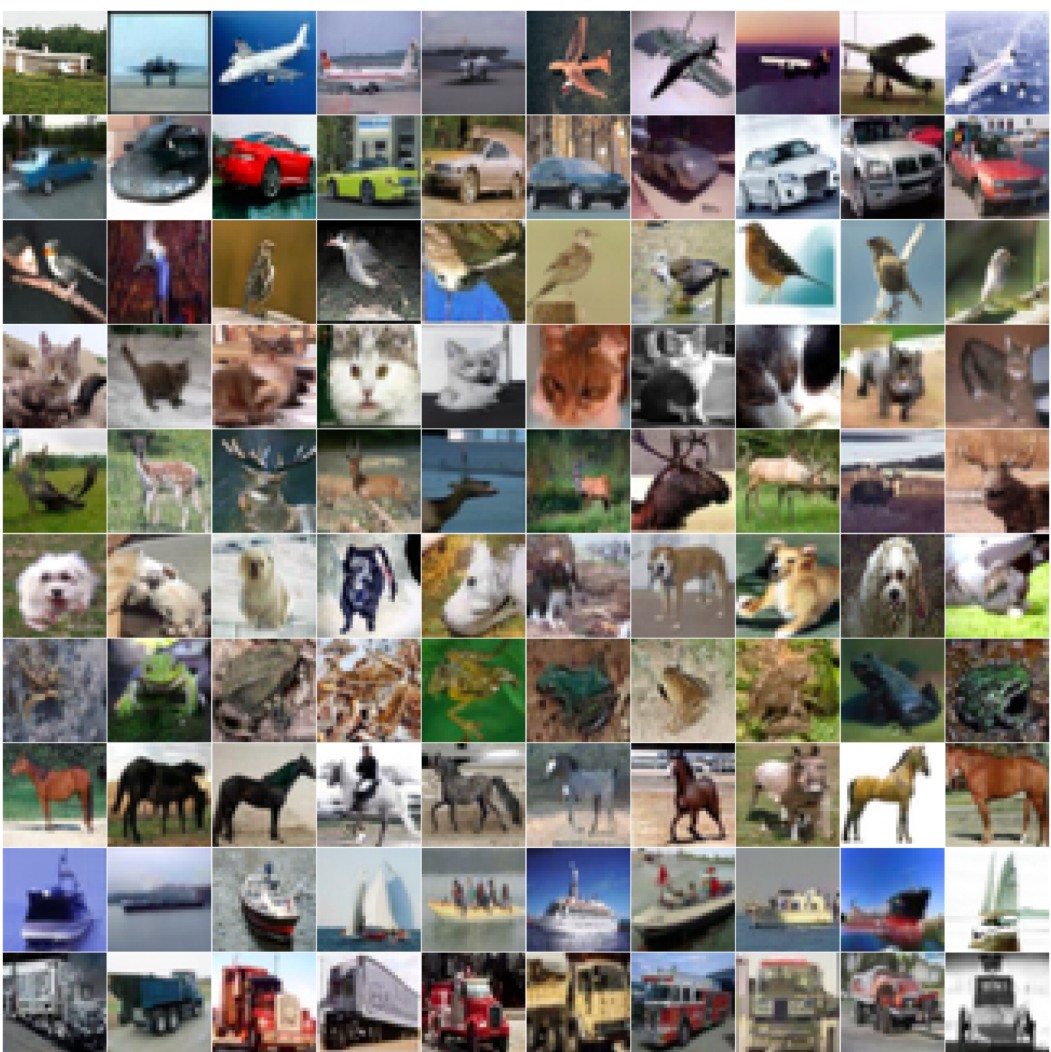

Figure 33: **Generated CIFAR-10 samples by PFGMPP**: one class per row.

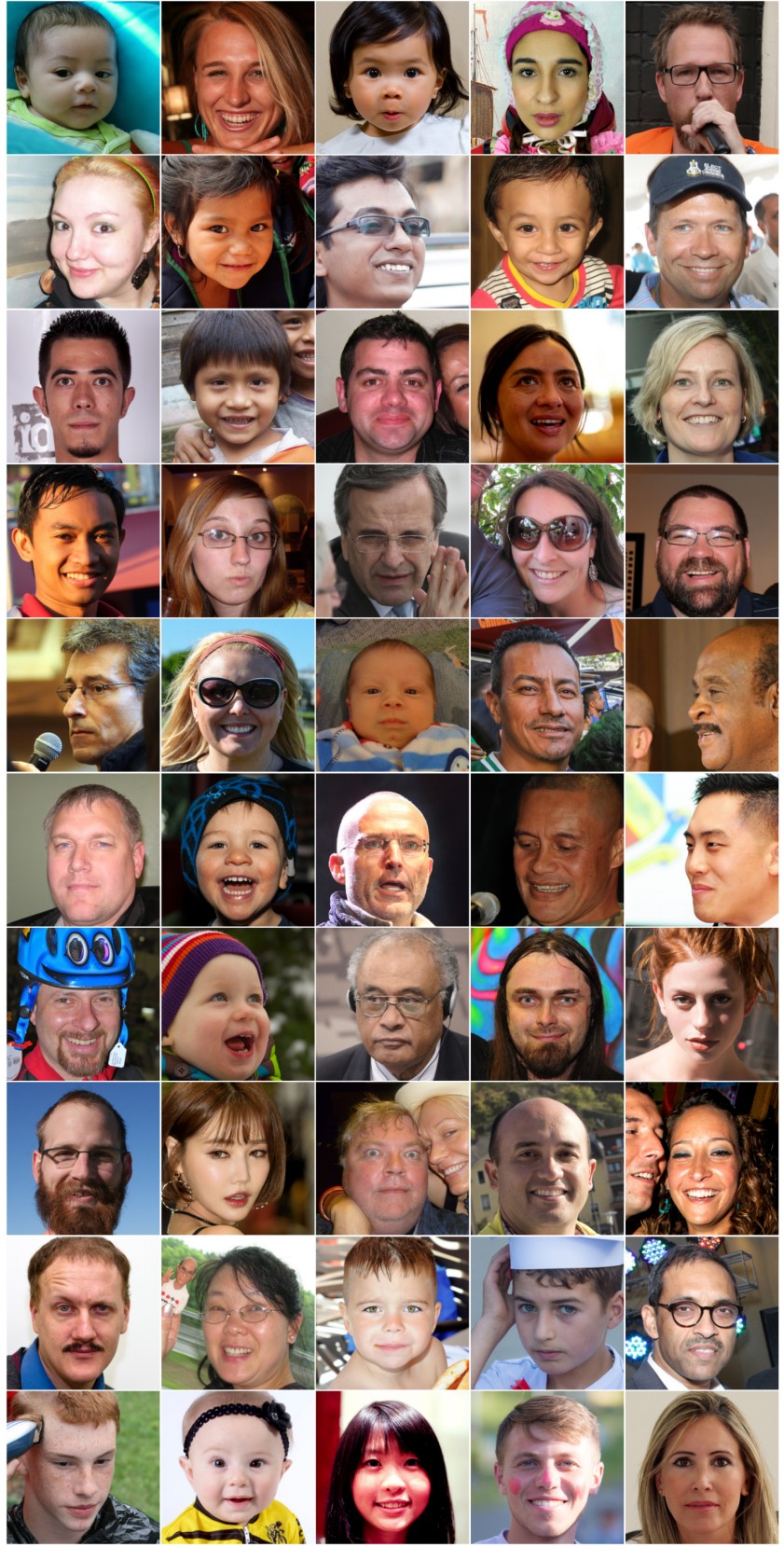

Figure 34: **Real samples from FFHQ**

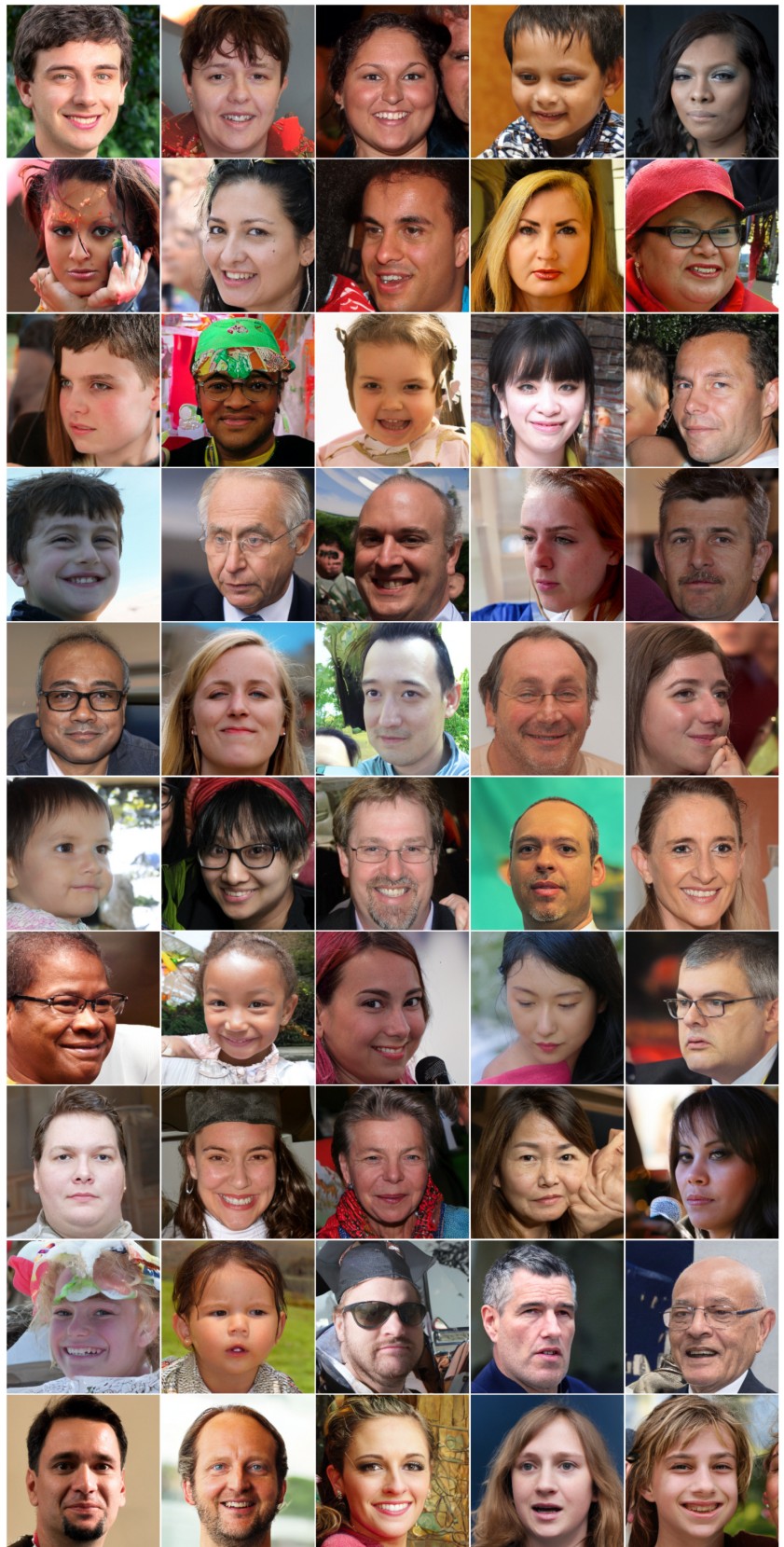

Figure 35: **Generated FFHQ samples by LDM**

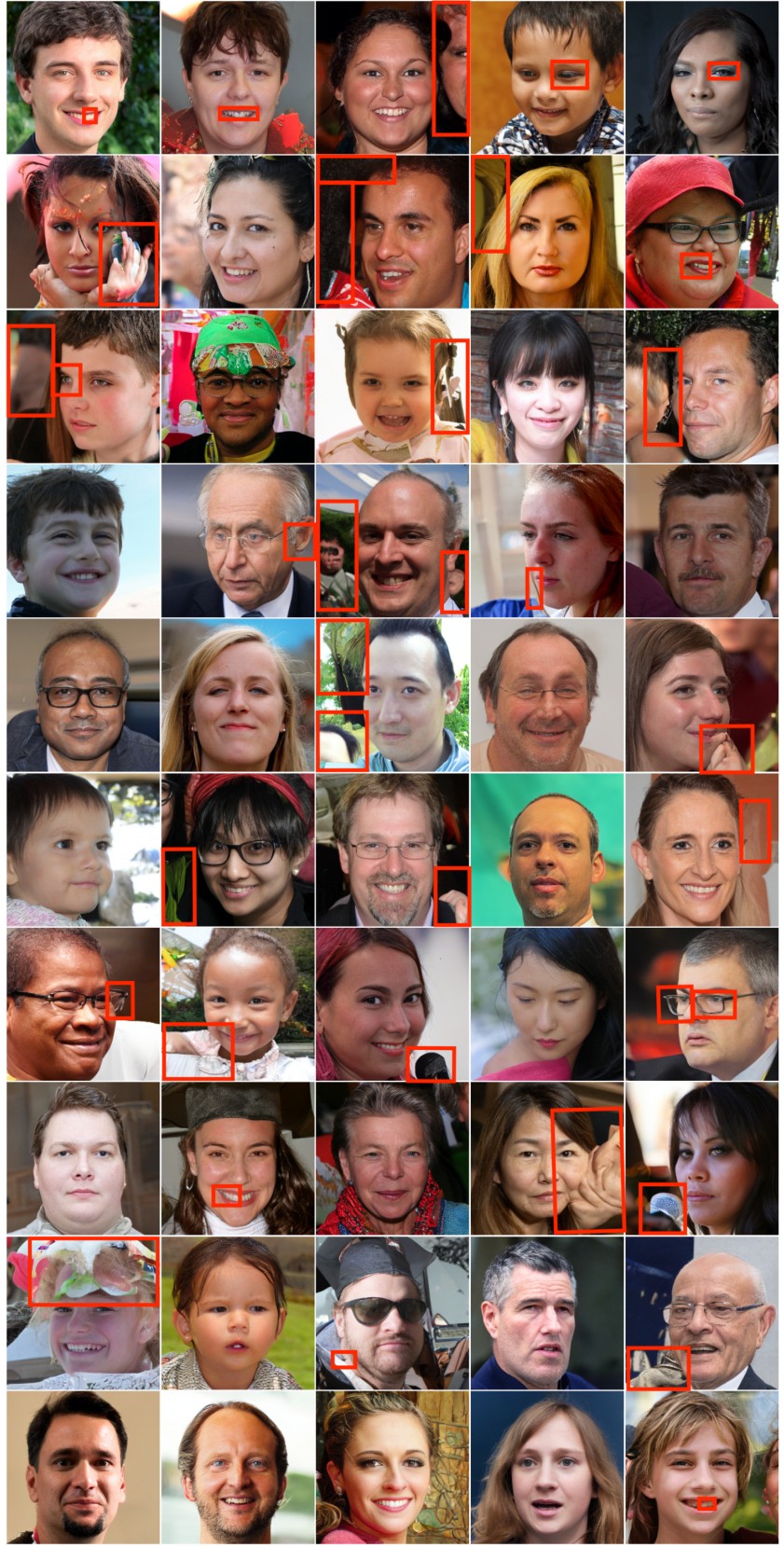

Figure 36: **Generated FFHQ samples by LDM with annotated defects.** We manually highlight visual artifacts and inconsistencies that may contribute to lower scores.

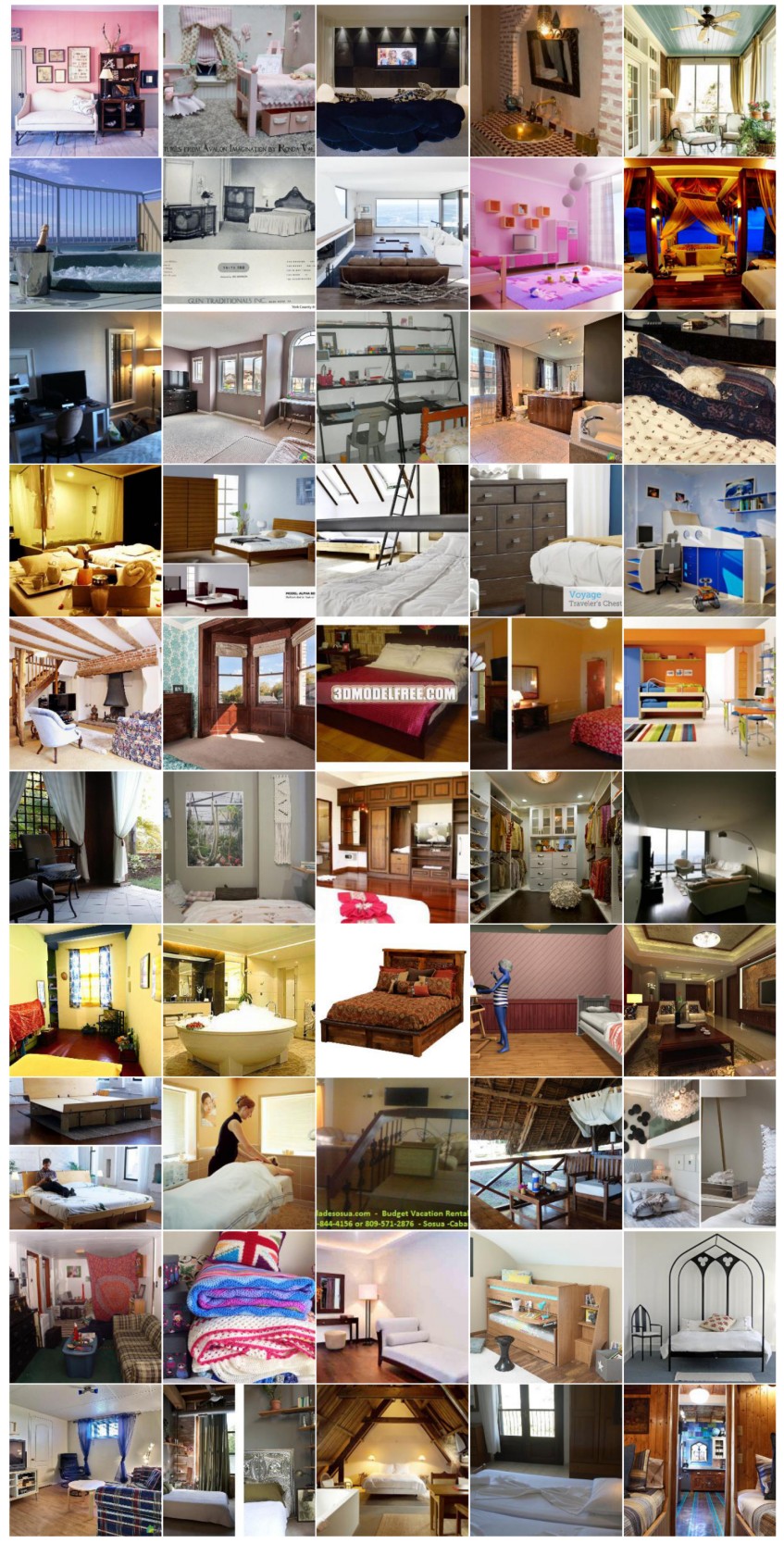

Figure 37: **Real samples from LSUN Bedroom**

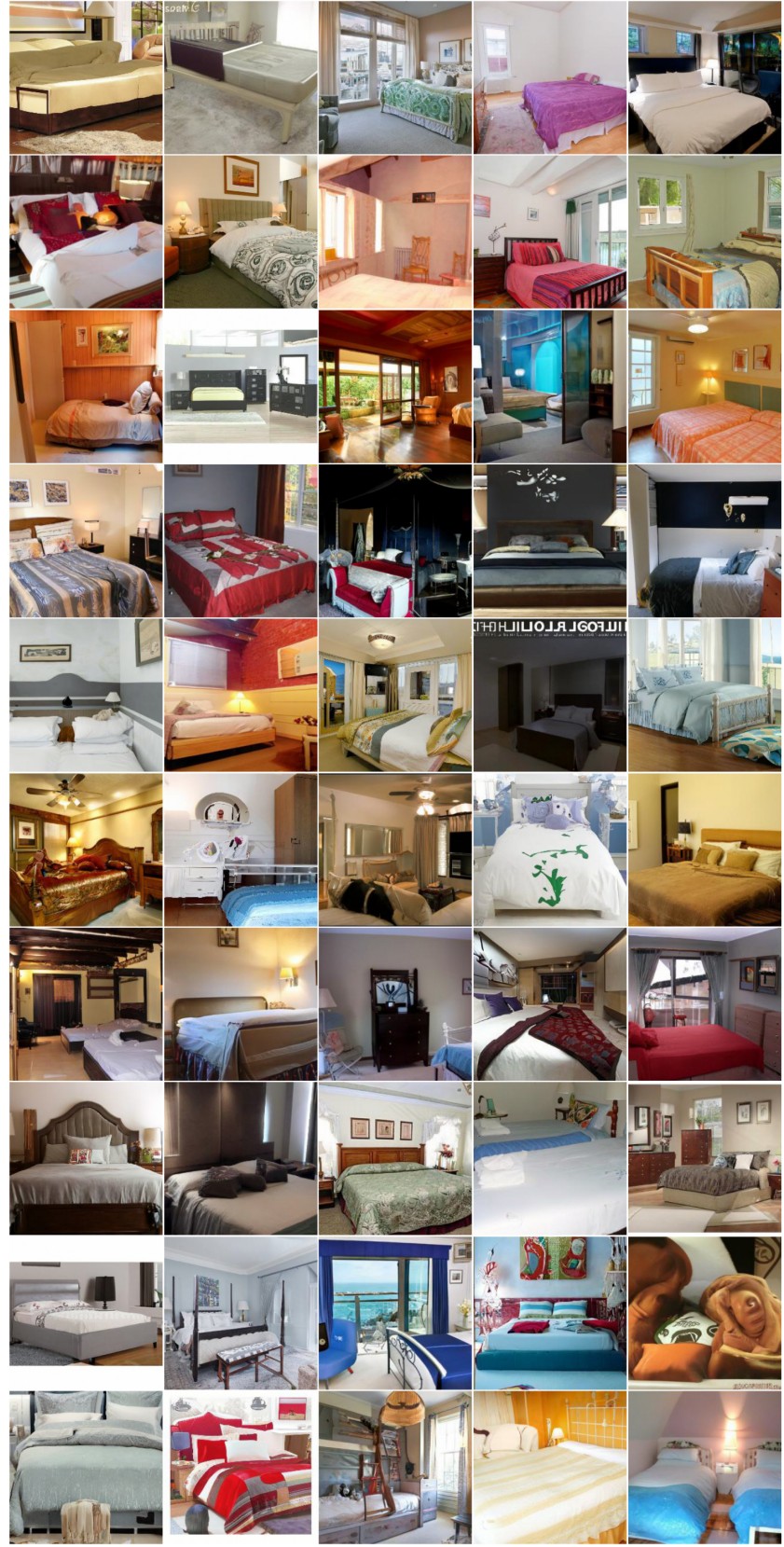

Figure 38: **Generated LSUN Bedroom samples by ADN-dropout**

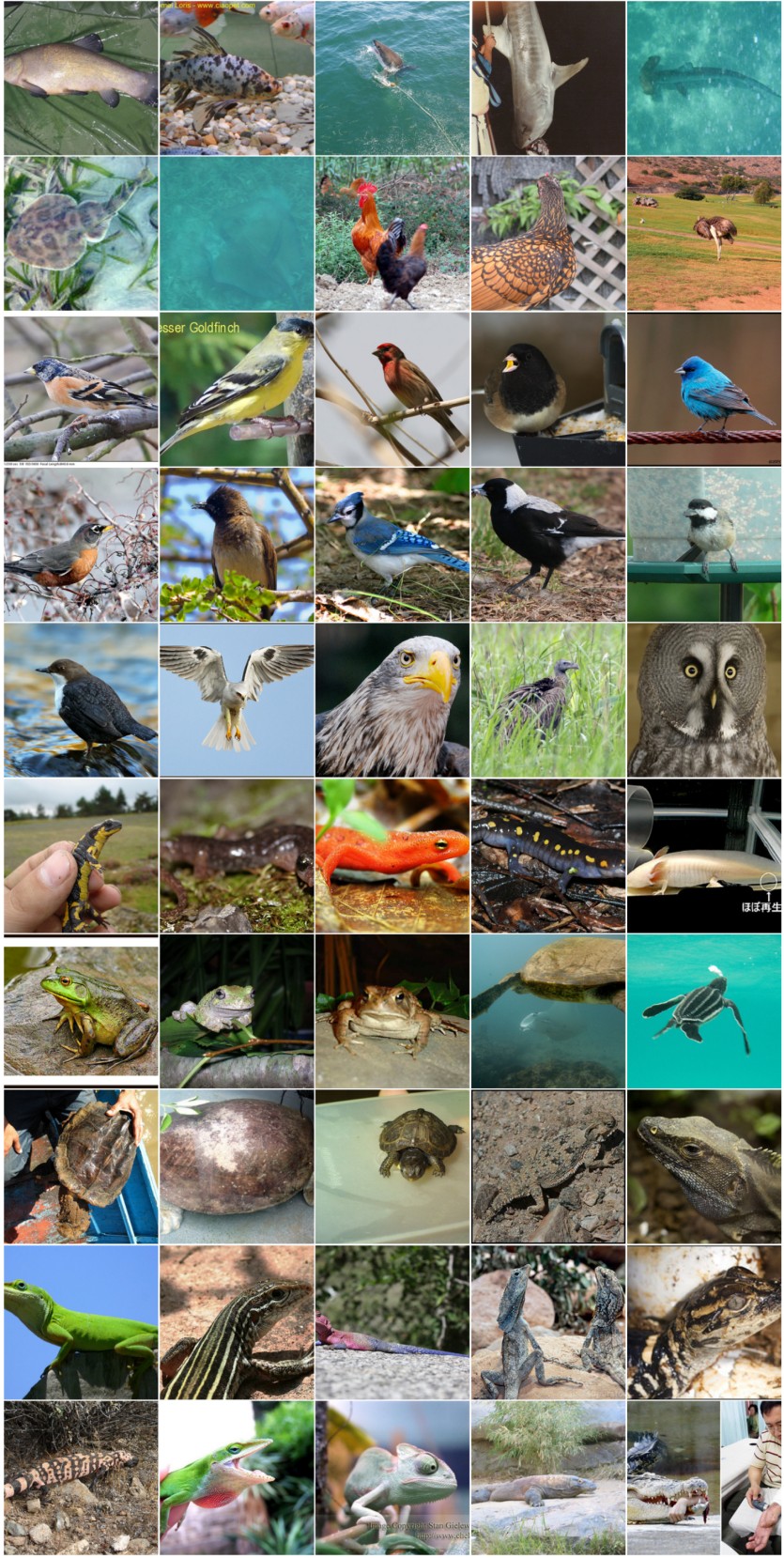

Figure 39: **Real samples from ImageNet**

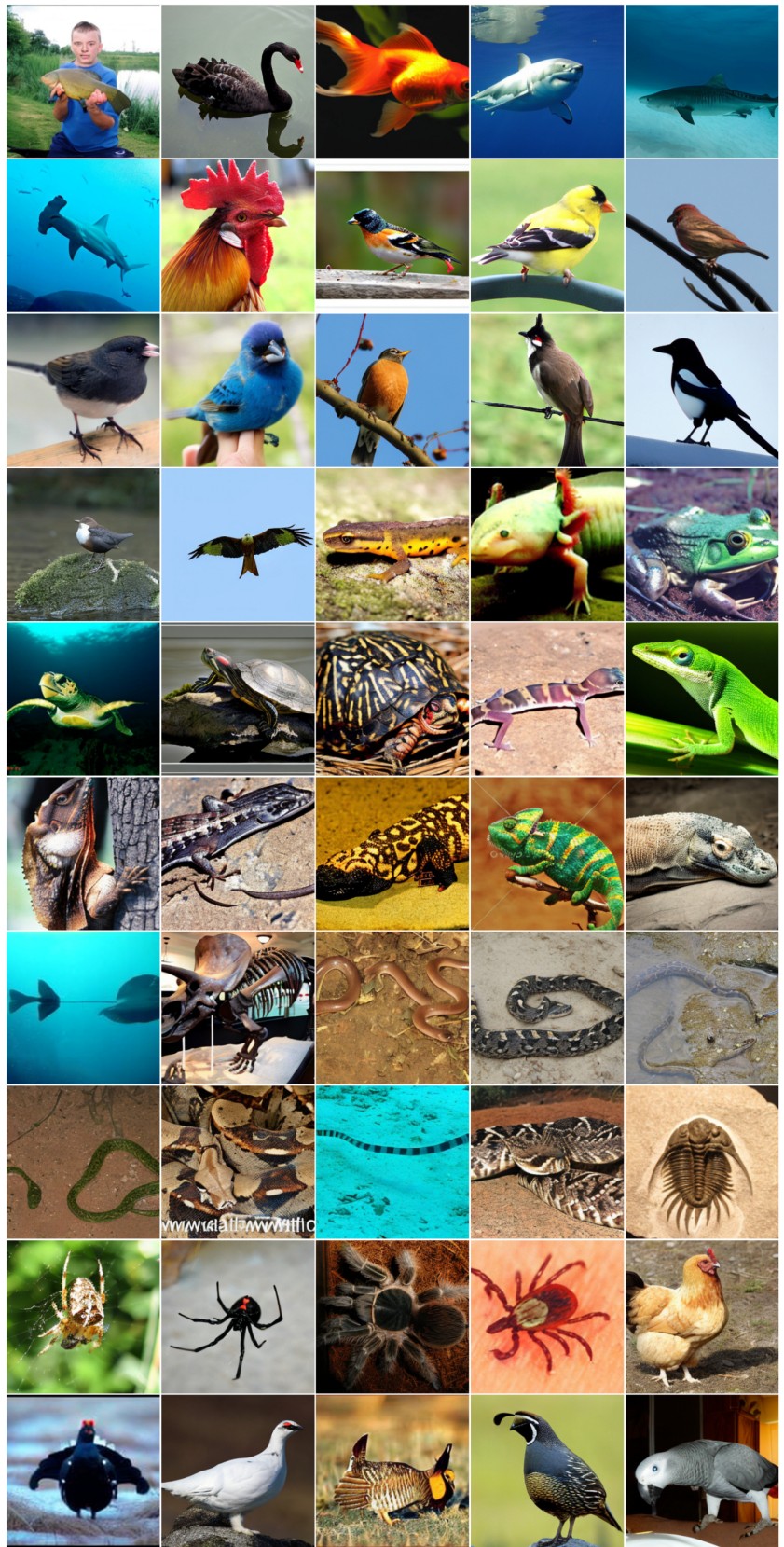

Figure 40: **Generated ImageNet samples by DiT-XL-2-guided**

