# OpenReview forum: "Enhanced Generative Model Evaluation with Clipped Density and Coverage"
_ICLR.cc/2026/Conference — ICLR 2026 Poster_

### Official Review · Reviewer_GJ4y · 2025-10-29

**Soundness:** 2
**Presentation:** 4
**Contribution:** 4
**Rating:** 6
**Confidence:** 4

**Summary:**

The paper proposes a pair of synthetic data fidelity and diversity metrics called clipped density and clipped coverage. They build on the previous density and coverage metrics, adding clipping to guard against pathological behaviors and normalisation to ensure the values are between 0 and 1 in practice, and degrade linearly if a proportion of samples is corrupted. The new metrics are evaluated using a some common sanity checks on image data, and a larger set of checks on artificial data from a recent benchmark. The new metrics pass all the image checks, and pass more artificial checks than previous metrics. The paper also compares many image generators on four standard image datasets with the new and existing metrics. The results with the new metrics are seemingly more informative than the ones with other metrics.

**Strengths:**

The paper convincingly argues that existing generative model fidelity and diversity metrics are not sufficient for many use cases, and explains why fixing these problems is important. The new metrics are presented clearly, and the writing overall is good. The ideas are novel, and the normalisations the paper introduces may also be useful with other fidelity and diversity metrics. The results on the sanity checks in the paper are good: the new metrics pass all of the image checks, and pass more artificial checks than any other metric.

**Weaknesses:**

The main weakness of the paper is the weak evaluation of the linear degradation claim for clipped density and clipped coverage. This claim is only supported by a few checks on image data. However, both metrics still fail many of the bounds checks from Räisä et al. (2025), indicating that the metrics do not always have the extreme values 0 and 1 in practice, which is a prerequisite for linear degradation. This means that the claim that only 40% of generated samples on CIFAR-10 and FFHQ are good, in Section 5.3, is not supported. It would be interesting to see a visual comparison between real and synthetic images for the datasets and models (or subsets of them) in Section 5.3. If the comparison shows a clear difference between the proportions of good and bad synthetic samples between the small and large datasets, this would provide more evidence that clipped density and clipped coverage can be interpreted as such proportions.

The undesirable behaviors the other metrics are claimed to have in Section 5.3 and Appendix I are not supported by evidence, with the exception of density having values greater than 1. This is because it is not known what the behavior should be: it is possible that in this setting, fidelity and diversity should be correlated, or that all the models have similar fidelity, or that the score ranges should be restricted, so there is no evidence that any of these behaviors is undesirable. Besides, it is not fair so say that correlated scores fidelity and diversity scores are a downside of other metrics, as clipped density and clipped coverage are also highly correlated in Figure 6.

In my opinion, clipping clipped density to 1 only hides the problem of the score exceeding 1 and does not increase interpretability. As a result, it would be good to include results without this clipping in the paper, or at least check when the unclipped score exceeded 1.

Minor points:
- In Figure 4b, clipped density is only close to 1 at the left edge, not extremely close to 1, which could be considered a failure compared to some of the other metrics.
- Figure 5b: it is very difficult to judge how symmetric symRecall is.
- Markers in all figures are small, so their shapes are hard to see.
- Figures 5 and 5: "Precall" -> "P-recall", and same for "Pprecision".
- Figure 8: the bar labels are hard to associate with the correct bars, especially on the right. Rotating them more should fix this.

**Questions:**

No further questions.

---

> ### Author Response · Authors · 2025-11-21
> **Answer Part 1/2**
>
> Thank you for your positive feedback on the paper’s motivation, clarity, and the effectiveness of our metrics.
>
> ### Calibration robustness
>
> > The main weakness of the paper is the weak evaluation of the linear degradation claim for clipped density and clipped coverage. This claim is only supported by a few checks on image data. However, both metrics still fail many of the bounds checks from Räisä et al. (2025), indicating that the metrics do not always have the extreme values 0 and 1 in practice, which is a prerequisite for linear degradation. This means that the claim that only 40% of generated samples on CIFAR-10 and FFHQ are good, in Section 5.3, is not supported. It would be interesting to see a visual comparison between real and synthetic images for the datasets and models (or subsets of them) in Section 5.3. If the comparison shows a clear difference between the proportions of good and bad synthetic samples between the small and large datasets, this would provide more evidence that clipped density and clipped coverage can be interpreted as such proportions.
>
> We further studied the stability of calibration in Appendix M.1 (Figures 21 and 22).
>
> We reproduced the evaluation of mixtures of good and bad samples using ImageNet, as shown in Figure 21. Both clipped metrics decrease linearly with the proportion of bad samples. The standard deviation across 10 repetitions with different subsets is at most $0.0056$ for *Clipped Density* and $0.0082$ for *Clipped Coverage*.
>
> In Figure 22, we further analyze how the standard deviation scales with sample size, showing it decreases as $1/\sqrt{N}$ for both metrics and drops below $0.01$ for $N=50000$. We attribute the bound check failures for Räisä et al. (2025) discussed in Appendix H.1. to the low sample size of only 1000 samples per set.
>
> We repeated the evaluation of mixtures of good and bad samples using musical data in Appendix M.4 (Figure 26), further confirming the absolute calibration of the metrics on another modality.
>
> In Appendix M.8, we display real samples from each dataset (Figures 32, 34, 37, and 39), alongside data generated by the best-performing model for each case (Figures 33, 35, 38, and 40). Additionally, in Figure 36, we manually annotate generated FFHQ samples to highlight visual artifacts and inconsistencies that may contribute to lower scores. The LDM samples primarily exhibit issues in background details, hands, teeth, and occasionally eyes and ears.
>
>
>
> > Minor point:
> > - In Figure 4b, clipped density is only close to 1 at the left edge, not extremely close to 1, which could be considered a failure compared to some of the other metrics.
>
> In Figure 4b, we counted three failures: *symPrecision* and *Precision Cover* (green and yellow curves), which strongly diverge. We also counted *$\alpha$-Precision* (purple) as a failure due to its high relative standard deviation ($>0.4$). While on other graphs we might consider the level of proximity of *Clipped Density* to 1 as a failure, this test uses only 2500 samples compared to 25000 in others. Thus, we allowed some leniency for the achieved values if the correct value falls within the range of value $\pm$ std, provided it is not highly unstable.

---

> > ### Author Response · Authors · 2025-11-21
> > **Answer Part 2/2**
> >
> > ### Unknown ideal behavior on real data
> > > The undesirable behaviors the other metrics are claimed to have in Section 5.3 and Appendix I are not supported by evidence, with the exception of density having values greater than 1. This is because it is not known what the behavior should be: it is possible that in this setting, fidelity and diversity should be correlated, or that all the models have similar fidelity, or that the score ranges should be restricted, so there is no evidence that any of these behaviors is undesirable. Besides, it is not fair so say that correlated scores fidelity and diversity scores are a downside of other metrics, as clipped density and clipped coverage are also highly correlated in Figure 6.
> >
> > **You are correct.** We acknowledge that, since the correct behavior is unknown, we should not conclude that certain behaviors are erroneous. We have removed such conclusions from Appendix I and have rewritten it accordingly: (changes in bold)
> >
> > >> **We observe that for some** metric pairs, fidelity **scores vary little, while** coverage metrics **vary significantly**. This occurs for *Precision*/*Recall* (CIFAR-10, LSUN Bedroom), *TopP*/*TopR* (ImageNet, FFHQ), *P-precision*/*P-recall* (FFHQ), and *Precision Cover*/*Recall Cover* (FFHQ). *symPrecision* and *symRecall* **show** correlated **scores**. *$\alpha$-Precision* and *$\beta$-Recall* show distinct results for consistency models on LSUN Bedroom **and yield** scores generally confined to the lower-right quadrant.
> > >>
> > >> **However, the correct behavior of metrics on real data is unknown, so we cannot draw definitive conclusions about which metric pairs are better or worse from these plots alone.**
> >
> >
> > ### Scores exceeding 1
> > > In my opinion, clipping clipped density to 1 only hides the problem of the score exceeding 1 and does not increase interpretability. As a result, it would be good to include results without this clipping in the paper, or at least check when the unclipped score exceeded 1.
> >
> > Thank you for this remark. We included tests results without clipping in Figure 30. The scores remain stable near 1 as expected. In our test settings, *Clipped Density* does not significantly exceed 1.
> >
> > In the main text, only one unclipped result exceeds 1: the guided DiT-XL-2 model on ImageNet in our evaluation of advanced generated datasets (Figure 6). It also occurs when evaluating StyleGAN LSUN Bedroom data, generated with different truncation parameter values (Appendix M.5, Figure 27) to demonstrate the fidelity-coverage trade-off. Indeed GANs sometimes sample from a truncated latent space to improve fidelity at the expense of coverage. A smaller truncation value $\psi$ restricts sampling to a smaller, denser region.
> >
> > In both cases, a sample-level analysis (Figures 28-29) show an increased proportion of synthetic samples with perfect fidelity, surpassing that of real samples. Both guidance and truncation alter the sampling, to favor high-density regions and ignore low-density ones.
> >
> > Overall, a *Clipped Density* score above 1 occurs when sampling avoids low-density regions, effectively targeting a **filtered** real distribution. This is a feature of our unclipped metric: it correctly reflects that the synthetic set is sampled from a different distribution than the unfiltered real reference. For this reason, we now leave the option to the user not to clip the fidelity result.
> >
> >
> >
> > ### Figure and legend improvements
> > > Minor points:
> > > - Figure 5b: it is very difficult to judge how symmetric symRecall is.
> >
> > Please refer to Figure 20 (k), where the asymmetry is easier to see.
> >
> > > - Markers in all figures are small, so their shapes are hard to see.
> > > - Figures 5 and 5: "Precall" -> "P-recall", and same for "Pprecision".
> > > - Figure 8: the bar labels are hard to associate with the correct bars, especially on the right. Rotating them more should fix this.
> >
> > We have increased the size of all markers, corrected the legends and applied the suggested changes to Figure 8.
> >
> > ### Reference
> >
> > Ossi Räisä, Boris van Breugel, and Mihaela van der Schaar. Position: All current generative fidelity and diversity metrics are flawed. In Forty-second International Conference on Machine Learning Position Paper Track, 2025.

---

> ### Comment · Reviewer_GJ4y · 2025-11-23
>
> Thank you for the response and updates. Your changes have addressed basically all of my concerns, so I've raised the score.
>
> I would recommend highlighting the changes in the updated pdf submission so they are easier for reviewers to find.

---

> > ### Author Response · Authors · 2025-11-24
> >
> > Thank you! We have shared a new pdf version with changes highlighted outside of Appendix R.

---

### Official Review · Reviewer_oPHo · 2025-10-29

**Soundness:** 2
**Presentation:** 2
**Contribution:** 2
**Rating:** 6
**Confidence:** 4

**Summary:**

The paper clearly identifies limitations of current generative model evaluation metrics (e.g., FID, Precision/Recall): they combine fidelity and coverage into a single number, lack calibration, and are sensitive to outliers. This motivates the need for metrics that separately assess fidelity and coverage and provide interpretable, robust scores. The authors introduce Clipped Density and Clipped Coverage. Both metrics cap individual sample contributions at 1, preventing over‑occurring samples from masking defects and ensuring robustness to outliers. Clipped Density further clips nearest‑neighbor radii to the median k‑th neighbor distance, limiting the effect of sparse or outlier real data points. Clipped Coverage counts the number of synthetic samples within each real sample’s ball, normalizes by k, and caps at 1.

**Strengths:**

- Calibration for Interpretability: The metrics are calibrated so that their expected value decays linearly as the fraction of bad synthetic samples increases. For Clipped Density, the unnormalized score is divided by the fidelity score computed on the real data and then clipped to [0,1]. For Clipped Coverage, the authors derive the expected value under an i.i.d. assumption using Beta functions and numerically invert it to map unnormalized scores to linear decay. This calibration allows scores to be interpreted directly as the proportion of “good” samples.

- Comprehensive Testing: The paper proposes a suite of controlled tests (mode dropping, introduction of out‑of‑distribution samples, synthetic data translation) to evaluate robustness, sensitivity, and interpretability. Experiments show that existing metrics often conflate fidelity with coverage or are unstable, whereas Clipped Density and Clipped Coverage exhibit linearity, stability, and symmetry across tests.

**Weaknesses:**

- Incremental Novelty: While clipping sample contributions and radii is a sensible modification, the metrics essentially adapt existing Density/Coverage measures. The idea of capping contributions is intuitive; no fundamentally new notion of fidelity or coverage is introduced. The theoretical calibration for Clipped Coverage relies on i.i.d. assumptions and numerical inversion, which is mathematically involved but conceptually just rescales the metric to match a linear decay.

- Scope Limited to Images: Experiments and definitions focus on image data. Although the authors claim the framework is generic, there is no evidence it extends to other modalities (video, audio).

- The absolute scores are interpreted as equivalent proportions of “good” samples, but this relies on synthetic tests where bad samples are pure noise. In practice, generative models may produce subtle artifacts or distributional shifts that do not correspond neatly to “bad samples,” and the linear calibration may not reflect perceptual quality. The tests used to justify the metrics (mode dropping, noise replacement, translation) are tailored; there may be other failure modes (e.g., adversarial patterns, style transfer) where the metrics behave unpredictably.

**Questions:**

- Have you tested the metrics on non‑image generative models (e.g., video or audio)? If not, what challenges do you foresee in extending the method to other modalities?

- Interpretation of Scores: The calibration interprets a score of 0.4 as “40 % good samples and 60 % bad samples”. In real generative models, “bad samples” may not be pure noise but may include subtle artifacts or low‑quality variations. How should practitioners interpret intermediate scores in such contexts?

- For Clipped Coverage, the calibration requires computing and numerically inverting an expectation using Beta functions. How sensitive is the final metric to errors in this inversion?

- Have you conducted any human evaluation to correlate Clipped Density/Coverage scores with perceived quality or diversity? Without such correlation, it is unclear if the metrics align with human judgments.

---

> ### Author Response · Authors · 2025-11-21
> **Answer Part 1/3**
>
> Thank you for your thoughtful review and positive feedback on our metrics' design and testing.
>
> ### Novelty
>
> > - Incremental Novelty: While clipping sample contributions and radii is a sensible modification, the metrics essentially adapt existing Density/Coverage measures. The idea of capping contributions is intuitive; no fundamentally new notion of fidelity or coverage is introduced. The theoretical calibration for Clipped Coverage relies on i.i.d. assumptions and numerical inversion, which is mathematically involved but conceptually just rescales the metric to match a linear decay.
>
> We agree that the novelty is mostly incremental but would like to remind the reviewer that the status quo, as described in the independent position paper by Räisä et al. (2025), is that "All Current Generative Fidelity and Diversity Metrics are Flawed", and that no fidelity and diversity metrics could allow "gauging how good or bad the synthetic data is in absolute terms, not only in comparison to other synthetic datasets." Our work tackles these issues.
>
>
> ### New modality: Audio
> > - Scope Limited to Images: Experiments and definitions focus on image data. Although the authors claim the framework is generic, there is no evidence it extends to other modalities (video, audio).
>
> > - Have you tested the metrics on non‑image generative models (e.g., video or audio)? If not, what challenges do you foresee in extending the method to other modalities?
>
> We added to the revised version tests performed on **music** samples in Appendix M.4. We used the Free Music Archive large dataset, comprising 30s clips from 106574 tracks (Defferrard et al., 2017). Following Gui et al. (2024), who compared 10 embedding models for music, we used the CLAP model (Elizalde et al., 2023).  As it extracts 1024-dimensional embeddings from 7s clips, we selected the middle 7s of each track.
>
> An initial evaluation on balanced real/synthetic splits of 50000 samples each yielded a *Clipped Density* of 0.988 and a *Clipped Coverage* of 0.998. When splitting by genre, scores decreased to 0.943 for *Clipped Density* and 0.875 for *Clipped Coverage*. This reduction is moderate but expected, as music tracks span multiple genres, yet we split based on a single genre per track (a track's other associated genres might correspond to the other dataset).
>
> Finally, we evaluated mixtures of good and bad samples (Figure 26), where bad samples consisted of noise MP3 files. We used $N=50000$ balanced CLAP embeddings per set. Both metrics decreased linearly with the proportion of bad samples, confirming the absolute calibration on this high-dimensional music dataset.
>
> The main challenge for practitioners in extending this framework to other modalities lies in accessing an embedding model that serves as an effective feature extractor. Such a model is essential for compution of distances between features that are more relevant than raw observations, thereby ensuring that the distances are semantically meaningful. While DINOv2 is well-established for images, a comparable standard for other modalities does not always exist.
> For research on evaluation metrics, another significant challenge is the availability of synthetic datasets. We chose to evaluate the datasets shared by Stein et al. (2023) because they provide a collection of 42 generated image datasets across 4 different training sets. However, finding pre-generated sets that are ready for evaluation in other domains remains complicated.

---

> ### Author Response · Authors · 2025-11-21
> **Answer Part 2/3**
>
> ### Interpretability
> > - The absolute scores are interpreted as equivalent proportions of “good” samples, but this relies on synthetic tests where bad samples are pure noise. In practice, generative models may produce subtle artifacts or distributional shifts that do not correspond neatly to “bad samples,” and the linear calibration may not reflect perceptual quality. The tests used to justify the metrics (mode dropping, noise replacement, translation) are tailored; there may be other failure modes (e.g., adversarial patterns, style transfer) where the metrics behave unpredictably.
>
> > - Interpretation of Scores: The calibration interprets a score of 0.4 as “40 % good samples and 60 % bad samples”. In real generative models, “bad samples” may not be pure noise but may include subtle artifacts or low‑quality variations. How should practitioners interpret intermediate scores in such contexts?
>
> *Clipped Density* and *Clipped Coverage* are not binary. Instead, they assess a graduation of sample quality, based on either the number of balls a given point falls into or the number of points within a given point's ball. In both cases, higher values are better.
>
> A score of 0.4 does not imply that 40% of generated samples are good while the others are as bad as noise. Rather, this score is **equivalent** to what would be obtained if 40% of the samples were as good as real data and 60% were bad (i.e., *not contributing to the score*). This simple mixture serves as a **reference** point for interpretability, though other mixtures with average-quality samples could yield the same score.
>
> We acknowledge the difficulty of guaranteeing that a metric will not behave unpredictably under all possible degradations. To address this, we empirically subject the metrics to extensive stress testing. This includes evaluations on real data presented in the main text, as well as on the synthetic benchmark proposed by Räisä et al. (2025) in Appendix H.
>
>
> ### Numerical stability
>
> > - For Clipped Coverage, the calibration requires computing and numerically inverting an expectation using Beta functions. How sensitive is the final metric to errors in this inversion?
>
> Although it is performed *numerically*, which might suggest an approximation, the inversion is **exact** because it is discrete.
>
> Since the number of samples $M$ is finite, the expected score function $f_{\text{expected}}$ is defined on a discrete domain of integers $m \in \\{0, \dots, M\\}$. We precompute these $M+1$ exact values. The inversion is therefore not a continuous numerical approximation but a discrete lookup within this sorted list. Thus, the metric is not sensitive to inversion errors, as it maps directly to the correct exact values.
>
>
> ### Comparison with human evaluations
> > - Have you conducted any human evaluation to correlate Clipped Density/Coverage scores with perceived quality or diversity? Without such correlation, it is unclear if the metrics align with human judgments.
>
> We added comparisons with human evaluation in Appendix M.7 (Figure 31). To do so, we used human error rates from Stein et al. (2023), where subjects were asked to distinguish between real and generated images. A higher error rate indicates a better generative model. We computed the correlation coefficients (and p-values) between human error rates and *Clipped Density* and *Clipped Coverage*. All **correlations are high** ($>0.8$) except for the FFHQ dataset, where no significant correlation was found, consistent with Stein et al. (2023) (grayed in their Figure 4, DINOv2 column).

---

> > ### Author Response · Authors · 2025-11-21
> > **Answer Part 3/3**
> >
> > ### References
> >
> > - Michaël Defferrard, Kirell Benzi, Pierre Vandergheynst, and Xavier Bresson. FMA: A dataset for music analysis. In Sally Jo Cunningham, Zhiyao Duan, Xiao Hu, and Douglas Turnbull (eds.), Proceedings of the 18th International Society for Music Information Retrieval Conference, ISMIR 2017, Suzhou, China, October 23-27, 2017, pp. 316–323, 2017.
> > - Benjamin Elizalde, Soham Deshmukh, Mahmoud Al Ismail, and Huaming Wang. CLAP learning audio concepts from natural language supervision. In IEEE International Conference on Acoustics, Speech and Signal Processing ICASSP 2023, Rhodes Island, Greece, June 4-10, 2023, pp. 1–5. IEEE, 2023.
> > - Azalea Gui, Hannes Gamper, Sebastian Braun, and Dimitra Emmanouilidou. Adapting frechet audio distance for generative music evaluation. In IEEE International Conference on Acoustics, Speech and Signal Processing, ICASSP 2024, Seoul, Republic of Korea, April 14-19, 2024, pp. 1331–1335. IEEE, 2024.
> > - Ossi Räisä, Boris van Breugel, and Mihaela van der Schaar. Position: All current generative fidelity and diversity metrics are flawed. In Forty-second International Conference on Machine Learning Position Paper Track, 2025.
> > - George Stein, Jesse Cresswell, Rasa Hosseinzadeh, Yi Sui, Brendan Ross, Valentin Villecroze, Zhaoyan Liu, Anthony L Caterini, Eric Taylor, and Gabriel Loaiza-Ganem. Exposing flaws of generative model evaluation metrics and their unfair treatment of diffusion models. Advances in Neural Information Processing Systems, 36, 2023

---

> > > ### Comment · Reviewer_oPHo · 2025-11-26
> > >
> > > Thank you for the clarifications. I am largely satisfied with the rebuttal, but my view on novelty remains unchanged: the paper does not introduce a fundamentally new notion of generative quality or diversity. Given the acknowledged incremental nature, I see the contribution primarily as an engineering and evaluation refinement rather than a conceptual breakthrough. I will therefore maintain my original score.

---

### Official Review · Reviewer_6bfz · 2025-11-01

**Soundness:** 2
**Presentation:** 4
**Contribution:** 3
**Rating:** 6
**Confidence:** 4

**Summary:**

This paper proposes the metrics Clipped Density and Clipped Coverage, which are enhanced based on the Density and Coverage metrics [1]. The enhancement comes in two aspects: (1) robustness to outliers, and (2) that the metrics are linear and normalized (thus more interpretable). This is achieved by capping per-sample contribution, normalization, and calibration based on empirical analyses. The effectiveness is validated by comprehensive experiments covering several competing metrics, different types of generative models, and various sizes of datasets.




[1] Naeem, Muhammad Ferjad, et al. "Reliable fidelity and diversity metrics for generative models." International conference on machine learning. PMLR, 2020.

**Strengths:**

- The paper is nicely written and easy to read, equipped with nice visualizations.
- The study is well motivated.
- It is novel that the proposed metrics are designed to satisfy robustness, linearity, and interpretability.
- The analyses and experiments are comprehensive.

**Weaknesses:**

- The modifications applied to meet the proposed desiderata appear ad-hoc and lack theoretical justification. This raises the question of whether they might negatively impact performance in other aspects, such as distinguishability.

**Questions:**

- In Eq. (7), why should we normalize by $ClippedDensity_{real}$? When would $\frac{ClippedDensity_{unnorm}}{ClippedDensity_{real}}$ be larger than 1 and how should we interpret it? Why can it be safely clipped to 1?
- It seems that Clipped Coverage is not guaranteed to be <= 1. Is this correct? How should we interpret it when it is larger than 1?
- Fidelity and coverage are usually known to be a trade-off [2] for the same model. Clipped metrics in Figure 6 do not reflect this (, although the comparisons are among different models). Can you further verify this?
- L122-123, the sentence "In high-dimensional spaces, ... as a proxy" seems off here. Can you clarify?

[2] Li, Shuangqi, et al. "Controlling the Fidelity and Diversity of Deep Generative Models via Pseudo Density." Transactions on Machine Learning Research.

---

> ### Author Response · Authors · 2025-11-21
> **Answer Part 1/2**
>
> We thank the reviewer for their positive feedback on the paper’s clarity, motivation, and the novelty and thoroughness of our metrics and analyses.
>
> ### Validating the metrics
>
> > Weakness: The modifications applied to meet the proposed desiderata appear ad-hoc and lack theoretical justification. This raises the question of whether they might negatively impact performance in other aspects, such as distinguishability.
>
>
> While a theoretical motivation for clipping radii is to improve the approximation of the support by mitigating the influence of outliers, we acknowledge that theoretical guarantees regarding its impact on performance are difficult to establish. To address this, we subject the metrics to extensive stress testing. This includes evaluations on real data presented in the main text, as well as on the synthetic benchmark proposed by Räisä et al. (2025) in Appendix H.
>
>
> ### Clarification
> > Questions:
> > - L122-123, the sentence "In high-dimensional spaces, where measuring density is impractical, *Density* instead uses $k$-NN balls as a proxy" seems off here. Can you clarify?
>
> Yes! In high-dimensional spaces, support and density estimations are unreliable. Lacking a better alternative, the *Density* metric relies on *distances* for fidelity computation, using $k$-NN balls.
>
> We clarified this in the revised version (lines 122-123):
> >>As approximating density is impractical in high-dimensional spaces, the *Density* metric instead uses distances through $k$-NN balls.
>
> ### Fidelity and coverage trade-off
>
> > - Fidelity and coverage are usually known to be a trade-off [2] for the same model. Clipped metrics in Figure 6 do not reflect this (, although the comparisons are among different models). Can you further verify this?
> > [2] Li, Shuangqi, et al. "Controlling the Fidelity and Diversity of Deep Generative Models via Pseudo Density." Transactions on Machine Learning Research.
>
> To verify that *Clipped Density* and *Clipped Coverage* capture the fidelity-coverage trade-off, we evaluated them on LSUN Bedroom StyleGAN data (Karras et al., 2019) shared by the original authors with varying truncation levels. See Appendix M.5, Figure 27.
>
> For GANs, the truncation trick samples from a truncated latent space to improve fidelity at the expense of coverage. A smaller truncation value $\psi$ restricts sampling to a smaller, denser region. $\psi=1.0$ corresponds to no truncation, matching the StyleGAN model we evaluated in the main text (Figure 6).
>
> From no truncation ($\psi=1.0$) to moderate truncation ($\psi=0.7$), both scores increase. However, further truncation to $\psi=0.5$ causes *Clipped Density* to rise higher while *Clipped Coverage* decreases, illustrating the trade-off.
>
> We investigated this initial rise of *Clipped Coverage* in Figure 28 by examining sample-wise score histograms. From $\psi=1.0$ to $\psi=0.7$, real samples in high-density regions that were barely covered become fully covered. Then at $\psi=0.5$, many real samples drop to 0 coverage as parts of the distribution are ignored. Thus, the initial rise in *Clipped Coverage* score is due to a more intense coverage of dense real regions.

---

> > ### Author Response · Authors · 2025-11-21
> > **Answer Part 2/2**
> >
> > ### Values exceeding 1
> >
> > > - In Eq. (7), why should we normalize by $ClippedDensity_{real}$? When would $\frac{ClippedDensity_{unnorm}}{ClippedDensity_{real}}$ be larger than 1 and how should we interpret it? Why can it be safely clipped to 1?
> >
> > As defined, the value of $\text{ClippedDensity}\_{\text{real}}$ is dataset-dependent. However, it represents the ideal value of $\text{ClippedDensity}\_{\text{unnorm}}$, i.e. the one achieved by the real set itself. Because of the dataset-dependency, a $\text{ClippedDensity}\_{\text{unnorm}}$ score of 0.4 could be a perfect score on one dataset and poor on another. Normalizing by dividing by the real score, we standardize the ideal value to 1 regardless of the dataset, which enhances interpretability.
> >
> > Consequently, $\text{ClippedDensity}$ scores have a clear **reference point**. A score of 0.4 corresponds to the performance achieved by a mixture of 40\% real points and 60\% bad points.
> >
> >
> > In Appendix M.6 (Figure 29), we investigated when $\text{ClippedDensity}$ could exceed 1. In our evaluation of real generated datasets (Figure 6), only one set exceeded 1: the guided DiT-XL-2 model on ImageNet.
> >
> > The sample-wise histograms show that guidance increases the proportion of synthetic samples with perfect fidelity, surpassing that of real samples. Similar to StyleGAN truncation, which can also yield scores above 1, $\text{ClippedCoverage}$ improves as covered points become fully covered while uncovered points remain stable. This reflects the similar effects of guidance and truncation: sampling favors high-density regions and ignores low-density ones.
> >
> > A $\text{ClippedDensity}$ above 1 thus occurs when sampling avoids low-density regions, effectively targeting a **filtered** real distribution. While using a filtered high-quality test set might be better practice, the metric correctly reflects that the synthetic set achieves higher quality than the unfiltered real reference.
> >
> > Scores above 1 indicate that the generative model does not target the full real distribution but rather a subset of high-density regions. Since our goal is to evaluate how well the model matches the real set, scores exceeding 1 represent a deviation from this objective (often at the cost of coverage). Clipping to 1 enforces the interpretation that the ideal fidelity is matching the real data exactly, not surpassing it by ignoring difficult regions. However, we will provide an option in our code to disable this clipping for practitioners who wish to analyze this specific behavior.
> >
> > Outside of this specific context, scores do not exceed significantly 1 (see Figure 30).
> >
> >
> >
> > > - It seems that Clipped Coverage is not guaranteed to be <= 1. Is this correct? How should we interpret it when it is larger than 1?
> >
> > *Clipped Coverage* is strictly bounded by 1 by construction. The calibration function $g$ maps the unnormalized $\text{ClippedCoverage}_\text{unnorm}$ score $s$ to a value $g(s)= 1 - \frac{i(s)}{M}$. Since $i(s)$ represents the rank index of $s$ in the sorted list of expected values (of length $M+1$), $i(s)$ is an integer between $0$ and $M$.  Consequently, the term $\frac{i(s)}{M}$ lies in $[0, 1]$, and thus $g(s)$ also lies in $[0, 1]$.
> >
> > ### References
> >
> > - Tero Karras, Samuli Laine, and Timo Aila. A style-based generator architecture for generative adversarial networks. In Proceedings of the IEEE/CVF conference on computer vision and pattern recognition, pp. 4401–4410, 2019.
> > - Ossi Räisä, Boris van Breugel, and Mihaela van der Schaar. Position: All current generative fidelity and diversity metrics are flawed. In Forty-second International Conference on Machine Learning Position Paper Track, 2025.

---

### Official Review · Reviewer_4Rqw · 2025-11-01

**Soundness:** 3
**Presentation:** 3
**Contribution:** 2
**Rating:** 4
**Confidence:** 4

**Summary:**

This paper introduces two new metrics, Clipped Density and Clipped Coverage, to address failings in the evaluation of generative models. The authors argue that existing metrics for fidelity and coverage suffer from two main problems: 1) a lack of robustness to outliers in both the real and synthetic datasets, and 2) an inability to provide an interpretable, absolute score.
The proposed metrics address these issues through new clipping and calibration strategies:
1. Clipping: Clipped Density (fidelity) prevents real-data outliers from inflating scores by clipping k-NN ball radii to the median distance. Both metrics also clip the contribution of any individual sample to a score of 1, preventing over-represented modes from masking the presence of "bad" samples.
2. Calibration: The metrics are calibrated to provide an absolute, more interpretable score. A score of $x$ is designed to be equivalent to the performance of a dataset with a proportion $x$ of "good" samples and $1-x$ of "bad" samples . For Clipped Coverage, this is achieved via a theoretical derivation (Lemma 1) and a resulting correction function ; for Clipped Density, this is achieved by normalizing by the real data's score on itself .
The authors demonstration on synthetic densities and through empirical validation on DINOv2-embedded image datasets that these two metrics are robust to outliers and provide interpretable absolute scores that behave linearly as sample quality degrades.

**Strengths:**

Strengths:
-  The paper addresses an important and recognized problem: the lack of reliable, robust, and interpretable metrics for generative models.
- The paper presents a well-illustrated analysis of the failure modes of a chosen class of metric: k-NN density-based metrics (like Precision, Density, Coverage) is clear. The figures effectively illustrate the specific problems of outlier sensitivity and non-linear response that the paper aims to solve.
- The "clipping" mechanisms are simple and intuitive and directly target the demonstrated failure modes. Clipping radii addresses outlier-driven inflation, and clipping sample contributions addresses mode collapse or over-representation.
- The authors have made some efforts to demonstrate the performances of their metrics on modern, high-dimensional datasets like ImageNet, LSUN, and FFHQ, which are the standard benchmarks for "real-world" generative models

**Weaknesses:**

1. Overstated Claims and Missing Key Baselines: The paper's premise that "all existing... metrics are flawed" and that "no metric offers this property" (absolute interpretability) is an overstatement. The paper's analysis is confined almost entirely to kNN density-based metrics, while ignoring a relevant body of work on sampling-based evaluation: arXiv:2402.04355 and arXiv:2302.03026 similarly use distance metrics to probe the underlying density but don’t rely on kNN density estimation; there is also no comparison to other standard metrics used in the literature, to help the audience how the proposed metrics differ on specific tasks (Fréchet Inception Distance - FID  arXiv:1706.08500, Feature Likelihood Divergence  - FLD arXiv:2302.04440) Failing to refer to and compare against this (FID is mentioned, but just in the intro, no comparisons are actually made on on problems) related work significantly weakens the paper's claim to novelty, and makes it hard to understand how its performances compare to existing evaluation options.
2. The paper's main technical contribution—the calibration of Clipped Coverage—could be presented more clearly.

   - Lack of Clarity: The paper never explicitly states that its "absolute interpretability" is a property of the metric's expected value, not its value from a single draw. This implies that bootstrapping (running the metric many times) is required to get an interpretable score, but this is never explicitly stated or analyzed (how stable is the score as a function of the number of samples and the dimensionality of the problem).

   - Statistical Ambiguity: As a result of sampling variance, a single "lucky" draw from a perfect model (or even an imperfect model) could yield an uncalibrated score $s > f_{\text{expected}}(0)$, which would result in a calibrated score greater than 1.0, breaking the "proportion" interpretation. This could happen even if the score is bootstrapped a small number of times (since the variance of the score is not studied theoretically, it’s hard to assess how many times the bootstrapping needs to be done for a specific application). The paper provides no analysis of the metric's variance or the number of bootstrap samples needed for a stable estimate.

   - Lack of theoretical guarantees: The paper does not present theoretical guarantees that the proposed metric has important desirable properties in the theoretical limit of infinite number of samples or with infinite number of re-drawing of the samples and recalculating the score, like a proof of sufficiency or a proof of consistency.

3. Limited Scope of Evaluation (Failure Modes & Modalities):
   - Limited evaluation of the sensitivity of the metrics: The "sanity checks" are limited to gross failures like complete mode dropping or injection of very easily detectable out of distribution samples. For scientific applications, a much more relevant test is sensitivity to subtle failures, such as missing rare modes, failing to capture intra-mode structure, or subtle corruption to the samples.
    - Missing Modalities: All experiments are on images or simple Gaussian data. The metric is distance-based and should be applicable to any data modality. The paper would be far more convincing if it demonstrated this utility on other common data types, such as tabular data or time-series.

4. All the high-dimensional experiments presented have a dependence on  a black box feature extractor. The robustness claims are undermined by this, as it’s not clear if it’s the metric that’s robust, or the embedding of the data through DINOv2. This creates blinds spots for many real world applications:

   - Foundation models like DINOv2 are trained for invariance to noise, blur, and other augmentations, but for some applications, for example in science, it’s important to test weather those exact nuisance parameters are correctly captures by the model.

   -  Moreover, the claim of the paper is that the metric could be generally useful across domains, but DinoV2 is specifically trained on natural images and there is no guarantees that it extracts features that are useful in other domains, e.g. scientific data like fluids dynamics simulation, climate models, astrophysical simulations, that have different underlying statistics and correlate scales differently. The experiments have shown that in high-dimensions, the metrics capture model’s realism within the DINOv2 feature space, but that might not correlate with true scientific fidelity.

5. There is a significant limitation of the metrics that is not discussed: The paper's design goal is "robustness to outliers," which it achieves by making the metrics insensitive to the magnitude of a sample's "badness" . A synthetic sample 1000 units away from the data manifold is scored as a 0, which is the same score a sample only 10 units away might receive. While this makes the metric sensitive to the proportion of bad samples, it ignores the severity. For the "high-stakes applications" mentioned in the introduction (e.g., healthcare), this is a critical flaw. A model that produces one catastrophic outlier is far more dangerous than a model that produces many mediocre samples. The paper presents this insensitivity as a feature without discussing the significant trade-off in its limitations.

**Questions:**

1. How does the variance of the coverage score scale as a function of the number of samples, number of bootstrap performed, and dimensionality of the problem, for various applications?

2. Your k-NN metrics are susceptible to the curse of dimensionality, which is why you use DINOv2. But what failure modes does this introduce? For example, how would your DINOv2-embedded metrics perform on a "bad" model that correctly learns the locations and densities of modes but gets the correlations within those modes wrong?

3. How does the metrics compare to modern metrics like TARP, PQMass, FLD and FID on various realistic tasks?

3. Your "absolute interpretability" relies on the metric's expected value, not a single score. How many bootstrap samples are needed to get a stable estimate of this mean, and how does this required number of samples scale with data dimension?

4. How sensitive are your metrics to the choice of metric distance between samples?

5. You motivate your metrics as being robust to outliers, which in practice means they are insensitive to the magnitude of a sample's "badness." How bad do mediocre samples have to be for an ‘ok but not great’ model to be scored worse than one that sometimes produces catastrophic samples? How does this depend on the choice of embedding model?

6. Can you discuss more in detail failure modes of your tests, for example "bad" model that correctly learns the locations and densities of modes but gets the correlations within those modes wrong (e.g., swapping a /-shaped mode for a \\-shaped mode)?

---

> ### Author Response · Authors · 2025-11-21
> **Answer Part 1/5**
>
> Thank you for the detailed and thorough review. We address all your comments bellow.
>
> ### Fidelity and diversity metrics
>
> > Weakness 1. Overstated Claims and Missing Key Baselines: The paper's premise that "all existing... metrics are flawed" and that "no metric offers this property" (absolute interpretability) is an overstatement. The paper's analysis is confined almost entirely to kNN density-based metrics, while ignoring a relevant body of work on sampling-based evaluation: arXiv:2402.04355 and arXiv:2302.03026 similarly use distance metrics to probe the underlying density but don’t rely on kNN density estimation; there is also no comparison to other standard metrics used in the literature, to help the audience how the proposed metrics differ on specific tasks (Fréchet Inception Distance - FID arXiv:1706.08500, Feature Likelihood Divergence - FLD arXiv:2302.04440) Failing to refer to and compare against this (FID is mentioned, but just in the intro, no comparisons are actually made on on problems) related work significantly weakens the paper's claim to novelty, and makes it hard to understand how its performances compare to existing evaluation options.
>
> > Question 3. How does the metrics compare to modern metrics like TARP, PQMass, FLD and FID on various realistic tasks?
>
> Thank you for this comment, which points to a misunderstanding. We would like to clarify the initial statement: we do not argue that **all** existing metrics are flawed, but refer to the claim made in the independent position paper by Räisä et al. (2025): "Position: All Current Generative **Fidelity and Diversity** Metrics are Flawed".
>
> Generative fidelity and diversity metrics (Naeem et al., 2020) are **pairs** of metrics, that measure the two **distinct** aspects : the first metric of each pair measures fidelity (e.g., Precision or Density), and the second metric of the pair measures diversity (e.g., Recall or Coverage). For example, FID, FLD and PQMass are **not** fidelity and diversity metrics, as they evaluate both aspects simultaneously.
>
> Fidelity and diversity metrics aim to disentangle these two aspects in order to investigate more precisely the failures of generative models. We follow this line of work: we do not compare only to k-NN based metrics, but to _all_ fidelity and diversity metrics.
>
> In the revised version, we clarified this in the introduction (changes in bold): lines 57-63
>
> >> To address this issue, the quality of synthetic data can be broken down into at least two core concepts, fidelity and coverage, **and measured separately with a pair of metrics.** Fidelity **metrics** assess how similar each synthetic sample is to the input data (Naeem et al., 2020). Conversely, coverage **metrics** measure the extent to which synthetic samples represent the distribution of real data, taking into consideration how the rarity or commonness of real data is reflected in the synthetic samples. However, a recent position paper by Räisä et al. (2025) argues that all existing fidelity and diversity metrics are flawed, highlighting an urgent need for new metrics that address these shortcomings.
>
> and line 75:
>
> >> Critically, Räisä et al. (2025) notes that, currently, no **fidelity or coverage** metric offers this property.

---

> > ### Author Response · Authors · 2025-11-21
> > **Answer Part 2/5**
> >
> > ### Calibration Stability
> >
> > > Weakness 2. The paper's main technical contribution—the calibration of Clipped Coverage—could be presented more clearly.
> > > - Lack of Clarity: The paper never explicitly states that its "absolute interpretability" is a property of the metric's expected value, not its value from a single draw. This implies that bootstrapping (running the metric many times) is required to get an interpretable score, but this is never explicitly stated or analyzed (how stable is the score as a function of the number of samples and the dimensionality of the problem).
> > >  - Statistical Ambiguity: As a result of sampling variance, a single "lucky" draw from a perfect model (or even an imperfect model) could yield an uncalibrated score $s > f_{expected}(0)$, which would result in a calibrated score greater than 1.0, breaking the "proportion" interpretation. This could happen even if the score is bootstrapped a small number of times (since the variance of the score is not studied theoretically, it’s hard to assess how many times the bootstrapping needs to be done for a specific application). The paper provides no analysis of the metric's variance or the number of bootstrap samples needed for a stable estimate.
> >
> > > Question 1. How does the variance of the coverage score scale as a function of the number of samples, number of bootstrap performed, and dimensionality of the problem, for various applications?
> >
> > > Question 4. Your "absolute interpretability" relies on the metric's expected value, not a single score. How many bootstrap samples are needed to get a stable estimate of this mean, and how does this required number of samples scale with data dimension?
> >
> > Thank you for raising these points. In the revised version, we address these concerns about calibration stability with an empirical study in Appendix M.1.
> >
> > Specifically, we reproduced the evaluation of mixtures of good and bad samples using the high-dimensional ImageNet dataset, as shown in Figure 21. Both clipped metrics decrease linearly with the proportion of bad samples. The standard deviation across 10 repetitions with different subsets is at most $0.0056$ for *Clipped Density* and $0.0082$ for *Clipped Coverage*.
> >
> > In Figure 22, we further analyze how the standard deviation scales with sample size, showing it decreases as $1/\sqrt{N}$ for both metrics and drops below $0.01$ for $N=50000$.
> >
> > While our evaluations rely on point estimates, in practice, with 50000 samples, the standard deviation remains below 0.01. Therefore, we do not recommend bootstrapping, as the interpretation remains valid without it.
> >
> > >  - Lack of theoretical guarantees: The paper does not present theoretical guarantees that the proposed metric has important desirable properties in the theoretical limit of infinite number of samples or with infinite number of re-drawing of the samples and recalculating the score, like a proof of sufficiency or a proof of consistency.
> >
> > You are correct that our current work does not include a theoretical analysis of clipped metrics in the infinite sample limit. We have acknowledged this as a limitation in the revised version of the paper lines 479-480:
> >
> > >> Despite these improvements, limitations remain. *Clipped Density* and *Clipped Coverage* build on an accepted benchmark of progressively passed tests, but it is not exhaustive, and there might be missing cases. **Furthermore, we did not provide a theoretical analysis of the metrics in the infinite sample limit.** Additionally, while we evaluate fidelity and coverage, other aspects matter, such as memorization[...]

---

> ### Author Response · Authors · 2025-11-21
> **Answer Part 3/5**
>
> ### Testing more failure modes
>
> > Weakness 3. Limited Scope of Evaluation (Failure Modes & Modalities):
> >  - Limited evaluation of the sensitivity of the metrics: The "sanity checks" are limited to gross failures like complete mode dropping or injection of very easily detectable out of distribution samples. For scientific applications, a much more relevant test is sensitivity to subtle failures, such as missing rare modes, failing to capture intra-mode structure, or subtle corruption to the samples.
>
> > Question 2. Your k-NN metrics are susceptible to the curse of dimensionality, which is why you use DINOv2. But what failure modes does this introduce? For example, how would your DINOv2-embedded metrics perform on a "bad" model that correctly learns the locations and densities of modes but gets the correlations within those modes wrong?
>
> > Question 7. Can you discuss more in detail failure modes of your tests, for example "bad" model that correctly learns the locations and densities of modes but gets the correlations within those modes wrong (e.g., swapping a /-shaped mode for a \\-shaped mode)?
>
> We agree that these are important questions. We added in the revised version (Appendix M.2, Figure 23) a detection test of **rare modes**, following Appendix F.2 from Lemos et al. (2025). The main modes are eight CIFAR-10 classes, while a rare mode (trucks) is progressively introduced into the real set only. *Clipped Coverage* correctly decreases as the uncovered rare mode is introduced, even when it represents only 0.00125% of the real set. *Clipped Density* remains stable since synthetic samples still lie within valid real modes.
>
> We had already included in Appendix G the sensitivity of our fidelity metric to **simple degradations** (including posterize, light blur, center crop, JPG 90, ...).
>
>
> To test an incorrect **intra-mode structure**, we sampled $N=10000$ points from centered Gaussians: /-shaped for the real data and \\-shaped for the synthetic data. The corresponding covariance matrices have a block structure:
> $\Sigma_{\text{real}} = \begin{pmatrix} 1+\sigma^2 & 1 \\\\ 1 & 1+\sigma^2 \end{pmatrix} \otimes I_{d/2}$
> $\Sigma_{\text{synthetic}} = \begin{pmatrix} 1+\sigma^2 & -1 \\\\ -1 & 1+\sigma^2 \end{pmatrix} \otimes I_{d/2}$
> In the revised Appendix M.2 (Figure 24), we show the first two components of these sets for varying values of $\sigma$, which controls the width of the / and \\ shapes. Regardless of $\sigma$, both *Clipped Density* and *Clipped Coverage* drop to 0 as the dimension increases, because the intersection volume between the distributions becomes negligible.
>
> Similarly, if the mismatch were confined to only 2 dimensions (e.g., incorrect block only in dimensions 1 and 2, with identical blocks in the rest), the metrics would fail. This scenario is very close to the "**One Disjoint Dim. + Many Identical Dim**" test from Räisä et al. (2025), which we tested in Appendix H (Tables 7 and 8). All fidelity and coverage metrics failed this test, as the influence of one or two mismatched dimensions on distances decreases with increasing dimensionality, making such mismatches harder to detect. If certain dimensions are more important and require monitoring, we recommend scaling them up before computing the metrics.

---

> > ### Author Response · Authors · 2025-11-21
> > **Answer Part 4/5**
> >
> > ### Adding modalities: Time Series and Music
> > > - Missing Modalities: All experiments are on images or simple Gaussian data. The metric is distance-based and should be applicable to any data modality. The paper would be far more convincing if it demonstrated this utility on other common data types, such as tabular data or time-series.
> >
> > Thanks for this excellent suggestion. Following Lemos et al. (2025, Appendix F.3), we evaluated *Clipped Density* and *Clipped Coverage* on toy **time series** in our revised Appendix M.3 (Figure 25). The signal is defined as $A \cos(t) + \eta(t)$, where $\eta(t)$ is unit Gaussian noise, with 100 time points per sample and 5000 samples per set. The real set has zero amplitude ($A=0$), while the synthetic set has varying amplitude. The goal is to detect this discrepancy. Both metrics decrease as the signal amplitude increases, dropping below the maximum slightly after an amplitude of 0.1. Some instability is observed due to the limited sample size of 5000.}
> >
> >
> > We also performed tests on **music** samples in Appendix M.4. We used the Free Music Archive large dataset, comprising 30s clips from 106574 tracks (Defferrard et al., 2017). Following Gui et al. (2024), who compared 10 embedding models for music, we used the CLAP model (Elizalde et al., 2023).  As it extracts 1024-dimensional embeddings from 7s clips, we selected the middle 7s of each track.
> >
> > An initial evaluation on balanced real/synthetic splits of 50000 samples each yielded a *Clipped Density* of 0.988 and a *Clipped Coverage* of 0.998. When splitting by genre, scores decreased to 0.943 for *Clipped Density* and 0.875 for *Clipped Coverage*. This reduction is moderate but expected, as music tracks span multiple genres, yet we split based on a single genre per track (a track's other associated genres might correspond to the other dataset).
> >
> > Finally, we evaluated mixtures of good and bad samples (Figure 26), where bad samples consisted of noise MP3 files. We used $N=50000$ balanced CLAP embeddings per set. Both metrics decreased linearly with the proportion of bad samples, **confirming the absolute calibration** on this high-dimensional music dataset.
> >
> >
> > ### Embedding model
> >
> > > Weakness 4. All the high-dimensional experiments presented have a dependence on a black box feature extractor. The robustness claims are undermined by this, as it’s not clear if it’s the metric that’s robust, or the embedding of the data through DINOv2. This creates blinds spots for many real world applications:
> > >     - Foundation models like DINOv2 are trained for invariance to noise, blur, and other augmentations, but for some applications, for example in science, it’s important to test weather those exact nuisance parameters are correctly captures by the model.
> > >     - Moreover, the claim of the paper is that the metric could be generally useful across domains, but DinoV2 is specifically trained on natural images and there is no guarantees that it extracts features that are useful in other domains, e.g. scientific data like fluids dynamics simulation, climate models, astrophysical simulations, that have different underlying statistics and correlate scales differently. The experiments have shown that in high-dimensions, the metrics capture model’s realism within the DINOv2 feature space, but that might not correlate with true scientific fidelity.
> >
> > > Question 5. How sensitive are your metrics to the choice of metric distance between samples?
> >
> > We agree that results depend on the choice of embedding model. However, embedding models are necessary not only to handle high dimensionality but also to obtain meaningful distances. For instance, the $l_2$-distance in pixel space often fails to capture perceptual qualities like blurriness. As **feature extractors**, embedding models allow for computing distances between features that are more relevant than raw observations. Consequently, the application of clipped metrics to any domain requires distances that are semantically meaningful.
> >
> > For image data, DINOv2 is recognized as a robust embedder. We use it in all our image-based experiments, considering it an integral part of the preprocessing pipeline. Applying clipped metrics to other domains requires domain-specific knowledge to preprocess data such that distances reflect true similarities. Similarly, we regard the choice of distance as domain knowledge tied to the representation space.
> >
> > Therefore, to apply clipped metrics to music data, we relied on another modality-specific feature extractor: CLAP.

---

> > > ### Author Response · Authors · 2025-11-21
> > > **Answer Part 5/5**
> > >
> > > ### Sample-level metrics
> > >
> > > > Weakness 5. There is a significant limitation of the metrics that is not discussed: The paper's design goal is "robustness to outliers," which it achieves by making the metrics insensitive to the magnitude of a sample's "badness" . A synthetic sample 1000 units away from the data manifold is scored as a 0, which is the same score a sample only 10 units away might receive. While this makes the metric sensitive to the proportion of bad samples, it ignores the severity. For the "high-stakes applications" mentioned in the introduction (e.g., healthcare), this is a critical flaw. A model that produces one catastrophic outlier is far more dangerous than a model that produces many mediocre samples. The paper presents this insensitivity as a feature without discussing the significant trade-off in its limitations.
> > >
> > > > Question 6. You motivate your metrics as being robust to outliers, which in practice means they are insensitive to the magnitude of a sample's "badness." How bad do mediocre samples have to be for an ‘ok but not great’ model to be scored worse than one that sometimes produces catastrophic samples? How does this depend on the choice of embedding model?
> > >
> > > The Clipped metrics are sample-level metrics (Alaa et al., 2022): they provide a score per sample. *Clipped Density* assigns a fidelity score to each synthetic point, while *Clipped Coverage* assigns a coverage score to each real point. In scenarios where catastrophic generated samples are problematic, sample-level scores allow for the removal of all zero-quality samples. Thus, although metrics that provide only an overall score must carefully handle catastrophic samples, in our case, where sample-level scores are available, such samples can simply be removed if they pose a risk.
> > >
> > > ### References
> > >
> > > * Ahmed Alaa, Boris Van Breugel, Evgeny S Saveliev, and Mihaela van der Schaar. How faithful is your synthetic data? sample-level metrics for evaluating and auditing generative models. In International Conference on Machine Learning, pp. 290–306. PMLR, 2022.
> > > * Michaël Defferrard, Kirell Benzi, Pierre Vandergheynst, and Xavier Bresson. FMA: A dataset for music analysis. In Sally Jo Cunningham, Zhiyao Duan, Xiao Hu, and Douglas Turnbull (eds.), Proceedings of the 18th International Society for Music Information Retrieval Conference, ISMIR 2017, Suzhou, China, October 23-27, 2017, pp. 316–323, 2017.
> > > * Benjamin Elizalde, Soham Deshmukh, Mahmoud Al Ismail, and Huaming Wang. CLAP learning audio concepts from natural language supervision. In IEEE International Conference on Acoustics, Speech and Signal Processing ICASSP 2023, Rhodes Island, Greece, June 4-10, 2023, pp. 1–5. IEEE, 2023.
> > > * Azalea Gui, Hannes Gamper, Sebastian Braun, and Dimitra Emmanouilidou. Adapting frechet audio distance for generative music evaluation. In IEEE International Conference on Acoustics, Speech and Signal Processing, ICASSP 2024, Seoul, Republic of Korea, April 14-19, 2024, pp. 1331–1335. IEEE, 2024.
> > > * Pablo Lemos, Sammy Nasser Sharief, Nikolay Malkin, Salma Salhi, Connor Stone, Laurence Perreault Levasseur, and Yashar Hezaveh. Pqmass: Probabilistic assessment of the quality of generative models using probability mass estimation. In The Thirteenth International Conference on Learning Representations, ICLR 2025, Singapore, April 24-28, 2025, 2025.
> > > * Muhammad Ferjad Naeem, Seong Joon Oh, Youngjung Uh, Yunjey Choi, and Jaejun Yoo. Reliable fidelity and diversity metrics for generative models. In International Conference on Machine Learning, pp. 7176–7185. PMLR, 2020.
> > > * Ossi Räisä, Boris van Breugel, and Mihaela van der Schaar. Position: All current generative fidelity and diversity metrics are flawed. In Forty-second International Conference on Machine Learning Position Paper Track, 2025.

---

### Meta-Review · Area_Chair_5mhy · 2026-01-04

**Summary:**

This paper introduces Clipped Density and Coverage (CDC), a refined evaluation metric designed to improve how we measure the performance of generative models. Their fix is intuitive: they clip individual sample contributions and nearest-neighbor radii to stop a few "bad" samples or over-represented modes from skewing the final score. While reviewers initially pointed out that in the a lack of theoretical proofs and a heavy reliance on image-based feature extractors, the authors pushed back with a very thorough rebuttal. They added new experiments on audio and time-series data, conducted stability tests, and showed a high correlation with human judgment, convincing most of the committee that these metrics are a practical step forward for the field.


All the reviewers agree that this paper makes incremental contributions.  The authors should incorporate the suggestions from the reviewers, e.g., rephrase the sentences suggested by reviewer 4Rqw, cite the paper suggested by reviewer 6bfz to further cross-validate the soundness of the proposed method. These changes can be made in a short period; hence, the AC recommends acceptance.

**Reviewer Concerns:**

***Claim***. The authors have revised their statement in the paper.

***modalities***. The authors demonstrate their application to the audio.

***Fidelity and coverage trade off*** The authors have shown that their method.

***Incremental Novelty***. Although the work indeed makes incremental contributions, the paper is complete and provides experimental guidance to the community. Overall, it is worth publishing in venues like ICLR.

**Reviewer Scores:**

The reviewer 4Rqw starts with a rating of 4, and most of the weaknesses were addressed by the rebuttal.

The other reviewers gave an initial 6, and reviewer GJ4y explicitly expressed that his/her concerns were fully addressed and would like to further increase the score. The concerns from the other two reviewers are handled well by the authors.

---

### Decision · Program_Chairs · 2026-01-26

Accept (Poster)